# The concentration, source and deposition flux of ammonium and nitrate in atmospheric particles during dust events at a coastal site in northern China

Jianhua Qi[1], Xiaohuan Liu[1], Xiaohong Yao[1], Ruifeng Zhang[1], Xiaojing Chen[1], Xuehui Lin[2], Huiwang Gao[1], Ruhai Liu[1]

[1]Key Laboratory of Marine Environment and Ecology, Ministry of Education, Ocean University of China, Qingdao, 266100, China

[2]Qingdao Institute of Marine Geology, Qingdao, 266100, China

Correspondence to: Jianhua Qi (qjianhua@ouc.edu.cn)

**Abstract.** Asian dust has been reported to carry anthropogenic reactive nitrogen during transport from source areas to the oceans. In this study, we attempted to characterize $NH_4^+$ and $NO_3^-$ in atmospheric particles collected at a coastal site in northern China during spring dust events from 2008 to 2011. Based on the mass concentrations of $NH_4^+$ and $NO_3^-$ in each total suspended particle (TSP) sample, the samples can be classified into increasing or decreasing types. In Category 1, the concentrations of $NH_4^+$ and $NO_3^-$ were 20%-440% higher in dust day samples relative to samples collected immediately before or after a dust event. These concentrations decreased by 10-75% in the dust day samples in Categories 2 and 3. Back trajectory analysis suggested that multiple factors such as the transport distance prior to the reception site, the mixing layer depth on the transport route and the residence time across highly polluted regions, might affect the concentrations of $NH_4^+$ and $NO_3^-$. $NH_4^+$ in the dust day samples was likely either in the form of ammonium salts existing separately with dust aerosols or as the residual of incomplete reactions between ammonium salts and carbonate salts. $NO_3^-$ in the dust day samples was attributed to various formation processes during the long-range transport. The positive matrix factorization (PMF) receptor model results showed that the contribution of soil dust increased from 23% to 36% on dust days with decreasing contributions from local anthropogenic inputs and associated secondary aerosols. The estimated deposition flux of $N_{NH4++NO3-}$ varied greatly from event to event, e.g., the dry deposition flux of $N_{NH4++NO3-}$ increased by 9-285% in Category 1, but decreased by 46%-73% in Category 2. In Categories 3, the average dry deposition fluxes of particulate nitrate and ammonium decreased by 46% and increased by 10%, respectively, leading

to 11-48% decrease in the fluxes of $N_{NH4++NO3-}$.
Keywords: aerosols, nitrogen, dust, source apportionment, dry deposition flux
**1 Introduction**
Reactive nitrogen carried in dust particles can be transported over a long distance, and the
atmospheric nitrogen deposition in oceans has been recognized as an important external source of the
nitrogen supporting phytoplankton growth (Duce et al., 2008; Zhang et al., 2010b). This hypothesis has
been evaluated through incubation experiments,  in situ experiments, and the use of satellite
observational data (Banerjee and Kumar 2014; Guo et al., 2012; Liu et al., 2013; Shi et al., 2012; Tan
and Wang, 2014). However, the process is dynamic due to the worldwide changing emissions of $NO_x$
and $NH_3$ in the last few decades. For example, China and most of the developing countries in Asia
experienced a large increase in emissions of $NH_3$ and $NO_x$ while a substantial decrease in emissions
occurred in Europe over the last three decades (Grice et al., 2009; Liu et al., 2017; Ohara et al., 2007;
Skjøth and Hertel, 2013). The change would affect the nitrogen carried by dust particles to some extent,
and updated studies are thereby essential.
Asian dust is one of three largest dust sources on earth. Asian dust has been reported to not only
frequently cross over the mainland and the China Seas, but also to occasionally reach the remote
northern Pacific Ocean or North America (Creamean 2013; Tan and Wang, 2014; Van Curen and Cahill,
2002; Zhang and Gao, 2007). In an extreme case, Asian dust was found to be transported more than one
full circuit around the globe in approximately 13 days (Uno et al 2009). During the long-range
transport, dust particles may mix with anthropogenic air pollutants and consequently undergo
complicated chemical reactions (Cui et al., 2009; Li et al., 2014; Ma et al., 2012; Wang et al., 2011;
Wang et al., 2016b; Wang et al., 2017a; Xu et al., 2014; Yang et al., 2002). For example, a few studies
have shown that the concentrations of atmospheric particulate $NO_3^-$ and $NH_4^+$ on dust storm days were
2-5 times larger than those prior to the events in Beijing (Liu et al., 2014; Liu and Bei, 2016).
Fitzgerald et al. (2015) found that almost all Asian dust events observed in Korea contained
considerable amounts of nitrate. However, Zhang et al. (2010a) reported an interesting result, i.e., the
concentrations of $NO_3^-$ and $NH_4^+$ were lower during strong dust storm events than weak dust events. A
high uncertainty appeared to exist for carrying amount of reactive nitrogen by dust particles.
A few contradictory results were also reported in the literature, which made the scientific issue even
more complicated. For example, the concentration of $NO_3^-$ in atmospheric aerosols on dust days was
significantly lower in comparison to the concentration measured immediately before or after the event
at a rural site in Yulin near the Asian dust source region (Wang et al., 2016b). The phenomenon was
also observed in Shanghai, a mega city at a few thousands of kilometers from dust source zones in
China, and more downwind sites (Kang et al., 2013; Li et al., 2014; Wang et al., 2013).
Inorganic nitrogen reportedly contributed to ~80% of the total water-soluble nitrogen (TDN) in
atmospheric particles collected over the Yellow Sea and in Qingdao (Shi et al., 2012). In the region, the
dry deposition flux of the inorganic nitrogen accounted for more than 75% for the TDN (Qi et al.,
2013). When deposited to the ocean via atmospheric dry deposition, inorganic nitrogen has great
impact on marine productivity due to its bioavailability. To update and improve our knowledge on
reactive nitrogen carried by dust particles, we collected atmospheric aerosol particles during and prior
to (or post, but only when no sample was collected prior to dust events) at a coastal site adjacent to the
Yellow Sea in each spring of 2008-2011. The concentrations of $NO_2^-$, $NO_3^-$, $NH_4^+$ and other
components were determined for analysis. In this study, we focused on nitrate and ammonium by
excluding nitrite because of its very low concentration. We first characterized the concentrations of
$NH_4^+$ and $NO_3^-$ in dust samples by comparing them with the values in atmospheric particles measured
either prior to or post the event. We then conducted source apportionment to quantify their sources.
Finally, we calculated and discussed the deposition flux of atmospheric particulate $NH_4^+$ and $NO_3^-$
during dust events.
**2 Experimental methods**
**2.1 Sampling**
Fig. 1 shows the sampling site, which is situated at the top of a coastal hill (Baguanshan) in Qingdao
in northern China (36 ° 6' N, 120 ° 19' E, 77 m above sea level) and is approximately 1.0 km from the
Yellow Sea to the east. A high-volume air sampler (Model KC-1000, Qingdao Laoshan Electronic
Instrument Complex Co., Ltd., China) was set up on the roof of a two-story office building to collect
total suspended particle (TSP) samples on quartz microfiber filters (Whatman QM-A) at a flow rate of
1 $m^3$/min. Prior to sampling, the filters were heated at 450 ℃ for 4.5 hrs to remove organic compounds.
Our sample collection strategy involved collecting dust samples representing long-range transported
particles. We followed the definition of dust events adopted in the regulations of surface meteorological
observations of China (CMA, 2004; Wang et al., 2008) and identified dust events based on the
meteorological records (Weather Phenomenon) of Qingdao from the Meteorological Information
Comprehensive Analysis and Process System (MICAPS) of the China Meteorological Administration.
Due to no dust events lasting over 12 hrs (Lee et al., 2015; Su et al., 2017; Zhang et al., 2007), we
collected one dust sample with a 4-hr duration in a day. The sampling for dust particles started only
when the measured $PM_{10}$ mass concentration in Qingdao (http://www.qepb.gov.cn/m2/) and the
forecasted dust mass over Asia (http://www-cfors.nies.go.jp/~cfors/) had greatly increased.
On March 20-21, 2010, two dust events subsequently swept Qingdao. The on-line data in high
time-resolution can allow identifying two dust events accurately from the start to the end. The data
confirmed that the 4 hr dust samples with IDs of 20100320 and 20100321 were well separated from
each other for the two events, although they may not capture the entirety of the two events. The same
was true for the dust samples with IDs of 20110501, 20110502. Table 1 lists the sampling information.
Based on the forecast, we also collected aerosol particle samples immediately before, which were
regarded as the reference samples. These reference samples were further classified into sunny day
samples and cloudy day samples. For those events missing sampling prior to dust events, we collected
post-dust samples under clear and sunny weather conditions as early as possible.
Asian dust events were mostly observed in the spring at the sampling site. Our intensive samplings
were concentrated in the period of March to May in 2008-2011, when a smaller outbreak for Asian dust
events was observed in northern China (Fig. S3). Overall, a total of 14 sets of dust samples and 8 sets
of comparison samples were available for analysis in this study.
To facilitate the coastal sampling data analysis, sand samples were collected at the remote site of
Zhurihe (42°22'N, 112°58'E) in the Hunshandake Desert, one of the main Chinese sand deserts, in April
2012. Sand samples were packed in clean plastic sample bags and were stored below -20°C before the
transfer. An ice-box was used to store the samples during transport to the lab for chemical analysis.
**2.2 Analysis**
The aerosol samples were weighted according to the standard protocol. The sample membranes were
then cut into several portions for analysis. One portion of each aerosol sample was ultrasonically
extracted with ultra-pure water in an ice water bath for determining inorganic water-soluble ions using
ICS-3000 ion chromatography (Qi et al., 2011). The sand samples collected at the Zhurihe site were
analyzed using the same procedure.
One portion of each aerosol filter was cut into 60 cm$^2$ pieces and digested with $HNO_3+HClO_4+HF$
(5:2:2 by volume) at 160 ℃ using an electric heating plate. The concentrations of Cu, Zn, Cr, Sc and Pb
were measured using inductively coupled plasma mass spectrometry (Thermo X Series 2), while the
concentrations of Al, Ca, Fe, Na and Mg were measured using inductively coupled plasma atomic
emission spectroscopy (IRIS Intrepid II XSP). Field blank membranes were also analyzed for
correction.
One portion of aerosol sample was digested with an $HNO_3$ solution (10% $HNO_3$, 1.6 M) at 160 ℃ for
20 min in a microwave digestion system (CEM, U.S.). The Hg and As in sample extracts were analyzed
following the U.S. Environmental Protection Agency method 1631E (U.S. EPA, 2002) using cold vapor
atomic fluorescence spectrometry (CVAFS). The detection limits, precisions and recoveries of
water-soluble ions and metal elements are listed in Table 2.
**2.3 Computational modeling**
The enrichment factor of metal elements was given by
$$EF_i = \frac{\left(X_i/X_{Re}\right)_{aerosols}}{\left(X_i/X_{Re}\right)_{crust}} \qquad (1)$$
where subscripts $i$ and $Re$ refer to the studied metal and the reference metal, respectively; $(X_i/X_{Re})_{aerosols}$
is the concentration ratio of metal $i$ to metal $Re$ in the aerosol samples; and $(X_i/X_{Re})_{crust}$ is the ratio of
metal $i$ to metal $Re$ in the Earth's crust. For the calculation of the enrichment factor of the metal
elements, scandium was used as the reference element (Han et al., 2012), and the abundance of
elements in the Earth's crust given by Taylor (1964) was adopted.
The 72-h air mass back trajectories were calculated for each TSP sample using TrajStat software
(Wang et al., 2009) and National Oceanic and Atmospheric Administration (NOAA) GDAS (Global
Data Assimilation System) archive data (http:// www.arl.noaa.gov/ready/hysplit4.html). The air mass
back trajectories were calculated at an altitude of 1500 m to identify the dust origin. In addition, the
distance over sea of the air mass for each sample was measured from the trajectory using TrajStat
software (Wang et al., 2009).
The positive matrix factorization (PMF) is a commonly used receptor modeling method. This model
can quantify the contribution of sources to samples based on the composition or fingerprints of the
sources (Paatero and Tapper, 1993; Paatero, 1997). The measured composition data can be represented
by a matrix X of $i$ by $j$ dimensions, in which $i$ number of samples and $j$ chemical species were
measured, with uncertainty $u$. X can be factorized as a source profile matrix ($F$) with the number of
source factors ($p$) and a contribution matrix ($G$) of each source factor to each individual sample, as
shown in Equation 2.
$X_{ij} = \sum_{k=1}^{p} G_{ik} F_{kj} + E_{ij}$ (2)
where $E_{ij}$ is the residual for species $j$ of the i-th sample.
The aim of the model is to minimize the objective function $Q$, which was calculated from the
residual and uncertainty of all samples (Equation 3), to obtain the most optimal factor contributions and
profiles.
$Q = \sum_{i=1}^{n} \sum_{j=1}^{m} (E_{ij}/u_{ij})^2$ (3)
The EPA PMF 3.0 model was used to obtain the source apportionment of atmospheric particulates on
dust and comparison days. Our modeled results satisfied the reasonable fit criteria, i.e. 90% of the
scaled residuals were located between the range −3 and +3 for each species. The correlation coefficient
between the predicted and observed concentrations was 0.97.
Dry deposition velocities were obtained using Williams' model (Williams, 1982) by accounting for
particle growth (Qi et al., 2005). Williams' model is a two-layer model used to calculate the dry
velocity of size-segregated particles over the water. In an upper layer below a reference height (10 m),
the deposition of aerosol particles is governed by turbulent transfer and gravitational settling. In the
deposition layer, the gravitational settling of particles is affected by particle growth due to high relative
humidity. To obtain the deposition velocity of different particle sizes, Williams' model needs many
input parameters, such as the wind speed at 10-m height ($U_{10}$), air/water temperature, and relative
humidity. Relative humidity, air temperature and $U_{10}$ from the National Centers for Environmental
Prediction (NCEP) were used in this study. Surface seawater temperature data was collected from the
European Centre for Medium-Range Weather Forecasts (ECMWF). The meteorological and seawater
temperature data had a six-hour resolution. According to a previously reported method (Qi et al., 2013),
the dry deposition fluxes of the particles and the nitrogen species were calculated for dust and
comparison days.
The CMAQ model (v5.0.2) was applied over the East Asia area to simulate the concentrations of
$PM_{10}$, $NO_x$ and $NH_3$ for 14 samples collected during 11 dust events. The simulated domain contains
164×97 grid cells with a 36-km spatial resolution, and the centered point was 110 E, 34 N. The vertical
resolution includes 14 layers from the surface to the tropopause, with the first model layer at a height of
36 m above the ground level. The meteorological fields were generated by the Weather Research and
Forecasting (WRF) Model (v3.7). Considering that the simulated area is connected to the Yellow Sea,
the CB05Cl chemical mechanism was chosen to simulate the gas-phase chemistry. Zhang et al. (2009)
generated the emissions of air pollutants in 2006 including $NO_X$ and $NH_3$ over East Asia and they
updated the emission inventory in 2008 for us being used in this study. Initial conditions (ICONs) and
boundary conditions were generated from a global chemistry model of GEOS-CHEM. All the dust
events simulations are performed separately, each with a 1-week spin-up period to minimize the
influence of the ICONs. The validation of the application of the CMAQ model in China has been
reported by Liu et al. (2010a, b).
**2.4 Other data sources and statistical analysis**
Meteorological data were obtained from the Qingdao Meteorological Administration
(http://qdqx.qingdao.gov.cn/zdz/ystj.aspx) and the MICAPS of the Meteorological Administration of
China. Different weather characteristics, such as sunny days, cloudy days and dust days, were defined
according to information from the MICAPS and Qingdao Meteorological Administration. According to
the altitude, longitude and latitude of the 72-hr air mass back trajectory of each dust sample, the
pressure level, temperature and relative humidity (RH) data along the path of the air mass were derived
from the NCEP/NCAR re-analysis system
(http://www.esrl.noaa.gov/psd/data/gridded/data.ncep.reanalysis.html) for each sample. The mixed
layer depth during the air mass transport of dust samples was obtained from the HYSPLIT Trajectory
Model (http://ready.arl.noaa.gov/hypub-bin/trajasrc.pl) using the same method. Then the average
mixing layer, transport altitude, air temperature and RH were calculated as an average of all points on
the air mass back trajectory of each sample. Spearman correlation analysis was applied to examine the
relationships of nitrate and ammonium with transport parameters, and P values of <0.05 were
considered to be statistically significant.
**3 Results**

**3.1 Characterization of aerosol samples collected during dust events**

We first examined the mass concentrations of TSP samples and the concentrations of crustal and anthropogenic metals therein through a comparison with the samples collected on dust days and reference samples on immediately before or after days, providing the background information for our target species analyzed later. The comparative results are highlighted below. For these reference samples, the TSP mass concentrations ranged from 94 to 275 $\mu g \cdot m^{-3}$, with an average of 201 $\mu g \cdot m^{-3}$ (Fig. 2, Table S1). The TSP mass concentration increased substantially to 410-3857 $\mu g \cdot m^{-3}$ in dust day samples, with an average of 1140 $\mu g \cdot m^{-3}$. In each individual pair of dust day sample against reference sample, a net increase in the mass concentration of TSPs was observed. The percentages varied from 82 to 1,303% on basis of events, with a mean value of 403% (Table S1). A similar increase was present in the crustal elements in each pair of samples. For example, the mean concentrations of Sc, Al, Fe, Mg and nss-Ca (usually used as a typical dust index) increased by more than a factor of two. On the other hand, the enrichment factors (EF) of Al, Fe, Ca, and Mg were less than three in dust day samples with values less than 14 in the reference samples (Table 3). Lower values are indicative of elements from a primarily crustal origin. The average mass concentrations of anthropogenic elements, such as Cu, Pb, Zn, Cr, Hg and As, in dust day samples increased by 107% to 722% against those in the reference sample; however, the EF of the anthropogenic metal elements decreased in the former. This indicates that dust particles likely carried more anthropogenic elements, although their relative contribution to the total mass was lower than that in the reference sample. Note that Sample 20110415 was excluded for further analysis. It was judged as a local blowing dust event because no corresponding dust event existed upwind.

**3.2 Concentrations of $NH_4^+$ and $NO_3^-$ in dust day samples**

When the mass concentrations of $NH_4^+$ and $NO_3^-$ in each pair of TSP samples were compared, the concentrations of $NH_4^+$ increased by 8%-473% in some dust day samples (20080301, 20080315, 20090316, 20100315, 20100320, 20100321, 20110418 and 20110502), but decreased by 28-84% in other dust day samples (Fig. 3, Column $NH_4^+$ and $NO_3^-$ in Table S1). The same was generally true for the measured concentrations of $NO_3^-$.

Considering the relative values of $NH_4^+$ and $NO_3^-$ in dust day samples relative to the reference samples, we classified the dust day samples into three categories (Table 4). In Category 1, the mass

concentrations of $NH_4^+$ and $NO_3^-$ were larger in dust day samples against the reference samples. In
Category 2, the reverse was true. In Category 3, the mass concentrations of $NO_3^-$ were lower in the dust
samples than in the reference samples, whereas the concentrations of $NH_4^+$ were close to the reference.
As reported, the Yellow Sea encountered dust storms mainly derived from the Hunshandake Desert
(Zhang and Gao, 2007). We thereby compared our observations with the sand particles collected from
this desert (Table 5). The ratios of mass concentrations of nitrate and ammonium to the total mass of
sand particles were very low, i.e., less than 81 μg/g, which are approximately three orders of magnitude
less than the corresponding values in our dust samples. The values obtained from atmospheric aerosols
at the urban sites of Duolun (Cui, 2009) and Alxa Right Banner (Niu and Zhang, 2000), which are
closer to the desert, increased on dust days, but were still over one order of magnitude lower than the
corresponding values in this study (Table 5). The mixing and chemical interaction between
anthropogenic air pollutants and dust particles during transport from the source zone to the reception
site likely played an important role in increasing the ratios, leading to extremely larger ratio values at
this site relative to those in source dust and in upwind atmospheric particles (Cui et al., 2009; Wang et
al., 2011; Wu et al., 2016). Since air pollutant emissions, meteorological conditions, chemical reactions,
and others can affect the concentrations of $NH_4^+$ and $NO_3^-$ in atmospheric particles collected in dust
days, the observed increase or decrease in the mass concentration of nitrate and ammonium in different
dust samples against the reference implied the combined effect of those factors.
**4. Discussion**
**4.1 Theoretical analysis of the three categories**
Ammonium salts are common in atmospheric particles with diameters of less than 2 μm (Yao et al.,
2003; Yao and Zhang, 2012). Many modeling studies have shown that the gas-aerosol thermodynamic
equilibrium is assumed to be fully attained for inorganic ions, including ammonium salts in $PM_{2.5}$
(Dentener et al., 1996; Underwood et al., 2001; Wang et al., 2017a; Zhang et al., 1994; Zhang and
Carmichael, 1999). Reasonably good agreements between ammonium salt modeling results and
observations reported in the literature support the validity of this assumption (Chen et al., 2016;
Penrodet al., 2014; Walker et al., 2012). Supposing that a thermodynamic equilibrium had been attained
by the ammonium salts in Category 1, the reactions between carbonate salts and ammonium salts, such
as 1) $(NH_4)_2SO_4 + CaCO_3 \Rightarrow CaSO_4 + NH_3$ (gas) $+ CO_2$ (gas) $+ H_2O$ and 2) $2NH_4NO_3 +$
$CaCO_3 \Rightarrow Ca(NO_3)_2 + 2NH_3$ (gas) $+CO_2$ (gas) $+H_2O$, will release $NH_3$ (gas) until $CaCO_3$ has been
completely used up. During dust events, very high concentrations of $Ca^{2+}$ were observed, and high
$CaCO_3$ concentrations were therefore expected. For example, the single-particle characterization
showed that Asia dust from the Gobi and Inner Mongolian Deserts had rich $CaCO_3$, with a ratio of
4.3-6.7% for reacted $CaCO_3$ and 3.0-4.6% for unreacted $CaCO_3$ (Hwang et al., 2008).
Heterogeneous chemical reactions of mineral dust mostly occurred on $CaCO_3$ mineral dust (Hwang and
Ro, 2006). However, when Category 1 was considered alone except for Sample 20100321, a good
correlation was obtained for $[NH_4^+]_{equivalent\ concentration}=0.98*[NO_3^-+SO_4^{2-}]_{equivalent\ concentration}$ ($R^2$=0.83,
P<0.05). The good correlation, together with the slope of 1, strongly indicated that the $NO_3^-$ and $SO_4^{2-}$
were almost completely associated with $NH_4^+$ in these dust day samples. Anthropogenic ammonium
nitrate and ammonium sulfate were thought to be produced by gas, aqueous phase reaction and
thermodynamic equilibrium processes and they usually internally mixed (Seinfeld and Pandis, 1998).
In reverse, the poor correlation of $Ca^{2+}$ to $NO_3^-$ and $SO_4^{2-}$ showed that the formation of $CaSO_4$ and/or
$Ca(NO_3)_2$ was probably negligible. Thus, ammonium salt aerosols very likely existed separately with
dust aerosols in these dust day samples. Wang et al. (2017a) also found that coarse mode ammonium
was quite low and fine mode dust particles existed separately with anthropogenic ammonium nitrate
and ammonium sulfate. The observed $NO_3^-$ and $NH_4^+$ in Asia dust samples were argued due to
physically mixing two types of particles rather than the heterogeneous formation of nitrate and
ammonium (Huang et al., 2010). The hypothesis appeared to be valid in Category 1, where $NH_4^+$ was
negatively correlated with $Ca^{2+}$ (Fig. S4). In the Sample 20100321 collected on 21 March 2010, $[NH_4^+]$
only accounted for ~70% of the observed $[NO_3^-+SO_4^{2-}]$ in an equivalent concentration. This result
suggested that ~30% of $(NO_3^-+SO_4^{2-})$ may be associated with dust aerosols via the formation of metal
salts of the two species. This hypothesis was supported by the correlation result, i.e., $NO_3^-$ was
positively correlated with $NH_4^+$ and Cu, and $SO_4^{2-}$ was correlated with $K^+$, $Na^+$ and $Mg^{2+}$ (Fig. S4).
Scheinhardt et al. (2013) found that $Cu^{2+}$ showed mixed organic and nitrate complexation in aerosol
particles, using a thermodynamic model (E-AIM III). Cu was also detected to be partly in the form of
nitrate in aerosol particles by single particle mass spectrometry (Wang et al., 2016a; Zhang et al., 2015).
Cu was once used as an effective marker of diesel and biodiesel-blend exhaust (Gangwar et al., 2012),
while it can also be derived from copper pyrites ($CuFeS_2$) in Inner Mongolia mines (Huang et al., 2010).
The increase of Cu in the mass concentration in dust samples implied dust particles mixed with
anthropogenic particles, particularly from industrial emissions, during transport. In addition, many
studies showed that $SO_4^{2-}$ can exist in many forms of metal salts in atmospheric particles, such as
$Na_2SO_4$, $K_2SO_4$, $K_2Ca(SO_4)_2 \cdot H_2O$, $Na_2Ca(SO_4)_2$, $Na_2Mg(SO_4)_2 \cdot 4H_2O$, $(NH_4)_2Mg(SO_4)_2 \cdot 6H_2O$,
$Na_3(NO_3)(SO_4) \cdot H_2O$ (Chabas and Lefèvre, 2000; Sobanska et al., 2012; Xie et al., 2005).
For Category 2, no correlation between $[NH_4^+]_{equivalent\ concentration}$ and $[NO_3^- + SO_4^{2-}]_{equivalent\ concentration}$
existed. When Category 2 was considered alone except for one Sample 20110501, the equivalent ratios
of $NH_4^+$ to $NO_3^- + SO_4^{2-}$ were generally much smaller than 1, suggesting that a larger fraction of
$NO_3^- + SO_4^{2-}$ may exist as metal salts due to reactions of their precursors with dust aerosols. $NO_3^-$ and
$SO_4^{2-}$ showed no correlations with $NH_4^+$ but did show significant correlations with Pb (Fig. S4). The
average concentration of $Ca^{2+}$ in Category 2 ($0.43\pm0.40$ $\mu mol/m^3$) was evidently higher than that in
Category 1 ($Ca^{2+}$: $0.17\pm0.04$ $\mu mol/m^3$), implying the probable formation of $CaSO_4$ and/or $Ca(NO_3)_2$
and the release of $NH_3$ (gas). Moreover, except for 20080502, the remaining dust samples in Category
2 were transported from the desert relatively enriched with $CaCO_3$ (1-25% in Wt%) (Formenti et al.,
2011). A positive correlation between $NO_3^-$ and $SO_4^{2-}$ in Category 2 against a negative correlation in
Category 1 also implied that the dust particles enriched with $CaCO_3$ in Category 2 might play an
important role to form $SO_4^{2-}$ and $NO_3^-$. Ca-rich dust particles coated with highly soluble nitrate were
observed at Kanazawa in Japan during Asian dust storm periods using SEM/EDX (scanning electron
microscopy equipped with an energy dispersive X-ray spectrometer) (Tobo et al.,2010). The
single-particle observation conducted by Hwang and Ro (2006) showed that $CaCO_3$ in dust particles
was almost completely consumed to produce mainly $Ca(NO_3)_2$ species.
There were only three samples in Category 3. $[NH_4^+]_{equivalent\ concentration} = 0.95*[NO_3^- + SO_4^{2-} + Cl^-]_{equivalent}$
$_{concentration}$ was obtained for Sample 20110418, implying that the $NH_4^+$ was not only associated with
$NO_3^-$ and $SO_4^{2-}$ but also with $Cl^-$. In the sample collected on 15 March 2010, $[NH_4^+]$ accounted for 78%
of the observed $[NO_3^- + SO_4^{2-}]$ in an equivalent concentration. As discussed above, ~20% of
$(NO_3^- + SO_4^{2-})$ may be associated with dust aerosols via the formation of metal salts of the two species.
The equivalent ratio of $NH_4^+$ to $NO_3^- + SO_4^{2-}$ was only 0.14 for Sample 20100320, and $Ca^{2+}$ for this
sample (0.47 $\mu mol/m^3$) was evidently higher than that for Sample 20100315 ($Ca^{2+}$: 0.12 $\mu mol/m^3$) and
20110418 ($Ca^{2+}$: 0.12 $\mu mol/m^3$), suggesting that a larger fraction of $NO_3^- + SO_4^{2-}$ may exist as metal
salts. However, the unique changes in $NH_4^+$ and $NO_3^-$, different from Category 1 and 2, need further
investigation.
4.2 Source apportionment of aerosols during dust and non-dust events
The sources of atmospheric aerosols in dust and reference samples were determined by PMF
modeling (Paatero and Tapper, 1993; Paatero, 1997). Fig. 4 shows that atmospheric aerosols in the
reference samples mainly included six sources, i.e., industry, soil dust, secondary aerosols, sea salt,
biomass burning, and coal combustion/other sources. In these dust samples, including Categories 1-3,
oil combustion, industry, soil dust, secondary aerosols, and coal combustion/other sources were
identified as five major sources (Table 6). The contribution of soil dust evidently increased from 23%
to 36% in the dust samples relative to the reference, consistent with the high concentrations of TSPs
and crustal metals observed on dust days. The calculated contribution of nitrate plus ammonium from
the soil dust source to the total mass of nitrate plus ammonium in the dust samples greatly increased.
The source profile for coal combustion in the dust day samples showed a high percentage of $K^+$, $Cl^-$, Ca,
Mg, Co, Ni, As, Al and Fe, indicating that coal combustion particles may exist contemporaneously
with other anthropogenic pollutants emitted along the transport path. Liu et al. (2014) also found a
larger net increase in the contribution of dust aerosols to the mass of $PM_{10}$, i.e., 31%-40%, on dust days
against non-dust days in Beijing which is approximately 600 km upwind of Qingdao. Accordingly, they
reported that the contributions of local anthropogenic sources decreased on dust days, especially those
from secondary aerosols, consistent with the EF of anthropogenic metals observed on dust days.
4.3 Influence of transport path ways on $NH_4^+$ and $NO_3^-$ in dust samples
The calculated air mass trajectories for 13 out of 14 samples showed that the air mass originated
from North and Inner Mongolia, China (Fig. 5), generally consistent with the results of Zhang and Gao
(2007). The remaining one, with ID of 20110418 originated from Northeast China. The calculated
trajectories showed that the entire dust air mass passed over those highly polluted regions with strong
modeled emissions of $NO_x$ and $NH_3$ shown in Fig 6 and experienced different residence times therein.
Fig. 5 shows that all air mass trajectories in Category 1 were transported from either the north or
northwest over the continent, except for the Sample 20110502. In Category 2, the air masses always
took a 94-255 km trip over the sea prior to arriving at the reception site. $NH_3$-poor conditions in the
marine atmosphere disfavored the formation and existence of ammonium nitrate. On the other hand,

the humid marine conditions (the calculated average RH ranged in 50-75% over the Bohai and Yellow Seas in 2006-2012 using NCEP/NCAR re-analysis data) might have enhanced hetero-coagulation between dust and smaller anthropogenic particles, leading to the release of $NH_3$ via reactions between preexisting ammonium salts and carbonate salts.

The average mixing layer was less than 900 m along the air mass transport routes for most sampling days in Category 1 (Table 7), favoring the trapping of locally emitted anthropogenic air pollutants in the mixing layer. The air masses in Category 1 took over 11-39 hrs to cross over the highly polluted area with appreciable modeled concentrations of $NO_x$ (5.7±1.4 ppb) and $NH_3$ (7.6±3.3 ppb). Except for two samples (ID of 20080529 and 20110319), air masses in Category 2 took less than 10 hrs to cross over the polluted areas with lower concentrations of $NO_x$ (modeled value: 3.6±3.4 ppb) and $NH_3$ (modeled value: 4.7±4.7 ppb) and the mixing layer height along the route was 916-1194 m (on average) for each dust event. Moreover, the averaged wind speed at sampling site was 2.8 m/s in Category 1, but 6.2 m/s in Category 2. The lower wind speed in Category 1 was unexpected, implying dust particles very likely traveled at aloft with a high speed and then mixed down to the ground through subsidence. This further led to the external mixing of anthropogenic particulate matters and dust. The correlation analysis results in Table S2 indirectly support these conclusions.

The concentrations of $PM_{10}$ and its major components $NO_3^-$ and $NH_4^+$ over East Asia on dust days and comparison days were modeled using the WRF-CMAQ model (Fig. S5-6). Spatial distributions of simulated $PM_{10}$ during each dust events were consistent with the records in the "Sand-dust Weather Almanac" (CMA, 2009; 2010; 2012; 2013). The dust particles were transported eastward by passing over the sampling site, the China Sea and arriving at the far remote ocean region, except for the local blowing dust sample with ID of 20110415, as mentioned previously. NMB (normalized mean bias) values of simulated $NO_3^-$ were -4% and -12% in dust and non-dust reference samples, respectively, indicating that CMAQ results reasonably reproduce the mass concentrations of $NO_3^-$ (Fig. S6). Simulated $NH_4^+$ concentrations in dust samples were severely under-predicted with NMB values at -71%. For reference samples, simulated $NH_4^+$ concentrations sometimes can well reproduce the observational values, but the simulation was sometimes severely deviated from the observation. The deviation could be related to many factors which were out of scope of this study. The separately mixing mechanism proposed in this study is urgently needed to be included in the model for accurately predicting the concentrations during dust events.

**4.4 Dry deposition fluxes of TSP, $NH_4^+$, $NO_3^-$ and metals**

Dust events are known to increase the deposition fluxes of aerosol particles along the transport path because of high particle loadings. For example, Fu et al. (2014) found that the long-range transported dust particles increased the dry deposition of $PM_{10}$ in the Yangtze River Delta region by a factor of approximately 20. In terms of atmospheric deposition in the oceans, a few studies reported enhancements in oceanic chlorophyll *a* following dust storm events (Banerjee and Kumar, 2014; Tan and Wang, 2014). In addition to those in high-nutrient and low-chlorophyll (HNLC) regions, the input of nitrogen and other nutrients associated with dust deposition is expected to promote the growth of phytoplankton in oceans with varying nutrient limitation conditions. Thus, we calculated the dry deposition fluxes of aerosols particles, $N_{NH4++NO3-}$ and metal elements during dust and reference periods using the measured component concentrations and modeled dry deposition velocities (Table 8). We also compared the calculated dry deposition flux of TSP and $N_{NH4++NO3-}$ with previous observations in the literature.

The calculated dry deposition fluxes of atmospheric particulates increased on dust days against the reference to some extent. For example, the particle deposition fluxes varied over a wide range from 5,200 to 65,000 mg/m$^2$/month in different dust sampling days, with an average of 18,453 mg/m$^2$/month, in comparison with the dry deposition flux of TSP of 2,800±700 mg/m$^2$/month from the reference periods in the coastal region of the Yellow Sea. The dry deposition fluxes of $N_{NH4++NO3-}$ varied, depending on Category 1, 2 or 3. In Category 1, the dry deposition fluxes of $N_{NH4++NO3-}$ increased by 9-75% with increased TSP flux by 86-252% (Table S3). In Categories 2 and 3, the dry deposition fluxes of TSP increased by 126% to 2,226% against the references. The dry deposition fluxes of particulate $N_{NH4++NO3-}$ decreased by 50%, on average, in Categories 2 and 3, although the fluxes of ammonium of two samples in Category 3 increased. A larger decrease against the reference in the flux of nitrate was present in Categories 2 and 3, i.e., decreases of 73% and 46%, respectively. The ammonium deposition flux also decreased by 47% in Category 2 but increased by 10% in Category 3.

Except for Pb and Zn in Category 2, the calculated dry deposition fluxes of Cu, Pb and Zn increased with those of nitrogen on dust days. Trace metals were found to have a toxic effect on marine phytoplankton and inhibit their growth (Bielmyer et al., 2006; Echeveste et al., 2012). Liu et al. (2013) found that inhibition coexisted with the promotion of phytoplankton species in incubation experiments

in the southern Yellow Sea in the spring of 2011 by adding Asian dust samples to collected seawater.
However, the calculated dry atmospheric deposition fluxes of Fe increased by a factor of 124-2,370%
in dust day samples. Wang et al. (2017b) recently reported that Fe can alleviate the toxicity of heavy
metals. Moreover, atmospheric inputs of iron to the ocean have been widely proposed to enhance
primary production in HNLC areas (Jickells et al., 2005).
Due to anthropogenic activity and economic development, $NO_x$ and $NH_3$ emissions were reported to
increase in China from 1980 to 2010 (Fig. S3; Liu et al., 2017). The dry deposition flux of $N_{NH4^++NO3^-}$
should have theoretically increased with the increase in the emission of inorganic nitrogen. Considering
the different dry deposition velocities to be used in various studies, we recalculated the dry deposition
flux of $N_{NH4^++NO3^-}$ in the literature using the dry deposition velocities of 1 cm/s for nitrate and 0.1 m/s
for ammonium, as reported by Duce et al. (1991). We thereby found that dry deposition fluxes of
$N_{NH4^++NO3^-}$ over the Yellow Sea during the dust days increased greatly from 1999 to 2007, but the
values in Qingdao varied narrowly within a range of 94.75-99.65 mg $N/m^2$/month during the dust days
from 1997 to 2011 (Table 9). The complicated results implied that even more updated works are needed
in the future.
**5 Conclusion**
The concentrations of nitrate and ammonium in TSP samples varied greatly from event to event on
dust days. Relative to the reference samples, the concentrations were both higher in some cases and
lower in others. The observed ammonium in dust day samples was explained by $NH_4^+$ was likely either
in the form of ammonium salts existing separately with dust aerosols or as the residual of incomplete
reactions between ammonium salts and carbonate salts. $NO_3^-$ in the dust day samples can be due to
either mixing or reactions between anthropogenic air pollutants and dust particles or combined both
during the transport from the source zone to the reception site. However, this process was generally
much less effective and led to a sharp decrease in nitrate in TSP samples of Category 2. The existence
of ammonium salt aerosols separately with dust aerosols and the extent of the reactions between
ammonium salts and carbonate salts were apparently associated with the transport pathway,
metrological conditions and precursor emissions, and other factors. Due to a sharp increase in dust
loads on dust days, the contribution of dust to the total aerosol mass increased against the samples
collected on other days. The contributions from local anthropogenic sources were accordingly lower on
dust days.
Overall, this study strongly suggested that atmospheric deposition of $N_{NH4+ +NO3-}$ on dust days varied
greatly. A simple assumption of a linear increase in $N_{NH4+ +NO3-}$ with increasing dust load, like that in the
literature, could lead to a considerable overestimation of the dry deposition flux of nutrients into the
oceans and the consequent primary production associated with dust events.
*Acknowledgments.* This work was supported by the Department of Science and Technology of the P. R.
China through the State Key Basic Research & Development Program under Grant No. 2014CB953701
and the National Natural Science Foundation of China (No. 41375143). We thank Prof. Yaqiang Wang
and Jinhui Shi for the valuable discussion regarding this research. We also express our appreciation to
Tianran Zhang for help with sand sampling, and Qiang Zhang, Yang Yu and Jiuren Lin for data
collection.

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

**Table 1**. Sampling information for the aerosol samples collected at the Baguanshan site in the coastal
region of the Yellow Sea.

| Sampling year | Sample category | Sampling number | Sampling time | Weather characteristics |
|---|---|---|---|---|
| 2008 | Samples on dust days | 20080301 | From 13:22 a.m. to 17:22 p.m. on Mar. 1st | Floating dust[a] |
| | | 20080315 | From 13:21 a.m. to 17:21 p.m. on Mar. 15th | Floating dust |
| | | 20080425 | From 13:14 a.m. to 17:14 p.m. on Apr. 25th | Floating dust |
| | | 20080528 | From 11:38 a.m. to 15:38 p.m. on May 28th | Floating dust |
| | | 20080529 | From 10:15 a.m. to 12:15 p. m. on May 29th[b] | Floating dust |
| | Reference Samples | 20080316 | From 13:00 a.m. to 17:00 p.m. on Mar. 16th | Sunny day |
| | | 20080424 | From 13:00 a.m. to 17:00 p.m. on Apr. 24th | Sunny day |
| | | 20080522 | From 13:00 a.m. to 17:00 p.m. on May 22nd | Cloudy day with mist |
| 2009 | Samples on dust days | 20090316 | From 8:25 a.m. to 12:25 p.m. on Mar. 16th | Floating dust |
| | Reference Samples | 20090306 | From 13:00 a.m.to 17:00 p.m. on Mar. 6th | Sunny day |
| 2010 | Samples on dust days | 20100315 | From 11:30 a.m.to 15:30 p.m. on Mar. 16th | Mist after floating dust |
| | | 20100320 | From 10:30 a.m. to 14:30 p.m. on Mar. 20th | Floating dust |
| | | 20100321 | From 10:30 a.m. to 14:30 p.m. on Mar. 21st | Floating dust |
| | Reference Samples | 20100324 | From 11:30 a.m. to 15:30 p.m. on Mar. 24th | Sunny day |
| 2011 | Samples on dust days | 20110319 | From 12:00 a.m. to 16:00 p.m. on Mar. 19th | Floating dust |
| | | 20110415 | From 12:00 a.m. to 16:00 p.m. on Apr. 15th | Floating dust |
| | | 20110418 | From 12:25 a.m. to 16:25 p.m. on Apr. 18th | Floating dust[c] |
| | | 20110501 | From 12:10 a.m. to 16:10 p.m. on May 1st | Floating dust |
| | | 20110502 | From 16:00 a.m. to 20:00 p.m. on May 2nd | Floating dust |
| | Reference Samples | 20110308 | From 12:00 a.m. to 16:00 p.m. on Mar. 8th | Sunny day |

| | | |
|---|---|---|
| 20110416 | From 12:00 a.m. to 16:00 p.m. on Apr. 16th | Sunny day |
| 20110523 | From 12:00 a.m. to 16:00 p.m. on May 23rd | Sunny day |

[a]Note that one dust sample 20080301 was collected on March 1 when no dust was recorded by the MICAPS. However, the MICAPS information indeed showed dust events in China on March 1. The modeled spatial distribution of the $PM_{10}$ mass concentration for this dust event on March 1 implies that the sample should be classified as a dust sample. The supporting figures are shown in Fig. S1.

[b]The sampling duration was reduced to only 2 hrs because of extremely high particle loads. In addition, the samples with IDs of 20080528 and 20080529 were subjected to two different dust events occurring over two days instead of continuous samples for one dust event (CMA, 2009).

[c]Note that one dust sample 20110418 was collected on April 18 when no dust was recorded by the MICAPS. However, blowing dust occurred and was recorded on April 17 by the Sand-dust Weather Almanac 2011 (CMA, 2013). The modeled spatial distribution of the $PM_{10}$ mass concentration for this dust event on April 18 implies that the sample should be classified as a dust sample. The supporting figure is Fig. S2.

**Table 2.** Detection limits, precisions and recoveries of water-soluble ions and metal elements.

| Component | Measurement method | Detection limit ($\mu g\ L^{-1}$) | Precision (RSD%) | Recovery (%) |
|---|---|---|---|---|
| $NO_3^-$ | | 2.72 | 1.54 | 97 |
| $SO_4^{2-}$ | IC | 1.62 | 1.55 | 98 |
| $NH_4^+$ | | 0.4 | 1.10 | 97 |
| $Ca^{2+}$ | | 0.44 | 0.79 | 94 |
| Cu | ICP-MS (Xin et | 0.006 | 4.0 | 106 |
| Zn | al., 2012) | 0.009 | 2.5 | 102 |
| Cr | | 0.004 | 3.0 | 95 |
| Sc | | 0.002 | 2.4 | 97 |
| Pb | | 0.008 | 3.9 | 104 |
| Al | ICP-AES (Lin et | 7.9 | 0.6 | 103 |
| Ca | al., 1998) | 5.0 | 1.2 | 99 |
| Fe | | 2.6 | 0.7 | 104 |
| Na | | 3.0 | 0.6 | 99 |
| Mg | | 0.6 | 0.6 | 105 |
| Hg | CVAFS | 0.0001 | 6.6 | 105 |
| As | CVAFS | 0.1 | 5.0 | 98 |


**Table 3.** The average concentrations and EFs of metal elements on dust and non-dust days.

| Element | Concentration (ng/m$^3$) | | EF* | |
|---------|--------------|-----------|----------------|-----------|
|         | Reference days | Dust days | Reference days | Dust days |
| Sc | 1.11 | 13.90 | - | - |
| Al | $8.53 \times 10^3$ | $6.86 \times 10^4$ | 3.8 | 1.4 |
| Fe | $4.91 \times 10^3$ | $3.88 \times 10^4$ | 3. | 1.2 |
| Ca | $1.05 \times 10^4$ | $4.29 \times 10^4$ | 14.0 | 2.1 |
| Mg | $1.62 \times 10^3$ | $1.58 \times 10^4$ | 3.5 | 1.1 |
| Cu | 50.2 | 124.5 | 36.3 | 6.1 |
| Pb | 127.9 | 221.0 | 389.4 | 56.1 |
| Zn | 340.0 | 457.7 | 248.9 | 20.6 |
| Cr | 33.8 | 244.0 | 44.0 | 11.1 |
| Hg | 0.26 | 0.36 | 176.0 | 13.8 |
| As | 25.5 | 27.4 | 707.2 | 43.9 |

*EF values less than 10 indicate that the studied element is mainly derived from crustal sources,
whereas EF values much higher than 10 indicate an anthropogenic source.

**Table 4.** Average measured concentrations of $NH_4^+$, $NO_3^-$, TSP, NOx, relative humidity (RH) and air temperature for each aerosol sample category in Qingdao.

| | Sample number | TSP (μg·m$^{-3}$) | $NO_3^-$ (μg·m$^{-3}$) | $NH_4^+$ (μg·m$^{-3}$) | RH (%) | T (℃) | NOx (μg·m$^{-3}$) | Summary |
|---|---|---|---|---|---|---|---|---|
| Category 1 | 20080301 | 527 | 20.5 | 12.7 | 57 | 7.0 | 36 | $NH_4^+$ and $NO_3^-$ concentration in dust day samples higher than reference samples |
| | 20080315 | 410 | 19. 5 | 29.9 | 62 | 11.0 | 59 | |
| | 20090316 | 688 | 15.9 | 17.2 | 27 | 16.0 | 75 | |
| | 20100321 | 519 | 16.5 | 9.4 | 51 | 8.8 | 76 | |
| | 20110502 | 810 | 21.0 | 11.0 | 49 | 17.7 | 62 | |
| Category 2 | 20080425 | 622 | 6.8 | 2.0 | 30 | 18.0 | 40 | $NH_4^+$ and $NO_3^-$ concentration in dust day samples lower than reference samples |
| | 20080528 | 2579 | 9.2 | 2.7 | 17 | 27.0 | 34 | |
| | 20080529 | 2314 | 17.5 | 4.8 | 60 | 20.0 | 29 | |
| | 20110319 | 939 | 12.3 | 9.4 | 16 | 12.6 | 93 | |
| | 20110501 | 502 | 4.5 | 5.3 | 23 | 21.6 | 66 | |
| Category 3 | 20100315 | 501 | 5.4 | 4.3 | 30 | 7.2 | 73 | $NO_3^-$ concentration in dust day samples lower than reference samples; $NH_4^+$ close to that on reference samples |
| | 20100320 | 3857 | 5.5 | 3.4 | 35 | 10.6 | 92 | |
| | 20110418 | 558 | 3.8 | 6.6 | 33 | 12.6 | 47 | |
| Reference samples[a] | 20080316 | 225 | 12.6 | 8.4 | 28 | 11.0 | 60 | |
| | 20080424 | 137 | 21.7 | 7.2 | 49 | 18.0 | 53 | |
| | 20080522 | 206 | 27.4 | 16.6 | 78 | 20.0 | 60 | |
| | 20090306 | 94 | 2.9 | 3.0 | 29 | 7.00 | 51 | |
| | 20100324 | 275 | 7.2 | 2.4 | 23 | 9.0 | 82 | |
| | 20110308 | 194 | 13.0 | 13.1 | 20 | 11.5 | 111 | |
| | 20110416 | 252 | 5.6 | 5.4 | 26 | 14.1 | 55 | |
| | 20110523 | 224 | 15.2 | 10.2 | 42 | 20.6 | 49 | |

[a]For the corresponding reference sample for each dust event, see Table 1.

**Table 5.** Comparison of the $NH_4^+$ and $NO_3^-$ content in sand and aerosol particles on dust days or close
to the dust source region (unit: μg/g).

| Sands sampled in dust source regions | | | Aerosols in or close to dust source region on dust days | | | Aerosols in the coastal region of the Yellow Sea | |
|---|---|---|---|---|---|---|---|
| Study region and data source | Relative concentration[a] | | Study region and data source | Relative concentration[a] | | | |
| | $NO_3^-$ | $NH_4^+$ | | $NO_3^-$ | $NH_4^+$ | $NO_3^-$ | $NH_4^+$ |
| Zhurihe (This study) | 25.46± 22.87 | 4.21± 1.03 | Duolun (Cui, 2009) | 1200 | 900 | Reference samples: 28,200±24,819 | Reference samples: 24,063±21,515 |
| Alxa Left Banner, Inner Mongolia (Niu and Zhang, 2000) | 62.1±7.4 | 79.1±1.1 | Alxa Right Banner, Inner Mongolia (Niu and Zhang, 2000) | 1975[b] | 4091[b] | Category 1: 34,892±9570 | Category 1: 22,571±7,016 |
| Yanchi, Ningxia (Niu and Zhang, 2000) | 46.4±2.2 | 80.9±1.3 | Hinterland of the Taklimakan Desert, Xinjiang (Dai et al., 2016) | 142-233 | 2-15 | Category 2: 5,542±5,117 | Category 2: 4,758±5,698 |
| | | | Average of Sonid Youqi, Huade (Inner Mongolia), Zhangbei (Hebei) (Mori et al., 2003) | 253 | 710 | Category 3: 6,359±4,697 | Category 3: 7,059±5,591 |
| | | | Yulin, the north edge of Loess Plateau (Wang et al., 2011) | 216.4 | 80.6 | | |
| | | | Golmud, Qinghai(Sheng et al., 2016) | 892.9 | -[c] | | |
| | | | Hohhot, Inner Mongolia (Yang et al., 1995) | 588.1 | No data | | |

[a]Relative concentration of $NH_4^+$ and $NO_3^-$ per aerosol particle mass
[b]Samples collected on a floating dust day (horizontal visibility less than 10000 m and very low wind
speed)
[c]The ammonium concentration was lower than the detection limit of the analytical instrument.

**Table 6.** Sources and source contributions (expressed in %) calculated for aerosol samples collected during dust and non-dust events

| Dust event | | Comparison days | |
|---|---|---|---|
| Source | % of TSP | Source | % of TSP |
| Soil dust | 36 | Soil dust | 23 |
| Industrial | 21 | Industrial | 24 |
| Secondary aerosol | 6 | Secondary aerosol | 23 |
| Oil combustion | 6 | Biomass burning | 16 |
| Coal combustion and other uncertain sources | 31 | Coal combustion | 5 |
| | | Sea salt | 9 |

Table 7. Concentrations of TSP, $NO_3^-$, and $NH_4^+$; transport speed; transport distance over the sea; transport distance; air temperature; RH; average mixed layer during transport and transport time in polluted region for atmospheric aerosol samples on dust days.

| Group | Sample number | TSP (µg/m³) | $NO_3^-$ (µg/g) | $NH_4^+$ (µg/g) | Speed (km/h) | Distance over the sea (km) | Transport altitude (m) | Mixed layer depth (m) | R-time[a] (h) | $T^b$ (℃) | $RH^c$ (%) |
|---|---|---|---|---|---|---|---|---|---|---|---|
| | 080301 | 527 | 38,984 | 24,107 | 40.1 | 0 | 1,160±702 | 864±745 | 39 | -2.9±11.7 | 29±10 |
| Category 1 | 080315 | 410 | 47,611 | 34,130 | 79.1 | 0 | 4,921±1,870 | 950±525 | 13 | -32.5± 16.4 | 34±16 |
| $NH_4^+>RS^d$ | 090316 | 688 | 23,050 | 25,012 | 86.2 | 0 | 3,739±1083 | 702±665 | 11 | -19.1±11.7 | 42±17 |
| $NO_3^->RS^d$ | 100321 | 519 | 31,741 | 18,155 | 87.2 | 0 | 3,407±1,249 | 1,113±760 | 19 | -23.0±13.6 | 42±22 |
| | 110502 | 810 | 25,995 | 13,632 | 30.2 | 177 | 3,666±1,371 | 747±957 | 26 | -13.2±15.8 | 31±13 |
| | 080425 | 256 | 4,089 | 372 | 29.6 | 0 | 887±656 | 1,161±1,040 | 10 | -2.7±6.1 | 66±13 |
| Category 2 | 080528 | 2579 | 232 | 72 | 88.2 | 244 | 4,336±1461 | 1,064±830 | 8 | -15.5±13.6 | 31±16 |
| $NH_4^+<RS^d$ | 080529 | 2314 | 26 | 166 | 63.7 | 94 | 2,148±1,725 | 1,194±816 | 43 | 3.6±18.4 | 25±17 |
| $NO_3^-<RS^d$ | 110319 | 939 | 13,088 | 10,067 | 70.6 | 132 | 4,271±1867 | 790±719 | 27 | -26.3±20.0 | 48±32 |
| | 110501 | 502 | 8,924 | 10,631 | 35.1 | 252 | 3,212±810 | 916±1,114 | 5 | -13.4±8.5 | 39±13 |
| Category 3 | 100315 | 501 | 10,767 | 8,515 | 57.3 | 0 | 5,009±1410 | 1,110±365 | 7 | -40.4±13.3 | 45±29 |
| $NO_3^-<RS^d$ | 100320 | 3857 | 1,418 | 884 | 76.9 | 0 | 1,284±401 | 525±371 | 10 | -12.2±6.3 | 61±16 |
| $NH_4^+\cong RS^d$ | 110418 | 558 | 6,891 | 11,778 | 35.6 | 931 | 1,344±780 | 695±672 | 2 | -0.1±8.2 | 52±28 |

[a]Residence time of the air mass passing over parts of highly polluted regions according to the trajectories of samples.

[b]Average air temperature with the definition in Section 2.4.

[c]Average relative humidity with the definition in Section 2.4.

[d]Reference samples collected on days immediately before or after dust event

Table 8. Dry deposition of TSP (mg/m$^2$/month), $N_{NH4++NO3-}$ (mg N/m$^2$/month) and some toxic trace metals (mg/m$^2$/month) on dust and reference days.

| | | | | Dry deposition flux | | | | |
|---|---|---|---|---|---|---|---|---|
| | TSP | NO$_3^-$ -N | NH$_4^+$-N | $N_{NH4++NO3-}$ | Fe | Cu | Pb | Zn |
| Category 1[a] | 8,000± 1800 | 65±9 | 24±14 | 90±17 | 533±179 | 2±0.3 | 0.3±0.3 | 6±2 |
| Category 2[a] | 18000± 11,000 | 13±18 | 8±4 | 21±22 | 1300±1000 | 3±2 | 0.08±0.04 | 4±1 |
| Category 3[a] | 29,000± 31,000 | 26±6 | 17±8 | 42±12 | 2100±2200 | 6±1 | 0.20±0.02 | 5±3 |
| Reference samples | 2,800± 700 | 48±33 | 15±8 | 63±39 | 190±110 | 1±1 | 0.09±0.1 | 5±4 |

[a]For the characterization of $N_{NH4++NO3-}$ concentration and sample information of the category, see Table 3.

**Table 9.** Comparison of dry deposition flux and normalized flux of TSP (mg/m$^2$/month) and $N_{NH4++NO3-}$
(mg N/m$^2$/month) with observations from other studies (mg N/m$^2$/month)

| Source | Year | Area | | TSP | $N_{NH4++NO3-}$ | Normalized average flux of $N_{NH4++NO3-}$[a] |
|---|---|---|---|---|---|---|
| This work | 2008-2011 | Qingdao, coastal region of the Yellow Sea | Reference day | 2,800±700 | 63±39 | 93.90 |
| | | | Dust day | 10,138±15,940 | 58±36 | 101.39 |
| | | | Average of dust and reference | | | 97.64 |
| Qi et al., 2013 | 2005-2006 | Qingdao, coastal region of the Yellow Sea | Average of nine months samples | 159.2 - 3,172.9 | 1.8-24.5 | 94.75 |
| Zhang et al., 2011 | 1997-2005 | Qingdao | Average of annual samples | | 132 | 99.65 |
| Zhang et al., 2007 | 1999-2003 | The Yellow Sea | | | 11.43 | 9.91 |
| Shi et al., 2013 | 2007 | The Yellow Sea | Reference day | | 19.2 | 132.17 |
| | | | Dust day | | 104.4 | 227.07 |
| | | | Average of dust and reference | | | 179.62 |

[a]The calculation method of the normalized flux of $N_{NH4++NO3-}$ was discussed in Section 3.7.







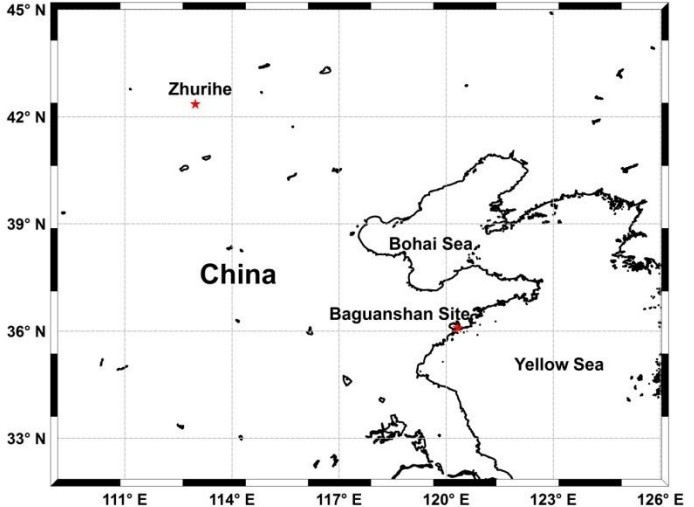

**Figure 1.** Location of the aerosol and dust sampling sites.


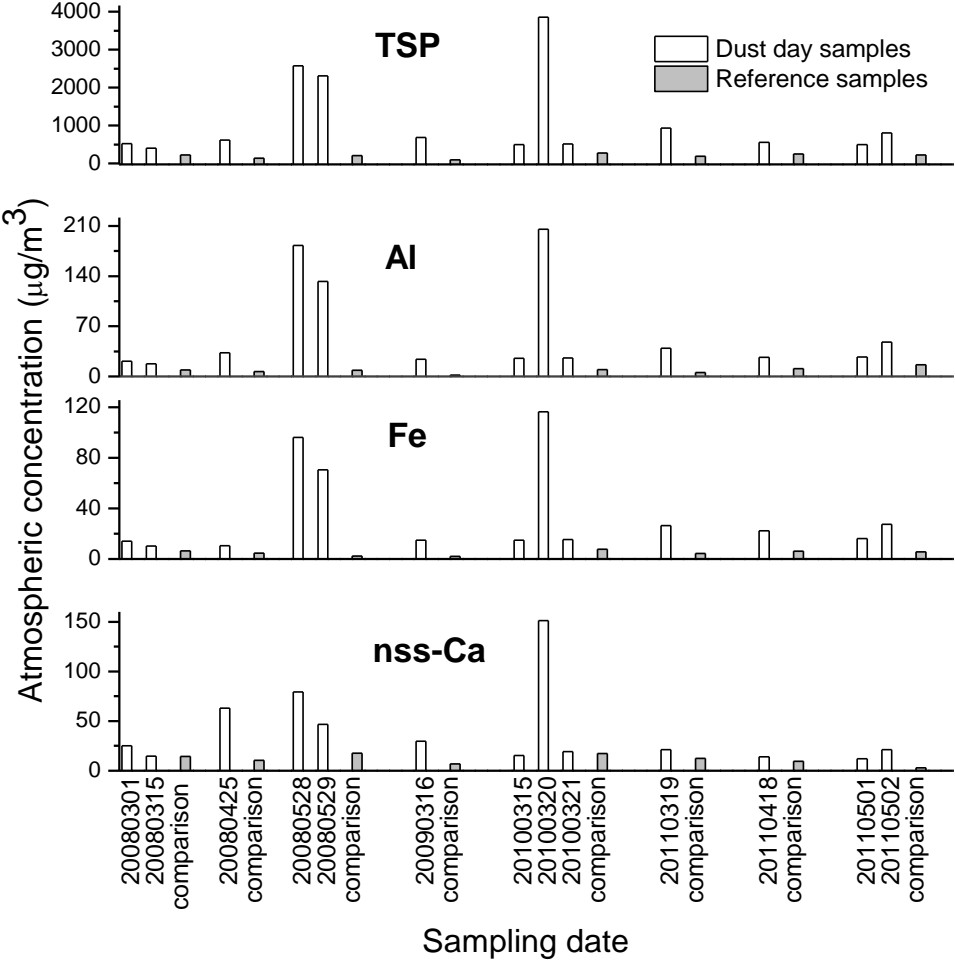


**Figure 2.** Mass concentrations of TSP, Al, Fe and nss-Ca in aerosol samples collected at the Baguanshan site on dust and reference days from 2008 to 2011.


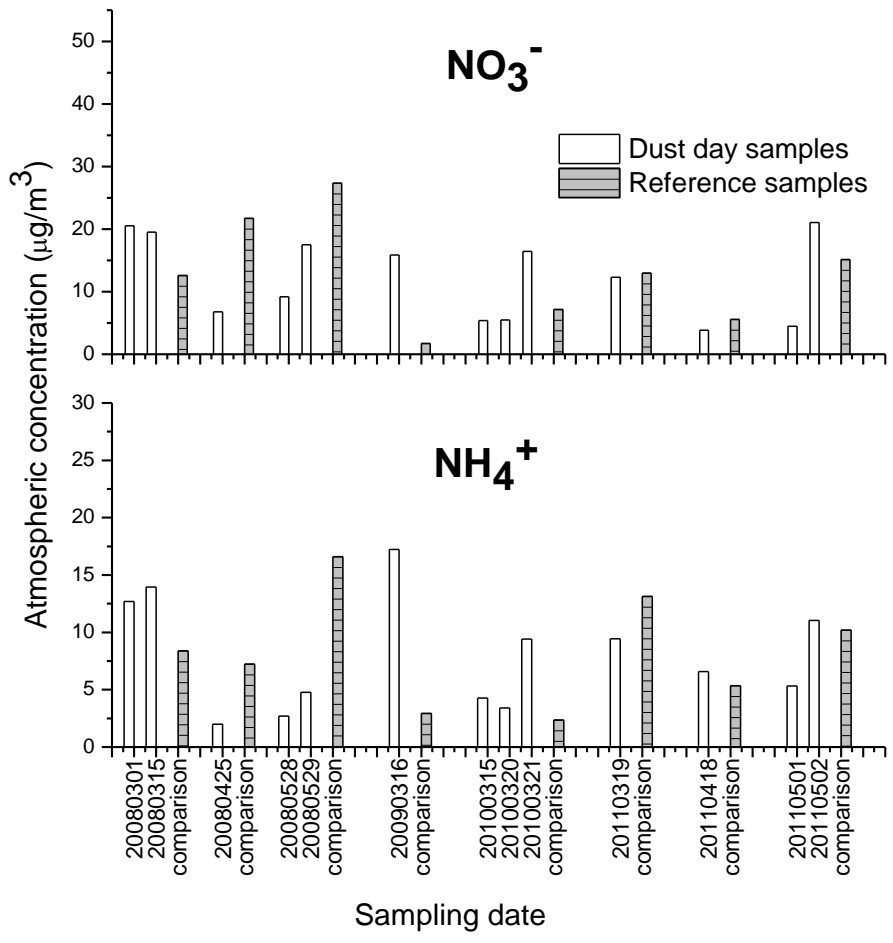


**Figure 3**. Mass concentrations of $NH_4^+$ and $NO_3^-$ in aerosol samples collected at the Baguanshan site
on dust and reference days during March-May in 2008 to 2011.




















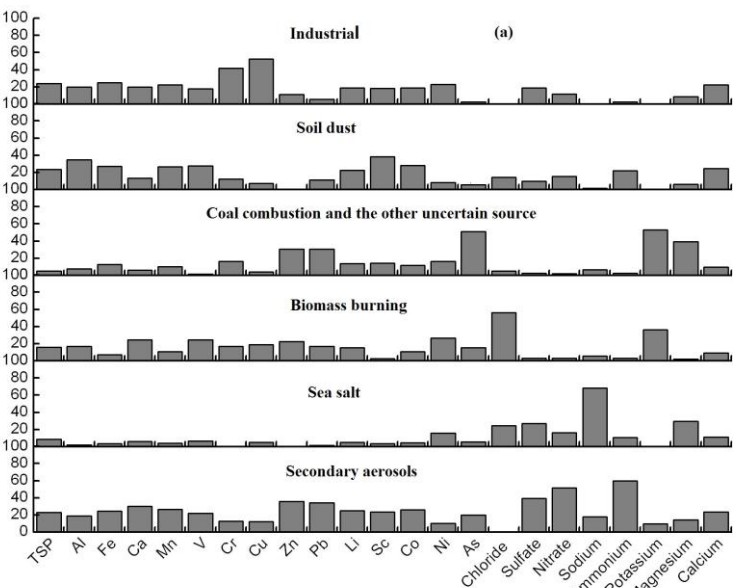


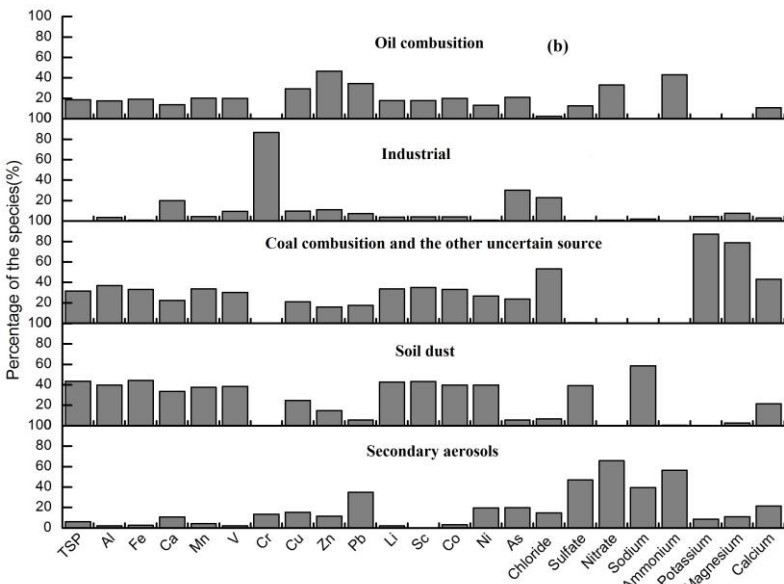

**Figure 4.** Source profiles of atmospheric aerosol samples collected on reference (a) and dust (b) days
using the PMF model.

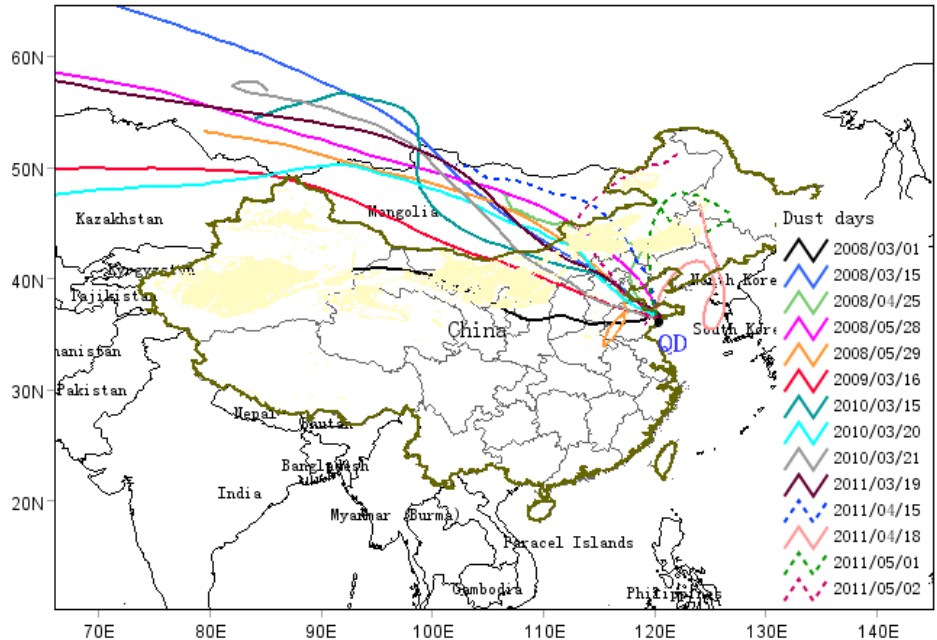


**Figure 5.** The 72-h backward trajectories for dust samples from 2008 to 2011 (the yellow domains in the map represent the dust source regions in China).


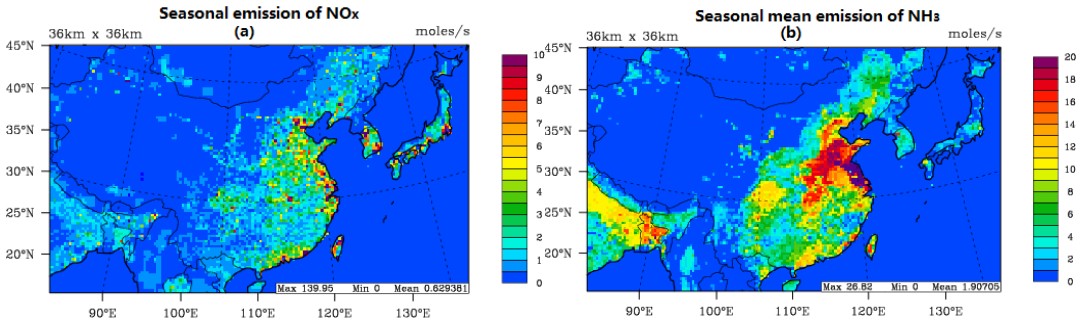

**Figure 6.** Seasonal mean emissions of NOx (a) and NH₃ (b) over East Asia from March-May 2008.