# Peer review of "The concentration, source and deposition flux of"

_Atmospheric Chemistry and Physics, 2016_

## Referee Comment (RC1) · Anonymous Referee #1 · 24 Mar 2017

General Comments:

The manuscript titled 'The concentration, source apportionment and deposition flux of atmospheric particulate inorganic nitrogen during dust events' written by Jianhua Qi presented the dust impacts on particulate inorganic nitrogen by analyzing the aerosol samples collected at Qingdao, China. The authors divided dust pattern into three parts, and investigated the dry deposition flux. To estimate the source, PMF receptor model was also used. Based on the above approaches, the authors tried to answer the questions of 'dust event always increase the atmospheric input of nitrogen to the ocean?'. The topic is interested ones because the impact of dust as atmospheric input on ocean ecosystem has been still unclarified. However, throughout the manuscript, it is not

well organized and hard to follow and understand. Overall, this manuscript will not be acceptable taking into account the high journal quality of Atmospheric Chemistry and Physics.

Before the discussion, first, the definition of "dust events" cannot be understood well. In L99-101, the authors explained that 'Samples were collected on dust days and selected ND days in spring from March 2008 to May 2011, with sampling duration of 4h for each sample. We refer to the ND days as sunny and cloudy days before and after dust events in the following discussion'. The authors should add the appropriate reference of the Meteorological Information Comprehensive Analysis and Process System (MICAPS) which defined the weather conditions (and also, the subsection 2.4 should be reorganized partly into this explanation). What is the definition of "dust events" here? Visibility? More information of how the dust events are defined in this system should be announced in detail. Total of 14 samples (sample numbers in Table 3) during dust events were analyzed throughout this study. The sampling duration was 4 hrs, so which data are used in the corresponded date in Table 3? All samples in the day? Moreover, what is the sample numbers of ND? The current information in Section 2.1 is severely lacked in the information which the readers can follow the authors methodology. Because this study discussed the dust impact, the explicit and detailed information regarding dust is required. In this sentence, I am worried about the explicit division of dust and non-dust samples. It is well known that some dust events are continued a few days. For example, the samples used in this study during 28-29 May 2008, 20-21 March 2010, 15 and 18 April 2011, and 1-2 May 2011 showed continuous dust events. In such cases, do the authors have confidence to the clear separation of dust and non-dust samples? How about the Al concentration definition (L171-172) of non-dust days samples? Why were other days samples not collected to clearly separate the dust impacts? The definition of ND is ambiguous. According to the definitions of dust and non-dust, the discussion on dust impact might be changed. The reconsideration of dust impact is needed based on the clear definitions of dust.

[Figure]

The second concern is the "dilution effect" which the authors claimed as the key factor for the discussion of inorganic nitrogen. Again, without the explicit definition of dust and non-dust, the dilution effect cannot be understood well. In this discussion, although the authors introduced the air mass speed, there were no implications on the intensity of dust events itself. Why the upwind (i.e., near desert) information was not used here to describe the dust intensity? The dilution is not so simple, hence more information are required to reinforce the authors finding. The authors discussed the inorganic nitrogen behavior. In these cases, what is the counter ion of $NH_4^+$ and $NO_3^-$? Are the main counter ions metal elements? If $NH_4NO_3$ are formed, due to its chemical unstablity according to the temperature and relative humidity, it is not simple to discuss only the viewpoint of "dilution effect". In addition, the authors used $NO_2$ data to investigate the inorganic nitrogen, but how about $NH_3$? Only from $NO_2$ data, it is insufficient to estimate the inorganic nitrogen variation. On the above reasons, the reconsideration is required to publish this manuscript from Atmospheric Chemistry and Physics.

Specific comments:

L35-36: This conclusion does not match to the manuscript contents. The authors stated that input of nitrogen to the ocean depends on the dust events.

L57-L67: In this paragraph, the authors used "ND days" simply. However, this wording should be used carefully; because the definition of non-dust days will be different in each study. Please consider to carefully define this wording.

L146: Some information should be replaced on Section 2.1 appropriately.

L162: "atmospheric particulate" is "TSP"?

L165: I cannot follow the calculation of "1.8-14.0 times (mean: 5.9)". The mean concentration have not been stated for dust days.

L167: The EF of Ca is 14.0 in Table 2.

L168: The statement of "decreased to less than three" cannot be followed from the

valued listed in Table 2.

L171: Again, I cannot follow the calculation of "1.7-21.9 times (mean: 6.9)".

L173-L174: To clarify the separation of dust and non-dust days, the information of criterion for samples on non-dust days will be needed.

L175: I cannot follow "10.3 times" for Fe. It can be calculated as 7.90 from the values in Table 2.

L175: In Figure 2, nss-Ca was shown, but nss-Ca was not listed in Table 2. What is the authors intention to introduce nss-Ca here?

L176: "3.6-fold" will not be followed from Fig. 2. It should be listed in Table 2.

L177: The EF of Ca on dust days is also greater that 10.

L183: The increasing ratio of concentration between dust days and non-dust days will be helpful to understand the discussion on Section 3.1.

L189: What is the comparison method on some dust days? The sample date are shown in Figure 3, so why the authors explicitly mention the date? I cannot follow the calculation of "a factor of 1.2-5.7 ".

L190: What means "less than 20% of that on ND days"? Averaged data over ND days?

L191: Again, what is the comparison method on some dust days? I cannot follow the calculation of "a factor of 1.4-9.2 ".

L194-L195: In this sentence, the authors stated "the effect of dust on inorganic nitrogen differed during different types of dust events". Why the authors suddenly focused on inorganic nitrogen here? In L192-193, it was mentioned "inorganic ion SO42- exhibited concentration variations that were similar to those of nitrate".

L197: The figures for inorganic nitrate will be helpful information here, if the authors focused on inorganic nitrogen.

L207: (respectively less than 50 ug/g and 6 ug/g) will be the correct expression for ammonium.

L211: So what is the source of atmospheric particulate nitrogen? The location of Duolun and Zhurihe Sand Desert is very close.

L214-L216: Without more information of the intensity of dust, the discussion on 'dilution effect' seems to be lacked in scientific understanding. This part should be fully revised based on not only dilution effect but also dust intensity.

L217: Averaged information were listed here, however, will the each sample information be valuable? The equation shown in summary column cannot be understood form (e.g., IN and ND were not comparable index).

L219: It seems that the discussion on this paragraph (e.g., "700 ug/m3 in Case 1" and "higher than 1100 ug/m3 in Cases 2 and 3") are based on Table 3. Please reorganize the paragraph, or please refer appropriate information here. It is hard to follow these values.

L219-L222: So what is the local source? What is the definition of the wording of "local" here? There was no information of the emissions here. It is hard to understand the "reaction" without the information of emissions intensity around dust source and downwind regions.

L224: "particle" is "TSP"?

L227-L228: The favorable condition to form ammonium cannot be discussed without the information of NH3. In addition, Table 3 indicated the aerosol samples in the coastal region of the Yellow Sea. How about the status over air mass path? Is it sufficient to conclude only from the downwind information to the formation of inorganic nitrogen?

L230: "strong dust storm" cannot be discussed without any information on dust intensity here.

L233-L234: But NOx concentration was high in Case 3. I cannot follow why the authors concluded "the strong dilution effect" on Case 3.

L244-L246: Because the Table 5 was lack in the information of ND days, we cannot follow the authors conclusion. The information of ND days on Table 5 will be required.

L254-L255: The authors simply mentioned "local emissions" here. Because the samples were collected on downwind regions in the coastal region of the Yellow Sea, I guess that the discussion on emission characteristics of each (or, at least, some categorized) air mass should be discussed in detail. The inorganic nitrogen concentrations are highly related to the local conditions both on emissions strength and meteorological parameters, so the discussion only on air mass speed and air mass path over ocean are insufficient.

L256: RH and NOx information are not shown in Table 5.

L260: The colors are overlapped, hence we cannot distinguish each trajectory. Some paths (e.g., thick green color: 2008/5/22 or 2011/4/15) are apparently indicated the west or south part of China. Are these events really related to dust events?

L278-L280: The source of coal combustion have increased compared to non-dust days. Short explanation will be needed here.

L305: If the authors discuss the dry deposition flux of "IN", the information should be inserted in Table 7. Table 7 only contained NO3- and NH4+ independently.

L306: I cannot follow the calculation of "a factor of 1.1-5.8" and "a factor of 1.8-6.3".

L307: "the dry deposition flux" of what?

L309: What is the calculation method of "63%" and "46%"?

L310: What is the calculation method of "14%"?

L317: I cannot follow the calculation of "a factor of 2-25".

L339: "aerosol particles" is "TSP"? In Table 7, please confirm the significant digits for each specie.

Technical Corrections:

L31: Comma is needed on '2800'.

L199: 'IN' should be defined in L194.

L236: Need appropriate comma for all numbers.

L301: Comma is needed on '2800±700'.

---

## Referee Comment (RC2) · Anonymous Referee #2 · 27 Mar 2017

General comments:

This paper attempts to study the impact of spring time dust storms in the Chinese deserts on the atmospheric concentration and deposition of inorganic nitrogen (IN) in the coastal Yellow Sea location. The study uses 4 years of particulate matter measurements and their chemical composition to understand how different types of dust storms can affect the abundance of inorganic nitrogen and calculate dry deposition of IN to the coastal Yellow Sea. This type of work is relevant since atmospheric deposition of nutrients and its implications is not a well understood topic and can be important for the regions that receive high atmospheric input like the Yellow Sea. However, the authors have not made best use of the data. There needs to be significant improvement in data
interpretation and much more analysis needs to be done to support the main results. I would thus recommend a resubmission and see if there is any substantial improvement in the manuscript. My main comments are given below.

First of all, the manuscript needs an overall improvement in writing. Not sufficient care has been given to the details and there are parts which are difficult to follow. I will provide some examples later, but there are many such cases of improper and awkward sentence constructions. Second, on what basis are these dusty and non-dusty days decided? More information are needed to show whether the dust storms originating from the deserts are actually passing over the measurement site and it is not the locally produced soil dust in cases of the days with low TSP values. Satellite aerosol products and meteorological data can be used to support this. Again later Al concentration is used to identify dust weather (lines 171-172). Please provide a clear definition of dust days and maintain that throughout the manuscript.

The authors have divided dusty days into 3 categories based on inorganic nitrogen (IN) concentrations relative to the non dust (ND) days. They are reporting that in some cases IN concentrations are more than the ND days, in some cases IN concentration are less than the ND days and in some cases nitrate concentration on dust days are less than ND days while ammonium concentrations on dust days are more compared to ND days. Next, the authors are reporting that sand from Duolun (which is a source for dust storms affecting the Yellow Sea region) is poor in nitrate and ammonium content. First of all, trajectory analysis does not seem to point that coastal Yellow Sea region is only affected by dust originating from Duolun region. There are many other dust sources over which the trajectories are passing. And if Duolun is deficient in IN you need to provide a detailed discussion on the possible sources of IN in dust sampled from the coastal region of the Yellow Sea and about the mixing of anthropogenic aerosols with the dusty air mass.

The authors have related the 3 cases of IN in dust samples to the wind speed concluding that when wind speed is less IN concentration increased and vice versa . I am

not clear how this conclusion is arrived at, especially, with only 5 cases for IN<ND. For example, in Table 5 sample number 110501 NO3- and NH4+ concentrations are low and the wind speed is also low. Again, sample 080315 has higher NO3- and NH4+ concentrations at higher wind speed. What is the rationale of using wind speed of 40.5 km/h in this study and how is this threshold derived? I would suggest the authors to group the trajectories according to dust and non dust days and also according to the levels of nitrogen and see which of these trajectories are passing over highly populated regions.

Specific comments:

L11-13: Reconstruct the sentence.

L20-25: This is not conclusive from the discussions that follow. Statements like "storms were weak or slow moving" and "rapid transport in a strong dust storm" are not supported by proper analysis of the storm characteristics.

L35-36: This is contradicting L32-33.

L40: These references point to anthropogenic contribution to atmospheric nitrogen deposition.

L62-64: The first part of the sentence seems to contradict the last part.

L79-81: Meaning is not clear.

L102: Zhurihe in Hunshandake Desert. Later you are using Duolun region which is not introduced in Section 2. It will be difficult for readers to follow if they are not very familiar with this region.

L135: Explain how the PMF model works.

L139: Explain how Williams' model works and cite the original paper.

L140. Expand U10. In general, all the abbreviations used should be clearly defined.

L143: Meteorological data not "climatic" data.

L146: This heading does not reflect the text content.

L152: Please provide details on the MICAPS information used.

L164: What is the average TSP concentration on dust days?

L167: Please provide a brief description on how EF is calculated and what is the significance of using this method.

L171-173: The authors were using MICAPS information of dust storm (which has to be explained) and now are relying on AI levels to define AD events. How are these two definitions consistent?

L178-179: This has to be explained with respect to the EF of the anthropogenic elements.

L214-216: You need to examine the dust sources, transport pathways (if passing over heavy populated regions), the height at which dust is transported together with dust concentration on a case by case basis to conclude these lines. This is very important for the entire paper. How do you decide "stronger a dust storm"?

L225-230: The average for Case 1 is 700 $\mu$g/m3 with values lying well above the average as is evident from Table 5. Again, there are TSP values in Case 2 in Table 5 which are lower than average of Case 1.

L230-231: Again, what is your definition of "strong dust storm"? Without sufficient analysis I am not sure how "dust might be transported quickly" is factoring here. This entire section has to be revisited.

L242: How is the value 40.5 km/h arrived at? Is it estimated at the dust source region? How many dust storms were studied to derive this value? More explanation is needed.

L250-252 needs explanation. Once the dust storms are properly categorized and the

pathways are determined the interpretation of Table 5 might change.

L250: Less than 40.5 km/h not 42.4 km/h.

L281-283: These statements need to be backed by more analysis of the dust events on a case by case basis.

L301-311: The text is very confusing. It is very difficult to follow when the authors are referring to TSP and when they are referring to IN or nitrate or ammonium.

Technical corrections:

L260: Colors used in Figure 4 are not clear. Please indicate the dust source regions on the map. L343-442: Not proper attention has been given to the References and needs to be corrected.

---

## Author Comment (AC1) · 25 Apr 2017

General Comments: The manuscript titled 'The concentration, source apportionment and deposition flux of atmospheric particulate inorganic nitrogen during dust events' written by Jianhua Qi presented the dust impacts on particulate inorganic nitrogen by analyzing the aerosol samples collected at Qingdao, China. The authors divided dust pattern into three parts, and investigated the dry deposition flux. To estimate the source, PMF receptor model was also used. Based on the above approaches, the authors tried to answer the questions of 'dust event always increase the atmospheric input of nitrogen to the ocean?'. The topic is interested ones because the impact of dust as atmospheric input on ocean ecosystem has been still unclarified. However,

throughout the manuscript, it is not well organized and hard to follow and understand. Overall, this manuscript will not be acceptable taking into account the high journal quality of Atmospheric Chemistry and Physics.

Reply: We will revise the manuscript according to the comments to improve the manuscript quality.

Q1.Before the discussion, first, the definition of "dust events" cannot be understood well. In L99-101, the authors explained that 'Samples were collected on dust days and selected ND days in spring from March 2008 to May 2011, with sampling duration of 4h for each sample. We refer to the ND days as sunny and cloudy days before and after dust events in the following discussion'. The authors should add the appropriate reference of the Meteorological Information Comprehensive Analysis and Process System (MICAPS) which defined the weather conditions (and also, the subsection 2.4 should be reorganized partly into this explanation). What is the definition of "dust events" here? Visibility? More information of how the dust events are defined in this system should be announced in detail. Total of 14 samples (sample numbers in Table 3) during dust events were analyzed throughout this study. The sampling duration was 4 hrs, so which data are used in the corresponded date in Table 3? All samples in the day? Moreover, what is the sample numbers of ND? The current information in Section 2.1 is severely lacked in the information which the readers can follow the authors methodology. Because this study discussed the dust impact, the explicit and detailed information regarding dust is required. In this sentence, I am worried about the explicit division of dust and non-dust samples. It is well known that some dust events are continued a few days. For example, the samples used in this study during 28-29 May 2008, 20-21 March 2010, 15 and 18 April 2011, and 1-2 May 2011 showed continuous dust events. In such cases, do the authors have confidence to the clear separation of dust and non-dust samples? How about the Al concentration definition (L171-172) of non-dustdays samples? Why were other days samples not collected to clearly separate the dust impacts? The definition of ND is ambiguous. According to the definitions of dust

and non-dust, the discussion on dust impact might be changed. The reconsideration of dust impact is needed based on the clear definitions of dust.

Reply: In this study, the dust event was defined according the definition adopted in regulations of surface meteorological observation of China (CMA, 2003;Wang et al., 2008) and identified based on the meteorological records information from Meteorological Information Comprehensive Analysis and Process System (MICAPS) of China Meteorological Administration.Each dust sample was collected for 4hrs duration and the sampling started only when the PM10mass concentration available on the website (http://www-cfors.nies.go.jp/ cfors/; http://www.qepb.gov.cn/m2/) was increased greatly. The approach made the dust sample more representative relative to urban background. However, for dust event with duration less than one day, only one sample was collected;; for dust event with longer duration, i.e. multiple days, the sample was collected once a day. The sampling information was listed in the Table S1. Based on the forecast, we also collected aerosol particle samples immediately before or after the dust event for comparison. These comparison samples were further classified into sunny day samples, cloudy day samples and post-dust samples. The post-dust samples were featured by collecting under a clear and sunny weather condition and lower mass concentration of PM10. Moreover, the concentration of Alreferring to the total Al concentration in TSPsamples were used to confirm the division of dust or comparison samples according to the criterion "geometric mean$\times$2GSD" proposed by Hsu et al. (2008).

CMA: Regulations of Surface Meteorological Observation, China Meteorological Press, Beijing, 154–156, 2004. Hsu, S. C., Liu, S. C., Huang, Y. T., Lung, S. C. C., Tsai, F., Tu, J. Y., and Kao, S. J.: A criterion for identifying Asian dust events based on Al concentration data collected from northern Taiwan between 2002 and early 2007, Journal of Geophysical Research Atmospheres, 113, 1044-1044, 2008. Wang Y. Q., Zhang X. Y., Gong S. L., Zhou C. H., Hu X. Q., Liu H. L., Niu T., Yang Y. Q.: Surface observation of sand and dust storm in East Asia and its application in CUACE/Dust, Atmos. Chem. Phys., 8, 545–553, 2008.

Q2.The second concern is the "dilution effect" which the authors claimed as the key factor for the discussion of inorganic nitrogen. Again, without the explicit definition of dust and non-dust, the dilution effect cannot be understood well. In this discussion, although the authors introduced the air mass speed, there were no implications on the intensity of dust events itself. Why the upwind (i.e., near desert) information was not used here to describe the dust intensity? The dilution is not so simple, hence more information are required to reinforce the authors finding. The authors discussed the inorganic nitrogen behavior. In these cases, what is the counter ion of NH4+ and NO3-? Are the main counter ions metal elements? If NH4NO3 are formed, due to its chemical unstablity according to the temperature and relative humidity, it is not simple to discuss only the viewpoint of "dilution effect". In addition, the authors used NO2 data to investigate the inorganic nitrogen, but how about NH3? Only from NO2 data, it is insufficient to estimate the inorganic nitrogen variation. On the above reasons, the reconsideration is required to publish this manuscript from Atmospheric Chemistry and Physics.

Reply: In revision, the part reads as "Inorganic nitrogen (IN) concentrations highly varied in different dust samples (Table 3). According to the concentrations relative to those in comparison samples, they can be classified into three categories, i.e., Category 1 in which higher IN concentrations were observed in dust samples, Category 2 in whichlowerIN concentrations were observedin dust samples, and Category 3 in which lower nitrate concentrations with slightly higherconcentrationsof ammoniumin dust samples. Category 1 was usually associated with a lower moving speed of dust air mass or a longer distance over the ocean (Table 5) while the reverse was true for Category 2. The moving speed and distance over the ocean of dust air mass in Category 3 was generally between them. Theoretically, lower moving speed of dust air mass favors reactions between dust particles and anthropogenic gaseous precursors of IN due to a longer reaction time. Largemoving speed of dust air mass was frequently associated with a large wind speed in the lower layer atmosphere (Gao et al., 2010; Gillette and Passi, 1988; Peng et al., 2007; Yue et al., 2008), leading to anthropogenic
gaseous precursors therein to be better diluted. Shorter reaction time and reduced concentrations of anthropogenic gaseous precursors likely lowered IN in Category 2. Moreover, the relative concentration of IN per aerosol particle mass in $\mu$g/g was analyzed and compared with those values in literature. .." It is questionable for using NOx observed in Qingdao to argue the generation of IN in dust samples. We agree this because most of IN observed in dust samples should be derived from secondary reactions upwind of Qingdao by considering a low conversion rate of NOx to IN. The former study (Liu et al., 2010) showed that NOx and NH3 generally capture the spatial distribution patterns with high values over eastern China and relatively lower values over central and western China, where dust source regions are located(Fig. S1-S3). Thus, we will add modeling results using a 3-D air quality model to support our analysis in revision.

Gao, Q X., Ren Z H. et al. : Dust events and its impacts on atmospheric environment, Science press, Beijing, 2010. Gillett e D A, Passi R.: Modeling dust emission caused bywind erosion, J G R., 1988, 93: 14234- 14242. Liu X. H., Zhang Y., Cheng S. H., Xing J., Zhang Q., Streets D. G., Jang C., Wang W. X., Hao J. M.:Understanding of regional air pollution over China using CMAQ, part I performance evaluation and seasonal variation, Atmospheric Environment , 44,2415-2426, 2010. Peng, Z., Liu X. M., Hong Z. X., Wang B. L.: Characteristics of Atmospheric BoundaryLayer Structure and Turbulent FluxTransfer during a Strong Dust StormWeather Process over Beijing Area, Climatic and Environmental Research, 2007, 12(3): 268-276. Qi J.H., Gao H.W., Yu L.M. , Qiao J.J.: Distribution of inorganic nitrogen-containing species in atmospheric particles from an island in the Yellow Sea, Atmospheric Research, 101,938-955, 2011. Wang Y. Q., Zhang X. Y., Gong S. L., Zhou C. H., Hu X. Q., Liu H. L., Niu T., Yang Y. Q.: Surface observation of sand and dust storm in East Asia and its application in CUACE/Dust, Atmos. Chem. Phys., 8, 545–553, 2008. Yue P., Niu S. J., Liu X. Y.: Dust Emission and Transmission during Spring Sand-dustStorm in Hunshandake Sand-land, Journal of Desert Research, 2008, 28(2): 227-230.

Specific comments: Q3.L35-36: This conclusion does not match to the manuscript contents. The authors stated that input of nitrogen to the ocean depends on the dust events.

Reply: We apologize for the confusion in the revision. We will revise the abstract sentence into "The atmospheric input of nitrogen into the ocean depends on the dust events; dust deposition was an uncertain source of nitrogen for the ocean".

Q4.L57-L67: In this paragraph, the authors used "ND days" simply. However, this wording should be used carefully; because the definition of non-dust days will be different in each study. Please consider to carefully define this wording.

Reply: Thank you for this suggestion. To avoid confusion, we will use "non-dust storm days" according to the original reference in L57-L67.

Q5.L146: Some information should be replaced on Section 2.1 appropriately.

Reply: We will move this information to Section 2.1 in the revised version.

Q6.L162: "atmospheric particulate" is "TSP"?

Reply: We apologize for the confusion. The term "atmospheric particulate" will be revised to "total suspended particulates". Atmospheric particulate concentrations were obtained by weighting TSP samples. We will revise the sentence and the corresponding figures.

Q7.L165: I cannot follow the calculation of "1.8-14.0 times (mean: 5.9)". The mean concentration have not been stated for dust days.

Reply: Each sample on dust day had its corresponding non-dust sample (Table S2).The 1.8-14.0 times was calculated as a ratio of the TSP concentration on a given dust day to the values in the comparison samples. The concentration and the ratio of samples on dust days were listed in Table S2. Q8.L167: The EF of Ca is 14.0 in Table 2. L168: The statement of "decreased to less than three" cannot be followed from the valued

listed in Table 2.

Reply: We apologize for this error and will revise the incorrect description to read "the enrichment factors (EFs) of Al, Fe, and Mg were lower than ten on ND days and decreased to less than three on dust days. These data are indicative of the primarily crustal origins of these elements. Furthermore, the EF of Ca was 14.0 on ND days, which indicated that Ca had a partially anthropogenic source on dust days".

Q9.L171: Again, I cannot follow the calculation of "1.7-21.9 times (mean: 6.9)".

Reply: We apologize for the confusion. The calculation method is the same as that for TSP (see the reply to Q7). The correct concentrations and the ratios of samples on dust days are listed in Table S2.

Q10.L173-L174: To clarify the separation of dust and non-dust days, the information of criterion for samples on non-dust days will be needed. Reply: As discussed above for Q1, the information will be supplemented in Section 2.1.

Q11.L175: I cannot follow "10.3 times" for Fe. It can be calculated as 7.90 from the values in Table 2.

Reply: The calculation method is the same as that for TSP (see the reply to Q7). The concentrations and corrected mean ratios of samples on dust days are listed in Table S2.

Q12.L175: In Figure 2, nss-Ca was shown, but nss-Ca was not listed in Table 2. What is the authors intention to introduce nss-Ca here?

Reply: Follow others' study, we calculated the EF of Ca in Table 2. The EFs of Ca on ND days indicated that Ca was affected by anthropogenic sources. nss-Ca usually was used as a typical dust index. Therefore we showed the nss-Cain Fig.2 and discussed the influence of dust on crustal elements using nss-Ca.

Q13.L176: "3.6-fold" will not be followed from Fig. 2. It should be listed in Table 2.

Reply: The calculation method is the same as that for TSP (see the reply to Q7). The concentrations and the corrected mean ratios of samples on dust days are listed in Table S2.

Q14.L177: The EF of Ca on dust days is also greater that 10.

Reply: The EF of Ca was 14, not much greater than 10, indicating that the Ca was mainly from a natural source mixed with an anthropogenic source.

Q15.L183: The increasing ratio of concentration between dust days and non-dust days will be helpful to understand the discussion on Section 3.1.

Reply: We will replace the times with ratios in our revised manuscript.

Q16.L189: What is the comparison method on some dust days? The sample date are shown in Figure 3, so why the authors explicitly mention the date? I cannot follow the calculation of "a factor of 1.2-5.7 ".

Reply: It will be revised as "The concentrations of ammonium were increased by 20

Q17.L190: What means "less than 20

Reply: We apologize for the confusion. The sentence has been revised to read " The concentrations of ammonium were increased by 20

Q18.L191: Again, what is the comparison method on some dust days? I cannot follow the calculation of "a factor of 1.4-9.2 ".

Reply: The calculation method is the same as that for ammonium (see the reply to Q16). The concentrations and the increasing factors of samples on dust days are listed in Table S2.

Q19.L194-L195: In this sentence, the authors stated "the effect of dust on inorganic nitrogen differed during different types of dust events". Why the authors suddenly focused on inorganic nitrogen here?In L192-193, it was mentioned "inorganic ion SO42-

exhibited concentration variations that were similar to those of nitrate". L197: The figures for inorganic nitrate will be helpful information here, if the authors focused on inorganic nitrogen.

Reply: The part will be revised as "Similar to ammonium, nitrate concentrations were sometimes increased by a factor of 1.4-9.2 relative to the comparison sample while they were decreased in others. Unlike substantially increased concentrations of crustal metal elements in dust samples, the concentrations of IN were likely determined by meteorological conditions as well as surface areas provided by dust particles."

Q20.L207: (respectively less than 50 ug/g and 6 ug/g) will be the correct expression for ammonium.

Reply: We have incorporated this suggestion.

Q21.L211: So what is the source of atmospheric particulate nitrogen? The location of Duolun and Zhurihe Sand Desert is very close.

Reply: Duolun and Zhurihe belong to the Hunshandake Desert in Inner Mongolia, one of the main Chinese sand deserts. According to studies, the Yellow Sea is mainly affected by dust storms from this sand source with a probability of 52

Zhang, Z K., and Gao, H.: The characteristics of Asian-dust storms during 2000–2002: From the source to the sea, Atmospheric Environment, 41, 9136-9145, 2007. Gao, Q X., Ren Z H. : Dust events and its impacts on atmospheric environment, Science press, Beijing, 2010.

Q22.L214-L216: Without more information of the intensity of dust, the discussion on 'dilution effect' seems to be lacked in scientific understanding. This part should be fully revised based on not only dilution effect but also dust intensity.

Reply: As discussed above, we will add modeling results of dust distribution to support our analysis in revision.

Q23.L217: Averaged information were listed here, however, will the each sample information be listed here? The equation shown in summary column cannot be understood form (e.g., IN and ND were not comparable index).

Reply: Thank you for the suggestion. According to the suggestion, we revised Table 3 and listed the sample information.

Q24.L219: It seems that the discussion on this paragraph (e.g., "700 ug/m3 in Case 1" and "higher than 1100 ug/m3 in Cases 2 and 3") are based on Table 3. Please reorganize the paragraph, or please refer appropriate information here. It is hard to follow these values.

Reply: We will revise this paragraph and refer to the appropriate information in the revised manuscript according to revised Table 3.

Q25.L219-L222: So what is the local source? What is the definition of the wording of "local" here? There was no information of the emissions here. It is hard to understand the "reaction" without the information of emissions intensity around dust source and downwind regions.

Reply: Local source refers to the gas or particle emissions from a local pollutant source, such as industry emission, coal burning, vehicle exhaust and agricultural activity, in the downwind region during the dust transport, which is not from the dust event itself. As we discussed above, the NOx and NH3 emissions increase greatly from the dust source region to the downwind region (see the reply to Q2). We have supplemented the modeled emissions intensity of NOx and NH3 in the revised manuscript.

Q26.L224: "particle" is "TSP"?

Reply: We apologize for the confusion. We will revise "particle" to read "total suspended particles".

Q27.L227-L228: The favorable condition to form ammonium cannot be discussed without the information of NH3. In addition, Table 3 indicated the aerosol samples in the

coastal region of the Yellow Sea. How about the status over air mass path? Is it sufficient to conclude only from the downwind information to the formation of inorganic nitrogen?

Reply: We will add modeling results using a 3-D air quality model to support our analysis in revision.

Q28.L230: "strong dust storm" cannot be discussed without any information on dust intensity here.

Reply: We will add modeling results of dust distribution to support our analysis in revision.

Q29.L233-L234: But NOx concentration was high in Case 3. I cannot follow why the authors concluded "the strong dilution effect" on Case 3.

Reply: Among three cases, the NOx concentration was the highest with an average value of 70.7$\mu$g•m-3 for Case 3 and increased by 17.8

Q30.L244-L246: Because the Table 5 was lack in the information of ND days, we cannot follow the authors conclusion. The information of ND days on Table 5 will be required.

Reply: We have supplemented the information for ND days in Table S1 and S2.

Q31.L254-L255: The authors simply mentioned "local emissions" here. Because the samples were collected on downwind regions in the coastal region of the Yellow Sea, I guess that the discussion on emission characteristics of each (or, at least, some categorized) air mass should be discussed in detail. The inorganic nitrogen concentrations are highly related to the local conditions both on emissions strength and meteorological parameters, so the discussion only on air mass speed and air mass path over ocean are insufficient.

Reply: As discussed above (see the Reply to Q2) , We will add modeling results using

a 3-D air quality model to support our analysis in revision.

Q32.L256: RH and NOx information are not shown in Table 5. Reply: We apologize for the mistake. We have revised the title of Table 5.

Q33.L260: The colors are overlapped, hence we cannot distinguish each trajectory. Some paths (e.g., thick green color: 2008/5/22 or 2011/4/15) are apparently indicated the west or south part of China. Are these events really related to dust events?

Reply: We apologize for the confusion. We have provided all trajectories of samples collected on dust and non-dust days. Fig.4 has been redrawn to distinguish each trajectory for samples collected on dust and non-dust days.

Q34.L278-L280: The source of coal combustion have increased compared to non-dust days. Short explanation will be needed here.

Reply: The source of coal combustion on dust days became complex. The source profile showed high percentages of K+, Cl-, Ca, Mg, Co, Ni, As, Al and Fe, indicating a mixture of coal combustion and other pollutants emitted along the transmission path on dust days, such as industry and building dust. This source increased due to the coal combustion emissions mixing with other uncertain sources emitted into the air in strong winds.

Q35.L305: If the authors discuss the dry deposition flux of "IN", the information should be inserted in Table 7. Table 7 only contained NO3- and NH4+ independently.

Reply: We inserted the flux of IN in Table 7 and corrected several mistakes.

Q37.L306: I cannot follow the calculation of "a factor of 1.1-5.8" and "a factor of 1.8-6.3".

Reply: These factors were the flux ratio of each dust sample in Case 1 to the ND average. The flux and ratio of each sample are listed in Table S3. We recalculated the increasing factors according to the revised values. The sentence was revised to read
"Compared with the average flux on ND days, the dry deposition flux of IN increased by a factor of 1.1-3.9, and the flux of atmospheric particles (TSP) increased by a factor of 1.8-6.3 in Case 1"

Q38.L307: "the dry deposition flux" of what?

Reply: We apologize for the mistake in the revision. The passage has been revised to read "the dry deposition flux of atmospheric particles (TSP)".

Q39.L309: What is the calculation method of "63

Reply: We apologize for the mistake. The sentence has been revised to read "Compared with the average dry deposition flux on ND days, the average nitrate flux of samples in Cases 2 and 3 decreased by 73

Q40.L310: What is the calculation method of "14

Reply: We corrected the calculation error and revised this sentence to read "Additionally, the average ammonium flux decreased by 47

Q41.L317: I cannot follow the calculation of "a factor of 2-25".

Reply: The factor was calculated by comparing the flux of the sample on dust days with the average Fe flux on ND days (see Table S3).

Q42.L339: "aerosol particles" is "TSP"? In Table 7, please confirm the significant digits for each specie.

Replyïij¼We apologize for the confusion. "aerosol particles" was revised to read "TSP". The former digits were revised according to the editor's suggestion. We will consider revising again to confirm the significant digits.

Q43. Technical Corrections: L31: Comma is needed on '2800'.

Reply: We have added a comma according to the suggestion.

Q44.L199: 'IN' should be defined in L194.

Reply: Due to the very low concentration of nitrite, in this manuscript, IN represents inorganic nitrogen, mainly including nitrate and ammonium. We have provided this definition in L194.

Q45.L236: Need appropriate comma for all numbers. L301: Comma is needed on '2800±700'.

Reply: We have added a comma according to the suggestion.

Please also note the supplement to this comment:
http://www.atmos-chem-phys-discuss.net/acp-2016-1183/acp-2016-1183-AC1-supplement.pdf

———————————————

[Figure]

Fig. S1 Daily mean NOx emission in China on Apr.24th (a), 25th (b) and 26th (c), 2008 (From Liu et al., 2010).

[Figure]

Fig. S2 Daily mean NH$_3$emission in China on Apr.24th (a), 25th (b) and 26th (c), 2008 (from Liu et al., 2010).

[Figure]

[Figure]

Fig. S3 Map of the main source regions of sand and dust storms in China (from Wang et al., 2008).

[Figure]

**Figure 4.**The 72-h backward trajectories for non-dust (a) and dust (b) samples from 2008 to 2011

Table S1. Sampling information for the aerosols samples collected at the Baguanshan site in the coastal region of the Yellow Sea.

| Sampling Year | Sample category | Sampling number | Sampling time | Weather characteristics |
|---|---|---|---|---|
| 2008 | Samples on dust days | 20080301 | From 13:22 a.m. to 17:22 p.m. on Mar. 1st | Floating dust[a] |
| | | 20080315 | From 13:21 a.m. to 17:21 p.m. on Mar. 15 th | Floating dust |
| | | 20080425 | From 13:14 a.m. to 17:14 p.m. on Apr. 25th | Floating dust |
| | | 20080528 | From 11:38 a.m. to 15:38 p.m. on May 28th | Floating dust |
| | | 20080529 | From 10:15 a.m.to 12:15 p. m. on May 29th[b] | Floating dust |
| | Samples on non-dust days | 20080316 | From 13:00 a.m. to 17:00 p.m. on Mar. 16th | Sunny day |
| | | 20080424 | From 13:00 a.m. to 17:00 p.m. on Apr. 24th | Sunny day |
| | | 20080522 | From 13:00 a.m. to 17:00 p.m. on May 22th | Cloudy day with mist |
| 2009 | Samples on dust days | 20090316 | From 8:25 a.m. to 12:25 p.m. on Mar. 16th | Floating dust |
| | Samples on non-dust days | 20090306 | From 13:00 a.m.to 17:00 p.m. on Mar. 6th | Sunny day |
| 2010 | Samples on dust days | 20100315 | From 11:30 a.m.to 15:30 p.m. on Mar. 16th | Floating dust |
| | | 20100320 | From 10:30 a.m. to 14:30 p.m. on Mar. 20th | Floating dust |
| | | 20100321 | From 10:30 a.m. to 14:30 p.m. on Mar. 21th | Floating dust |
| | Samples on non-dust days | 20100324 | From 11:30 a.m. to 15:30 p.m. on Mar. 24th | Sunny day |
| 2011 | Samples on dust days | 20110319 | From 12:00 a.m. to 16:00 p.m. on Mar. 19th | Floating dust |
| | | 20110415 | From 12:00 a.m. to 16:00 p.m. on Apr. 15th | Floating dust |
| | | 20110418 | From 12:25 a.m. to 16:25 p.m. on Apr. 18th | Floating dust |
| | | 20110501 | From 12:10 a.m. to 16:10 p.m. on May 1st | Floating dust |
| | | 20110502 | From 16:00 a.m. to 20:00 p.m. on May 2nd | Floating dust |
| | Samples on non-dust days | 20110308 | From 12:00 a.m. to 16:00 p.m. on Mar. 8th | Sunny day |
| | | 20110416 | From 12:00 a.m. to 16:00 p.m. on Apr. 16th | Sunny day |
| | | 20110523 | From 12:00 a.m. to 16:00 p.m. on May 23th | Sunny day |

[a]Note that one exterior dust sample was collected on 1 March when no dust was recorded by MICAPS. However, the MICPAS information indeed showed the dust events in Qingdao on 29 February and 2 March. Both the $PM_{10}$ mass concentration and our measured Al in TSP on 1 March implied that the sample should be classified into dust sample.

[b] In addition, the sampling duration was reduced down to only 2 hrs because of extremely high particle loadings.

**Fig. 4.** Table

Table S2. Sampling information for the aerosol samples collected at the Baguanshan site in thecoastal region of the Yellow Sea.

| Sampling Month | Sample category | Sampling number | TSP Concentration (µg/m³) | Ratio of DD to NDS | Al Concentration (µg/m³) | Ratio of DD to NDS | Fe Concentration (µg/m³) | Ratio of DD to NDS | nss-Ca Concentration (µg/m³) | Ratio of DD to NDS | NH$_4^+$ Concentration (µg/m³) | Ratio of DD to NDS | NO$_3^-$ Concentration (µg/m³) | Ratio of DD to NDS |
|---|---|---|---|---|---|---|---|---|---|---|---|---|---|---|
| Mar., 2008 | Dust days (DD) | 20080301 | 526.7 | 2.3 | 21.3 | 2.4 | 14.1 | 2.2 | 25.2 | 1.8 | 12.7 | 1.5 | 20.5 | 1.6 |
| | | 20080315 | 409.5 | 1.8 | 17.5 | 1.9 | 10.3 | 1.6 | 14.6 | 1.0 | 29.9 | 3.6 | 19. 5 | 1.5 |
| | Non-dust days (NDS) | 20080316 | 225.1 | | 9.0 | | 6.5 | | 14.3 | | 8.4 | | 12.6 | |
| Apr., 2008 | Dust days (DD) | 20080425 | 622.2 | 4.5 | 33.2 | 5.0 | 10.6 | 2.3 | 63.1 | 6.1 | 2.0 | 0.3 | 6.8 | 0.3 |
| | Non-dust days (NDS) | 20080424 | 137.5 | | 6.6 | | 4.7 | | 10.4 | | 7.2 | | 21.7 | |
| May, 2008 | Dust days (DD) | 20080528 | 2578.7 | 12.5 | 182.8 | 20.9 | 96.1 | 41.4 | 79.2 | 4.5 | 2.7 | 0.2 | 9.2 | 0.3 |
| | | 20080529 | 2313.8 | 11.2 | 132.7 | 15. 2 | 70.5 | 30.4 | 46.8 | 2.7 | 4.8 | 0.3 | 17.5 | 0.6 |
| | Non-dust days (NDS) | 20080522 | 206.1 | | 8.8 | | 2.3 | | 17.6 | | 16.6 | | 27.4 | |
| Mar., 2009 | Dust days (DD) | 20090316 | 688.4 | 7.3 | 24.2 | 13.7 | 14.8 | 7.7 | 29.6 | 4.2 | 17.2 | 5.7 | 15.9 | 5.4 |
| | Non-dust days (NDS) | 20090306 | 94.3 | | 1.8 | | 1.9 | | 7.0 | | 3.0 | | 2.9 | |
| Mar., 2010 | Dust days (DD) | 20100315 | 501.1 | 1.8 | 25.6 | 2.7 | 14.8 | 2.0 | 15.2 | 0.9 | 4.3 | 1.8 | 5. 4 | 0.8 |
| | | 20100320 | 3856.7 | 14.0 | 205.4 | 21. 9 | 116.3 | 15.3 | 151.1 | 8. 7 | 3.4 | 1.4 | 5.5 | 0.8 |
| | | 20100321 | 518.6 | 1.9 | 25.8 | 2. 8 | 15.3 | 2.0 | 19.2 | 1.1 | 9.4 | 4.0 | 16.5 | 2.3 |
| | Non-dust days(NDS) | 20100324 | 274.8 | | 9.38 | | 7.6 | | 17.4 | | 2.4 | | 7.2 | |
| Mar., 2011 | Dust days (DD) | 20110319 | 938.6 | 4.8 | 39.3 | 7.0 | 26.3 | 6.0 | 21.2 | 1.7 | 9.4 | 0.7 | 12.3 | 0.9 |
| | Non-dust days (NDS) | 20110308 | 194.1 | | 5.6 | | 4.4 | | 12.4 | | 13.1 | | 13.0 | |
| Apr., 2011 | Dust days (DD) | 20110415 | 1224.6 | 4.9 | 52.8 | 4.8 | 35.3 | 5.7 | 41.6 | 4.4 | 25.0 | 4.7 | 51. 4 | 9.2 |
| | | 20110418 | 557.9 | 2.2 | 26.6 | 2.4 | 22.3 | 3.6 | 14.2 | 1.5 | 6.6 | 1.2 | 3.8 | 0.7 |

**Fig. 5.** Table

**Table 3.** Concentrations of inorganic nitrogen, TSP, NOx, Relative Humidity (RH) and T for aerosol samples of different casein the coastal region of the Yellow Sea.

| | Sample number | TSP μg·m⁻³ | NO₃⁻ μg·m⁻³ | NH₄⁺ μg·m⁻³ | RH % | T °C | NOx μg·m⁻³ | Summary |
|---|---|---|---|---|---|---|---|---|
| Case 1 | 20080301 | 527 | 20.5 | 12.7 | 57 | 7.0 | 36 | IN concentration on dust days higher than that on ND days |
| | 20080315 | 410 | 19.5 | 29.9 | 62 | 11.0 | 59 | |
| | 20090316 | 688 | 15.9 | 17.2 | 27 | 16.0 | 75 | |
| | 20100321 | 519 | 16.5 | 9.4 | 51 | 8.8 | 76 | |
| | 20110415 | 1224 | 51.4 | 25.0 | 34 | 22.3 | 68 | |
| | 20110502 | 810 | 21.0 | 11.0 | 49 | 17.7 | 62 | |
| Case 2 | 20080425 | 622 | 6.8 | 2.0 | 30 | 18.0 | 40 | IN concentration on dust days lower than that on ND days |
| | 20080528 | 2579 | 9.2 | 2.7 | 17 | 27.0 | 34 | |
| | 20080529 | 2314 | 17.5 | 4.8 | 60 | 20.0 | 29 | |
| | 20110319 | 939 | 12.3 | 9.4 | 16 | 12.6 | 93 | |
| | 20110501 | 502 | 4.5 | 5.3 | 23 | 21.6 | 66 | |
| Case 3 | 20100315 | 501 | 5.4 | 4.3 | 30 | 7.2 | 73 | NO₃⁻ concentration on dust days lower than that on ND days; NH₄⁺close to that on ND days |
| | 20100320 | 3857 | 5.5 | 3.4 | 35 | 10.6 | 92 | |
| | 20110418 | 558 | 3.8 | 6.6 | 33 | 12.6 | 47 | |
| Non-dust [a] | 20080316 | 225 | 12.6 | 8.4 | 28 | 11.0 | 60 | |
| | 20080424 | 137 | 21.7 | 7.2 | 49 | 18.0 | 53 | |
| | 20080522 | 206 | 27.4 | 16.6 | 78 | 20.0 | 60 | |
| | 20090306 | 94 | 2.9 | 3.0 | 29 | 7.00 | 51 | |
| | 20100324 | 275 | 7.2 | 2.4 | 23 | 9.0 | 82 | |
| | 20110308 | 194 | 13.0 | 13.1 | 20 | 11.5 | 111 | |
| | 20110416 | 252 | 5.6 | 5.4 | 26 | 14.1 | 55 | |
| | 20110523 | 224 | 15.2 | 10.2 | 42 | 20.6 | 49 | |

[a] The corresponding ND day for each dust event see Table S2.

**Fig. 6.** Table

**Table 7.** Dry deposition of aerosol particles (mg/m$^2$/month), particulate inorganic nitrogen (mg N/m$^2$/month) and some toxic trace metals (mg/m$^2$/month) on dust and non-dust days

| | TSP | $NO_3^-$-N | $NH_4^+$-N | IN | Fe | Cu | Pb | Zn |
|---|---|---|---|---|---|---|---|---|
| | | | | Dry deposition flux | | | | |
| Case 1[a] | 9600± 4300 | 87±53 | 28±16 | 114±64 | 650±340 | 2±1 | 0.3±0.2 | 6±3 |
| Case 2 [a] | 18000± 11,000 | 13±18 | 8±5 | 21±22 | 1300±1000 | 3±2 | 0.08±0.04 | 4±1 |
| Case 3 [a] | 29,000± 31,000 | 26±6 | 17±8 | 42±12 | 2100±2200 | 6±1 | 0.20±0.02 | 5±3 |
| Non-dust | 2800± 700 | 48±33 | 15±8 | 63±39 | 190±110 | 1±1 | 0.09±0.1 | 5±4 |

[a] The characterizations of IN concentrations and sample information of the Cases are provided in Table 3.

[Figure]

**Table S3.** Dry deposition flux of aerosol particles (mg/m$^2$/month), particulate inorganic nitrogen (mg N/m$^2$/month) and some toxic trace m...

non-dust days

| Case | Sample number | Particles (TSP) | | NO$_3^-$-N | | NH$_4^+$-N | | IN | | Fe | | Cu | | P |
|---|---|---|---|---|---|---|---|---|---|---|---|---|---|---|
| | | Flux | Ratio[a] | Flux | Ratio[a] | Flux | Ratio[a] | Flux | Ratio[a] | Flux | Ratio[a] | Flux | Ratio[a] | F |
| Case 1 | 20080301 | 7680 | 2.7 | 78.4 | 1.6 | 23.8 | 1.6 | 102.2 | 1.6 | 509 | 2.7 | 2.1 | 1.7 | 0. |
| | 20080315 | 5207 | 1.9 | 62.1 | 1.3 | 47.7 | 3.2 | 109.8 | 1.8 | 295 | 1.6 | 1.5 | 1.2 | 0. |
| | 20090316 | 9254 | 3.3 | 54.6 | 1.1 | 13.3 | 0.9 | 67.9 | 1.1 | 483 | 2.6 | 1.2 | 1.0 | 0. |
| | 20100321 | 8049 | 2.9 | 67.7 | 1.4 | 19.2 | 1.3 | 86.9 | 1.4 | 588 | 3.1 | 1.2 | 1.0 | 0. |
| | 20110415 | 17782 | 6.3 | 193.6 | 4.1 | 47.6 | 3.2 | 241.1 | 3.9 | 1260 | 6.7 | 4.0 | 3.2 | 0. |
| | 20110502 | 9887 | 3.5 | 63.8 | 1.3 | 16.9 | 1.1 | 80.7 | 1.3 | 789 | 4.2 | 1.6 | 1.3 | 0. |
| | **Average** | 9643 | 3.4 | 86.7 | 1.8 | 28.1 | 1.8 | 114.8 | 1.8 | 654 | 3.5 | 1.9 | 1.6 | 0. |
| Case 2 | 20080425 | 11356 | 4.0 | 5.2 | 0.1 | 5.1 | 0.34 | 10.3 | 0.2 | 424 | 2.2 | 1.7 | 1.4 | 0. |
| | 20080528 | 31391 | 11.2 | 1.8 | 0.04 | 4.1 | 0.27 | 5.9 | 0.1 | 2631 | 13.9 | 6.0 | 4.9 | 0. |
| | 20080529 | 28053 | 10.0 | 0.2 | 0.004 | 7.3 | 0.48 | 7.4 | 0.1 | 2020 | 10.7 | 2.0 | 1.6 | 0. |
| | 20110319 | 12682 | 4.5 | 42.5 | 0.9 | 14.8 | 0.98 | 57.3 | 0.9 | 847 | 4.5 | 1.1 | 0.9 | 0. |
| | 20110501 | 6340 | 2.3 | 14.3 | 0.3 | 8.5 | 0.56 | 22.8 | 0.4 | 454 | 2.4 | 2.0 | 1.6 | 0. |
| | **Average** | 17964 | 6.4 | 12.8 | 0.27 | 8.0 | 0.53 | 20.7 | 0.33 | 1275 | 6.7 | 2.6 | 2.1 | 0. |
| Case 3 | 20100315 | 12174 | 4.3 | 32.2 | 0.7 | 23.6 | 1.6 | 55.8 | 0.9 | 666 | 3.5 | 6.8 | 5.4 | 0. |
| | 20100320 | 65267 | 23.3 | 24.5 | 0.5 | 7.9 | 0.5 | 32.4 | 0.5 | 4675 | 24.7 | 5.0 | 4.0 | 0. |
| | 20110418 | 10695 | 3.8 | 19.9 | 0.4 | 17.9 | 1.2 | 37.9 | 0.6 | 951 | 5.0 | 6.1 | 4.9 | 0. |
| | **Average** | 29379 | 10 | 25.6 | 0.54 | 16.5 | 1.1 | 42.0 | 0.67 | 2097 | 11 | 5.9 | 4.8 | 0. |
| ND | HDT080316 | 2840 | | 39.7 | | 13.3 | | 52.9 | | 193 | | | 0.6 | 0. |
| | HDT080424 | 2851 | | 102.6 | | 17.9 | | 120.5 | | 199 | | | 0.8 | 0. |
| | HDT080522 | 2705 | | 91.7 | | 27.6 | | 119.3 | | 73 | | | 0.9 | 0. |
| | HDT090306 | 1596 | | 13.2 | | 6.8 | | 20.0 | | 110 | | | 0.9 | 0. |
| | HDT100324 | 3992 | | 27.3 | | 4.6 | | 31.9 | | 449 | | | 4.2 | 0. |
| | HDT110308 | 2573 | | 43.6 | | 22.8 | | 66.3 | | 135 | | | | 0. |
| | HDT110416 | 3236 | | 18.1 | | 8.6 | | 26.7 | | 198 | | | | 0. |
| | HDT110523 | 2658 | | 44.4 | | 18.7 | | 63.1 | | 156 | | | 1.2 | 0. |
| | **Average** | 2806 | | 47.6 | | 15.0 | | 62.6 | | 189 | | | 1.2 | 0. |

---

## Author Response (AR1)

Referee #2

General comments: This paper attempts to study the impact of spring time dust storms in the Chinese deserts on the atmospheric concentration and deposition of inorganic nitrogen (IN) in the coastal Yellow Sea location. The study uses 4 years of particulate matter measurements and their chemical composition to understand how different types of dust storms can affect the abundance of inorganic nitrogen and calculate dry deposition of IN to the coastal Yellow Sea. This type of work is relevant since atmospheric deposition of nutrients and its implications is not a well understood topic and can be important for the regions that receive high atmospheric input like the Yellow Sea. However, the authors have not made best use of the data. There needs to be significant improvement in data interpretation and much more analysis needs to be done to support the main results. I would thus recommend a resubmission and see if there is any substantial improvement in the manuscript. My main comments are given below.

**Reply:** Thank you for the suggestion. We have reanalyzed the data and supplemented the modeling results using a 3-D air quality model to support the main results in the revised version. We have added "Theoretical analysis of three categories" and "The impact of dust on nitrogen dry deposition flux under anthropogenic activity" to the discussion section. Some figures have been redrawn, and some references have been updated. The data interpretation and the writing have also been improved. In brief, we have made substantial improvements to our manuscript.

**Q1.** First of all, the manuscript needs an overall improvement in writing. Not sufficient care has been given to the details and there are parts which are difficult to follow. I will provide some examples later, but there are many such cases of improper and awkward sentence constructions. Second, on what basis are these dusty and non-dusty days decided? More information are needed to show whether the dust storms originating from the deserts are actually passing over the measurement site and it is not the locally produced soil dust in cases of the days with low TSP values. Satellite aerosol products and meteorological data can be used to support this. Again later Al concentration is used to identify dust weather (lines 171-172). Please provide a clear definition of dust days and maintain that throughout the manuscript.

**Reply:** We have revised all sections of this manuscript, especially the sections "Experimental Method" and "Results and Discussion", to provide detailed methods and a logical analysis of the data. The discussion was revised substantially. Additionally, the writing has also been improved by a native English speaker.

In this study, a dust event was defined according the definition adopted in regulations of surface meteorological observation in China (CMA, 2004; Wang et al., 2008) and was identified based on the meteorological record information from Meteorological Information Comprehensive Analysis and Process System (MICAPS) of China Meteorological Administration. The sampling information is listed in the Table 1. Based on the forecast, we also collected aerosol particle samples immediately before or after the dust event for comparison. The post-dust event samples were collected under clear and sunny weather conditions and at low mass concentrations of $PM_{10}$.

All air mass trajectories showed that dust storms originated from deserts or Gobi and passed over the measurement site. More information on the dust events has been added to Tables 1 and 4. According to the suggestion, we tried to obtain aerosol particle concentrations using satellite aerosol products; however, the daily satellite data were incomplete. Therefore, we have included modeled dust concentrations over East Asia based on the CFORS model by Uno et al. (2003) (http://www-cfors.nies.go.jp/) and the PM10 modeling results for each dust event using a 3-D air quality model to support our analysis in the revision. The model results for each event are shown in Fig. 2 and Fig.S3. Based on Fig. 2, we found that the Sample 20110418 was mainly affected by local blowing dust. To focus on the impact of long-range transported dust on aerosol samples in downwind areas, this sample 20110415 was not included in the analysis and discussion in later sections.

As the reviewer suggested, the samples can be affected by local soil dust. Therefore, we used the criterion method of total Al concentration in TSP samples proposed by Hsu et al. (2008) to confirm that our samples originated from the sandy sources, as proposed in the previous version of this manuscript. We have added the distribution of hourly dust concentrations modeled over East Asia by the CFORS model for dust events in the revised version. The model results show the dust occurrence and transport patterns, thereby providing much more information than the Al criteria method. Therefore, we revised this section and replaced the Al criteria method with the model results in the revised manuscript.

Table 1. Sampling information for the aerosols samples collected at the Baguanshan site in the coastal region of the Yellow Sea.

| Sampling Year | Sample category | Sampling number | Sampling time | Weather characteristics |
|---|---|---|---|---|
| 2008 | Samples on dust days | 20080301 | From 13:22 a.m. to 17:22 p.m. on Mar. 1st | Floating dust[a] |
| | | 20080315 | From 13:21 a.m. to 17:21 p.m. on Mar. 15 th | Floating dust |
| | | 20080425 | From 13:14 a.m. to 17:14 p.m. on Apr. 25th | Floating dust |
| | | 20080528 | From 11:38 a.m. to 15:38 p.m. on May 28th | Floating dust |
| | | 20080529 | From 10:15 a.m.to 12:15 p. m. on May 29th[b] | Floating dust |
| | Samples on non-dust days | 20080316 | From 13:00 a.m. to 17:00 p.m. on Mar. 16th | Sunny day |
| | | 20080424 | From 13:00 a.m. to 17:00 p.m. on Apr. 24th | Sunny day |
| | | 20080522 | From 13:00 a.m. to 17:00 p.m. on May 22nd | Cloudy day with mist |
| 2009 | Samples on dust days | 20090316 | From 8:25 a.m. to 12:25 p.m. on Mar. 16th | Floating dust |
| | Samples on non-dust days | 20090306 | From 13:00 a.m.to 17:00 p.m. on Mar. 6th | Sunny day |

| | | 20100315 | From 11:30 a.m.to 15:30 p.m. on Mar. 16th | Mist after Floating dust |
|---|---|---|---|---|
| 2010 | Samples on dust days | 20100320 | From 10:30 a.m. to 14:30 p.m. on Mar. 20th | Floating dust |
| | | 20100321 | From 10:30 a.m. to 14:30 p.m. on Mar. 21st | Floating dust |
| | Samples on non-dust days | 20100324 | From 11:30 a.m. to 15:30 p.m. on Mar. 24th | Sunny day |
| 2011 | Samples on dust days | 20110319 | From 12:00 a.m. to 16:00 p.m. on Mar. 19th | Floating dust |
| | | 20110415 | From 12:00 a.m. to 16:00 p.m. on Apr. 15th | Floating dust |
| | | 20110418 | From 12:25 a.m. to 16:25 p.m. on Apr. 18th | Floating dust[c] |
| | | 20110501 | From 12:10 a.m. to 16:10 p.m. on May 1st | Floating dust |
| | | 20110502 | From 16:00 a.m. to 20:00 p.m. on May 2nd | Floating dust |
| | Samples on non-dust days | 20110308 | From 12:00 a.m. to 16:00 p.m. on Mar. 8th | Sunny day |
| | | 20110416 | From 12:00 a.m. to 16:00 p.m. on Apr. 16th | Sunny day |
| | | 20110523 | From 12:00 a.m. to 16:00 p.m. on May 23rd | Sunny day |

[a]Note that one exterior dust sample was collected on March 1 when no dust was recorded by the MICAPS. However, the MICAPS information indeed showed dust events in China on March 1. The modeled spatial distribution of the $PM_{10}$ mass concentration for this dust event on March 1 implies that the sample should be classified as a dust sample. The supporting figure is Fig. S1.

[b] The sampling duration was reduced to only 2 hrs because of extremely high particle loads.

[c]Note that one exterior dust sample was collected on April 18 when no dust was recorded by the MICAPS. However, blowing dust occurred and was recorded on April 17 by the Sand-dust weather almanac 2011 (CMA, 2011). The modeled spatial distribution of the $PM_{10}$ mass concentration for this dust event on April 18 implies that the sample should be classified as a dust sample. The supporting figure is Fig. S2

**Table 4.**Average concentrations of inorganic nitrogen, TSP, NOx, Relative Humidity (RH) and air temperature for each Category in aerosol samples in Qingdao

| | Sample number | TSP µg·m⁻³ | NO₃⁻ µg·m⁻³ | NH₄⁺ µg·m⁻³ | RH % | T °C | NOx µg·m⁻³ | Summary |
|---|---|---|---|---|---|---|---|---|
| Category 1 | 20080301 | 527 | 20.5 | 12.7 | 57 | 7.0 | 36 | IN concentrationon dust days higher than that on ND days |
| | 20080315 | 410 | 19. 5 | 29.9 | 62 | 11.0 | 59 | |
| | 20090316 | 688 | 15.9 | 17.2 | 27 | 16.0 | 75 | |

| | | | | | | | | |
|---|---|---|---|---|---|---|---|---|
| | 20100321 | 519 | 16.5 | 9.4 | 51 | 8.8 | 76 | |
| | 20110502 | 810 | 21.0 | 11.0 | 49 | 17.7 | 62 | |
| Category 2 | 20080425 | 622 | 6.8 | 2.0 | 30 | 18.0 | 40 | IN concentrationon dust days lower than that on ND days |
| | 20080528 | 2579 | 9.2 | 2.7 | 17 | 27.0 | 34 | |
| | 20080529 | 2314 | 17.5 | 4.8 | 60 | 20.0 | 29 | |
| | 20110319 | 939 | 12.3 | 9.4 | 16 | 12.6 | 93 | |
| | 20110501 | 502 | 4.5 | 5.3 | 23 | 21.6 | 66 | |
| Category 3 | 20100315 | 501 | 5.4 | 4.3 | 30 | 7.2 | 73 | $NO_3^-$ concentration on dust days lower than that on ND days;$NH_4^+$close to that on ND days |
| | 20100320 | 3857 | 5.5 | 3.4 | 35 | 10.6 | 92 | |
| | 20110418 | 558 | 3.8 | 6.6 | 33 | 12.6 | 47 | |
| Non-dust[a] | 20080316 | 225 | 12.6 | 8.4 | 28 | 11.0 | 60 | |
| | 20080424 | 137 | 21.7 | 7.2 | 49 | 18.0 | 53 | |
| | 20080522 | 206 | 27.4 | 16.6 | 78 | 20.0 | 60 | |
| | 20090306 | 94 | 2.9 | 3.0 | 29 | 7.00 | 51 | |
| | 20100324 | 275 | 7.2 | 2.4 | 23 | 9.0 | 82 | |
| | 20110308 | 194 | 13.0 | 13.1 | 20 | 11.5 | 111 | |
| | 20110416 | 252 | 5.6 | 5.4 | 26 | 14.1 | 55 | |
| | 20110523 | 224 | 15.2 | 10.2 | 42 | 20.6 | 49 | |

[a] The corresponding ND day for each dust event see Table 1.

[Figure]

Figure 2. Modeled dust concentrations over East Asia by CFORS model during each dust sampling day from 2008 to 2011 (http://www-cfors.nies.go.jp/). (The figures show the modeled dust concentration in the middle of each sampling duration). No data are available for Mar. 19, 2011, because of the earthquake in Japan. Hourly PM10 concentrations were modeled by the WRF-CMAQ model for each sampling day, and the results are shown in Fig. S3.

[Figure]

FigureS3. Hourly PM10 concentration in China modeled by the WRF-CMA model for Sample 20110418  (The figure shows the modeled PM10 concentration at the middle of the sampling period).

CMA: Regulations of Surface Meteorological Observation, China Meteorological Press, Beijing, 154–156, 2004.

CMA: Sand-dust weather almanac 2011, China Meteorological Press, Beijing, 36-53, 2013.

Hsu, S. C., Liu, S. C., Huang, Y. T., Lung, S. C. C., Tsai, F., Tu, J. Y., and Kao, S. J.: A criterion for identifying Asian dust events based on Al concentration data collected from northern Taiwan between 2002 and early 2007, J. Geophys. Res-Atmos., 113, 1044-1044, 2008.

Uno, I., Carmichael, G. R., Streets, D. G., Tang, Y., Yienger, J. J., Satake, S., Wang, Z., Woo, J. H., Guttikunda, S., Uematsu, M., Matsumoto, K., Tanimoto, H., Yoshioka, K., and Iida, T.: Regional chemical weather forecasting system CFORS: Model descriptions and analysis of surface observations at Japanese island stations during the ACE-Asia experiment, J. Geophys. Res-Atmos., 108, 1147-1164, 2003.

Wang, Y. Q., Zhang, X. Y., Gong, S. L., Zhou, C. H., Hu, X. Q., Liu, H. L., Niu, T., and Yang, Y. Q.: Surface observation of sand and dust storm in East Asia and its application in CUACE/Dust, Atmos. Chem. Phys., 8, 545-553, 2008.

Q2. The authors have divided dusty days into 3 categories based on inorganic nitrogen (IN) concentrations relative to the non dust (ND) days. They are reporting that in some cases IN concentrations are more than the ND days, in some cases IN concentration are less than the ND days and in some cases nitrate concentration on dust days are less than ND days while ammonium concentrations on dust days are more compared to ND days. Next, the authors are reporting that sand from Duolun (which is a source for dust storms affecting the Yellow Sea region) is poor in nitrate and ammonium content. First of all, trajectory analysis does not seem to point that coastal Yellow Sea region is only affected by dust originating from Duolun region. There are many other dust sources over which the trajectories are passing. And if Duolun is deficient in IN you need to provide a detailed discussion on the possible sources of IN in dust sampled from the coastal region of the Yellow Sea and about the mixing of anthropogenic aerosols with the dusty air mass. The authors have related the 3 cases of IN in dust samples to the wind speed concluding that when wind speed is less IN concentration increased and vice versa . I am not clear how this conclusion is arrived at, especially, with only 5 cases for IN<ND. Forexample, in Table 5 sample number 110501 NO3- and NH4+ concentrations are lowand the wind speed is also low. Again, sample 080315 has higher NO3- and NH4+concentrations at higher wind speed. What is the rationale of using wind speed of 40.5km/h in this study and how is this threshold derived? I would suggest the authors togroup the trajectories according to dust and non dust days and also according to the levels of nitrogen and see which of these trajectories are passing over highly populated regions.

**Reply:** First of all, East Asia has three major dust aerosol sources: the western China sources, including the Taklimakan Desert and surrounding areas; the Mongolia sources, including the desert and semi-desert areas in southern Mongolia; and the northern China sources, including the BadainJaran Desert, the Tengger Desert, the Ulan Buh Desert and the Hunshandake Desert (Fig. SR1). Duolun and Zhurihe belong to the Hunshandake Desert in northern China. According to studies, 52%-71% of the dust storms that affect the Yellow Sea are from this sand source (Zhang and Gao, 2007; Gao et al., 2010). For our 14 dust samples, 13 events were from northern sources, and the remaining one was from northern sources but was transported to the northeast before being transported to the measurement site (the 72 h trajectory only showed transport from the northeast). Therefore, we collected sand samples at the Zhurihe site and used reported aerosol values from Duolun to characterize the aerosol particles in the source region. As the reviewer suggested, the coastal Yellow Sea region is affected by dust originating not only from the Duolun region but also from other regions. We added the nitrate and ammonium contents in sand samples collected from other sand sources in China. We also added the reported concentrations of nitrate and ammonium in aerosol particles on dust days at or close to the dust source region in Table 5 in the revised manuscript. All these data verified that the mass concentrations of nitrate and ammonium relative to the total mass of sand particles were very low, i.e., less than 81μg/g. We have added the modeling results of $NO_x$ and $NH_3$ emissions using a 3-D air quality model and discussed the theoretical analysis of three categories in section 3.3 in the revised manuscript.

To characterize the transport paths of dust day samples more exactly, we reanalyzed the air mass trajectory of each sample at 1500 m altitude and redrew Fig. 5 to distinguish trajectories for samples collected on dust and non-dust days. According to the suggestion, we first we investigated whether these trajectories passed over highly populated regions. All trajectories involved transport from the source over highly populated regions and exhibited different transport times in these regions. We found that the concentrations of nitrate and ammonium depended on the transport distance over the sea, the mixed layer depth and the transport time over highly polluted regions. We have rewritten this section according to the suggestion to discuss the influence of transport paths on nitrate and ammonium contents in the revised manuscript.

[Figure]

Figure SR1. Map of the main source regions of sand and dust storms in China (from Wang et al., 2008).

[Figure]

Figure 5. The 72-h backward trajectories for non-dust (a) and dust (b) samples from 2008 to 2011 (the yellow domains in the maps represent the dust source regions in China.

**Table 5.** Comparison of nitrate and ammonium content in sand and aerosol particles on dust days or close to the dust source region(unit: µg/g)

| Sands sampled in dust source regions | | | Aerosols in or close to dust source region on dust days | | | Aerosols in the coastal region of the Yellow Sea | |
|---|---|---|---|---|---|---|---|
| Study region and data source | Relative concentration[a] | | Study region and data source | Relative concentration[a] | | | |
| | $NO_3^-$ | $NH_4^+$ | | $NO_3^-$ | $NH_4^+$ | $NO_3^-$ | $NH_4^+$ |
| Zhurihe (This study) | 25.46± 22.87 | 4.21± 1.03 | Duolun (Cui, 2009) | 1200 | 900 | Non-dust: 28,200±24,819 | Non-dust: 24,063±21,515 |
| AlxaLeft Banner, Inner Mongolia (NiuandZhang, 2000) | 62.1±7.4 | 79.1±1.1 | AlxaRightBanner, Inner Mongolia (NiuandZhang, 2000) | 1975[b] | 4091[b] | Category 1: 34,892±9570 | Category 1: 22,571±7,016 |
| Yanchi, Ningxia (NiuandZhang, 2000) | 46.4±2.2 | 80.9±1.3 | Hinterland of theTaklimakan Desert, Xinjiang (Dai et al., 2016) | 142-233 | 2-15 | Category 2: 5,542±5,117 | Category 2: 4,758±5,698 |
| | | | Average of SonidYouqi, Huade (Inner Mongolia), Zhangbei (Hebei) (Mori et al., 2003) | 253 | 710 | Category 3: 6,359±4,697 | Category 3: 7,059±5,591 |
| | | | Yulin, the north edge of Loess Plateau (Wang et al., 2011) | 216.4 | 80.6 | | |
| | | | Golmud, Qinghai (Sheng et al., 2016) | 892.9 | -[c] | | |
| | | | Hohhot, Inner Mongolia (Yang and Wang, 1995) | 588.1 | No data | | |

[a]Relative concentration ofnitrate and ammonium per aerosol particle mass

[b]Samples collected on a floating dust day (Horizontal visibility less than 10000 m and very low wind speed)

[c] The ammonium concentration was lower than the detection limit of the analytical instrument.

Cui, W. L.: Chemical transformation of dust components and mixing mechanisms of dust with pollution aerosols during the long range transport from north to south China, M.S. thesis, Department of Environmental Science and Engineering, FudanUniversity, China, 38 pp., 2009.

Dai, Y. J.: Vertical distribution of characteristics of dust aerosols in the near-surface in hinterland of Taklimakan Desert, M.S. thesis,

College of Resources and Environmental Science, Xinjiang University, China, 26 pp., 2016.

Mori, I., Nishikawa, M., Tanimura, T., and Quan, H.: Change in size distribution and chemical composition of kosa (Asian dust) aerosol during long-range transport, Atmos. Environ., 37, 4253-4263, 2003.

Niu, S. J. and Zhang, C. C.: Researches on Sand Aerosol Chemical Composition and Enrichment Factor in the Spring at Helan Mountain Area, Journal of Desert Research, 20, 264-268, 2000.

Sheng, Y., Yang, S., Han, Y., Zheng, Q., and Fang, X.: The concentrations and sources of nitrate in aerosol over Dolmud, Qinghai, China, Journal of Desert Research, 36, 792-797, 2016.

Wang, Q. Z., Zhuang, G. S., Li, J., Huang, K., Zhang, R., Jiang, Y. L., Lin, Y. F., and Fu, J. S.: Mixing of dust with pollution on the transport path of Asian dust — Revealed from the aerosol over Yulin, the north edge of Loess Plateau, Sci. Total. Environ., 409, 573–581, 2011.

Yang, D. Z., Wang, C., Wen, Y. P., Yu, X. L., and Xiu, X. B.: Ananalysis of Two Sand Storms In Spring 1990, Quarterly Journal of Applied Meteorology, 6, 18-26, 1995.

Specific comments:

Q3. L11-13: Reconstruct the sentence.

Reply:These sentences have been rewritten as "Asian dust has been reported to transport anthropogenic reactive nitrogen from source areas to the oceans. In this study, we attempted to characterize the $NH_4^+$ and $NO_3^-$ in atmospheric particles collected at a coastal site in northern China during spring dust events from 2008 to 2011."

Q4. L20-25: This is not conclusive from the discussions that follow. Statements like "storms were weak or slow moving" and "rapid transport in a strong dust storm" are not supported by proper analysis of the storm characteristics.

Reply: We have written the conclusion.

Q6. L35-36: This is contradicting L32-33.

Reply: We apologize for the confusion in the revision. We will revise the abstract sentence into "The dust deposition was an uncertain source of nitrogen for the ocean".

Q7. L40: These references point to anthropogenic contribution to atmospheric nitrogen deposition.

Reply: We apologize for the confusion in the revision. We have revised the abstract.

Q8. L62-64: The first part of the sentence seems to contradict the last part.

Reply: To avoid confusion, we have revised the sentence to "Although increased concentrations of $NO_3^-$ and $NH_4^+$ in aerosol particles were observed on dust storm days in northern China relative to those non-dust days prior to the dust storm events, Zhang et al. (2010) also found that the concentrations of the two species were associated with the intensity of the dust storm, i.e., the stronger dust storms corresponded to the smaller increases. In other words, lower $NO_3^-$ and $NH_4^+$ concentrations occurred during strong dust storm events than during weak dust events (Zhang et al., 2010)."

Q9. L79-81: Meaning is not clear.

Reply: We have deleted this sentence in the revised manuscript because this reference is not closely related to Asian dust.

Q10. L102: Zhurihe in Hunshandake Desert. Later you are using Duolun region which isnot introduced in Section 2. It will be difficult for readers to follow if they are not veryfamiliar with this region.

Reply: Thank you for the suggestion. The data at Duolun was adapted from the reference (Cui et al., 2009b). We have added the citation in the text of section 3.2. According to the former suggestion inQ2, we added new references for the dust components in source region to Table 4 in the revised manuscript (See reply to Q2). Additionally, we introduce Duolun and other study regions using references in Table 4.

Cui W. L.: Chemical transformation of dust components and mixing mechanisms of dust with pollution aerosols during the long range transport from north to south China,M.S. thesis, Department of Environmental Science and Engineering, Fudan University, 32-38 pp., 2009.

Q11. L135: Explain how the PMF model works.

Reply: We have added the PMF work principle in section 2.

Q12. L139: Explain how Williams' model works and cite the original paper.

Reply: We have added the work principle of the Williams' model and cited the original paper.

Q13. L140. Expand U10. In general, all the abbreviations used should be clearly defined.

Reply: We have added definitions for all the abbreviations in our revised manuscript.

Q14. L143: Meteorological data not "climatic" data.

Reply: The term "climatic" has been changed to "meteorological data".

Q15. L146: This heading does not reflect the text content.

Reply: The heading has been revised to "Other data sources and statistical analysis ".

Q16. L152: Please provide details on the MICAPS information used.

Reply: We have supplemented the details on the MICAPS in section 2.1.

Q17. L164: What is the average TSP concentration on dust days?

Reply: We have added the average TSP concentration (1140.3 $\mu g \cdot m^{-3}$) on dust days in the revised manuscript.

Q18. L167: Please provide a brief description on how EF is calculated and what is the significance of using this method.

Reply: We have provided a brief description of EF and the significance of using EF in section 2.3.

Q19. L171-173: The authors were using MICAPS information of dust storm (which has to be explained) and now are relying on Al levels to define AD events. How are these two definitions consistent?

Reply: First, we defined dust days according to MICAPS information in Qingdao. However, the intensity of a dust event decreases gradually during the long-range transport, and high PM10 episodes of anthropogenic origin (dust is regarded as of natural origin) can be erroneously judged as a dust event in downwind regions. Hsu et al. (2008) proposed the "geometric mean $\times2$ GSD" of Al concentrations in order to identify major Asian dust (AD) events in downwind regions and obtained good results. Our focus was the impact of long-range transported dust on aerosol samples in downwind area; therefore, we verified the long-range AD using reported GSD criterion in former manuscript. Just as discussed above, we have revised this section and replaced the Al criteria method using the model results in the revised manuscript.

Hsu, S. C., Liu, S. C., Huang, Y. T., Lung, S. C. C., Tsai, F., Tu, J. Y., and Kao, S. J.: A criterion for identifying Asian dust events based on Al concentration data collected from northern Taiwan between 2002 and early 2007, Journal of Geophysical Research Atmospheres, 113, 1044-1044, 2008.

Q20. L178-179: This has to be explained with respect to the EF of the anthropogenic elements.

Reply: We have added the calculation and significance of EF in section 2.3 in revised manuscript.

Q21. L214-216: You need to examine the dust sources, transport pathways (if passing over heavy populated regions), the height at which dust is transported together with dust concentration on a case by case basis to conclude these lines. This is very important for the entire paper. How do you decide "stronger a e "stronger a dust storm"?

Reply: Similar to the reply for Q2, we have revised the section according to the suggestion. In addition, the modeling dust concentration, and transport altitude were supplemented.

Q22. L225-230: The average for Case 1 is 700 μg/m3 with values lying well above the average as is evident from Table 5. Again, there are TSP values in Case 2 in Table 5 which are lower than average of Case 1.

Reply: We have rewritten this section. By comparing the relative values of $NH_4^+$ and $NO_3^-$ in dust day samples to those in comparison samples, the characteristics of dust day samples were discussed, and a theoretical analysis of the three categories was added.

Q23. L230-231: Again, what is your definition of "strong dust storm"? Without sufficient analysis I am not sure how "dust might be transported quickly" is factoring here. This entire section has to be revisited.

Reply: According to the suggestion inQ2, we have added modeled dust concentrations over East Asia based on the CFORS model for dust event samples (http://www-cfors.nies.go.jp/) and rewritten the entire section considering whether trajectories passed over highly populated regions.

Q24. L242: How is the value 40.5 km/h arrived at? Is it estimated at the dust source region? How many dust storms were studied to derive this value? More explanation is needed.

Reply: According to Asian dust observations in China and Mongolia(He et al., 2008; Li et al., 2006; Natsagdorjaet al., 2003; Zhan et al., 2009), we estimated an average wind speed threshold as 40.5±9.9 km/h during dust storms. However, according to the suggestion inQ2, we have revised the entire section and discussed the influence of transport on the concentrations of nitrate and ammonium in dust day samples.

He, Q., Wei, W., Li, X., Ali, M., and Li, S.: Profile Characteristics of Wind Velocity, Temperature and Humidity in the Surface Layer during a Sandstorm Passing Taklimakan Desert Hinterland, Desert and Oasis Meteorology, 2, 6-11, 2008.

Li, N., Du, Z. X., Liu, Z. Y., Yang, H. J., Wu, J. D., and Lei, Y.: Change of wind speed and soil moisture during occurrence of dust storms, Journal of Natural Disasters, 15, 28-32, 2006.

Natsagdorj, L., Jugder, D., and Chung, Y. S.: Analysis of dust storms observed in Mongolia during1937–1999, Atmos. Environ., 37, 1401–1411, 2003.

Zhan, K. J., Zhao, M., Fang, E., Yang, Z. H., Zhang, Y. C., Guo, S. J., Zhang, J. C., Wang, Q. Q., and Wang, D. Z.: The wind speed characteristics of near-surface vertical gradient of 50m in sandstorm process in 2006, Journal of Arid Land Resources & Environment, 23, 100-105, 2009.

Q25. L250-252 needs explanation. Once the dust storms are properly categorized and the pathways are determined the interpretation of Table 5 might change

Reply: We have reanalyzed the trajectories and rewritten this section.

Q26. L250: Less than 40.5 km/h not 42.4 km/h.
Reply: We have revised this section.

Q27. L281-283: These statements need to be backed by more analysis of the dust events on a case by case basis.

Reply: We cannot discuss the contributions of dust to aerosols for each dust case because of the limited number of samples. The limited number of samples will result in an unrealistic source apportionment. Thus, we have rewritten this sentence.

Q28. L301-311: The text is very confusing. It is very difficult to follow when the authors are referring to TSP and when they are referring to IN or nitrate or ammonium.

Reply: We have rewritten these sentences. Additionally, we added the IN (now $N_{NH4++NO3}$- in revised manuscript) definition in section 3.6 and the flux of $N_{NH4++NO3}$- in Table 8 and revised several mistakes.

**Table 8.** Dry deposition of TSP (mg/m$^2$/month), particulate inorganic nitrogen (mg N/m$^2$/month) and some toxic trace metals (mg/m$^2$/month) on dust and non-dust days.

| | Dry deposition flux | | | | | | | |
| --- | --- | --- | --- | --- | --- | --- | --- | --- |
| | TSP | NO$_3^-$-N | NH$_4^+$-N | N$_{NH4++NO3}$- | Fe | Cu | Pb | Zn |
| Category 1[a] | 8,000±1800 | 65±9 | 24±14 | 90±17 | 533±179 | 2±0.3 | 0.3±0.3 | 6±2 |
| Category 2[a] | 18000±11,000 | 13±18 | 8±4 | 21±22 | 1300±1000 | 3±2 | 0.08±0.04 | 4±1 |
| Category 3[a] | 29,000±31,000 | 26±6 | 17±8 | 42±12 | 2100±2200 | 6±1 | 0.20±0.02 | 5±3 |
| Non-dust | 2,800±700 | 48±33 | 15±8 | 63±39 | 190±110 | 1±1 | 0.09±0.1 | 5±4 |

[a] The characterization of N$_{NH4++NO3}$- concentration and sample information of the Category see in Table 3.

Technical corrections:

Q29. L260: Colors used in Figure 4 are not clear. Please indicate the dust source regions on the map.

Reply: We have redrawn Fig. 4 (now Fig. 5 in the revised version) to distinguish the trajectory of each sample collected on dust and non-dust days. The dust source regions in China are also now indicated in this figure.

[Figure]

Figure 5. The 72-h backward trajectories for non-dust (a) and dust (b) samples from 2008 to 2011 (the yellow domains in the maps represent the dust source regions in China).

Q30. L343-442: Not proper attention has been given to the References and needs to be corrected.

Reply: We have updated and corrected the reference citations and format.

[revised manuscript text omitted]

For example, a few studies have showedn that Some researchershave found that inorganic nitrogen species in aerosols havehigh concentrationsduring Asian dust events. Tthe concentrations of atmospheric particulate $NO_3^-$ and $NH_4^+$ on dust storm days were 2-5 times higher larger than those on those non-dust storm (ND) days prior to the events in Beijing (Liu et al., 2014; Liu and Bei, 2016). Xu et al. (2014) also reported found that concentrations of particulate $SO_4^{2-}$ and $NO_3^-$ were simultaneously increased in during dust storm events on inalong the northern boundary of the Tibetan Plateau because of the enriched dust including more acidic species or anthropogenic aerosols. Compared to those on

ND days, higher concentrations of $NO_3^-$ and $NH_4^+$ in aerosol particles were observed on dust storm days in northern China, and $NO_3^-$ and $NH_4^+$showed lower concentrations during strong dust storm events than during weak dust events (Zhang et al., 2010). Fitzgerald et al. (2015) found that almost nearly all Asian dust observed in Korea containscontained considerable amounts of nitrate and proposed that because pollution plumes mix with dust the dust from the Gobi and Taklamakan Deserts probably mixed and reacted with anthropogenic air pollutants during the transport and are transported over the Asian continent. Although increased concentrations of $NO_3^-$ and $NH_4^+$ in aerosol particles were observed on dust storm days in the northern China relative to Compared to those on Ndnon-dust days prior to the dust storm events;.

Zhang et al. (2010a) also found that the concentrations of the two species were associated with the intensity of the dust storm, i.e., the stronger dust storms corresponded to the smaller increases. In other words, lower $NO_3^-$ and $NH_4^+$ concentrations occurred during strong dust storm events than during weak dust events (Zhang et al., 2010a).

higher concentrations of $NO_3^-$ and $NH_4^+$ in aerosol particles were observed on dust storm days in northern China, and $NO_3^-$ and $NH_4^+$showed lower concentrations during strong dust storm events than during weak dust events (Zhang et al., 2010).

On the other hand, However, some studies reporetdreportedobserved that the reverse resultthe

. For example,

At Yulin, a rural site near the Asian dust source region, the concentration of $NO_3^-$ in atmospheric aerosols on dust days was significantly significantly lower in comparison to the concentration measured immediately before or after the event, as a result of the dilution effect (Wang et al., 2016). Even in Shanghai, a mega city being located at a few thousands kilometers from dust source zones in China, t  concentrations of $NO_3^-$ and $NH_4^+$ were notably lower in the observed dust plumes than in a polluted air parcel immediately  prior to the dust events (Wang et al., 2013).

. Li et al. (2014) also found that the concentrations of nitrate and ammonium   decreased on dust storm days, with a decreasing ratio of the total soluble inorganic ions to $PM_{2.5}$ in the Yellow River Delta, China. When dust was rapidly transported from desert regions without passing through  major urban area and lingering over the Yellow Sea, the concentrations and size distributions of nitrate and ammonium had no significant variation in heavy Asian dust (AD) plumes (Kang et al., 2013).

The contradictory results highlight the importance of investigating the concentrations of ammonium and nitrate in atmospheric particles during dust events based on a larger database. In this study, a

we collected atmospheric aerosol particles during and prior to (or after, when no sample was collected prior to) dust events at a coastal site adjacent to the Yellow Sea  during the spring from 2008 to 2011 when  smaller   outbreak peak  of  dust storms occurred

We measured  concentrations of the inorganic nitrogen concentrationsin the aerosol samples as well as other components for facilitating analysis. We first characterized The the concentrations of inorganic nitrogen concentrations in various dust events relative to the concentrations in samples collected either prior to or after the events. We then conducted , source apportionment to quantify their sources. Finally, we calculated and deposition flux of atmospheric particulate inorganic nitrogen in during dust events and compared the results with the values in theliterature in order to update the flux values due todynamic changes inanthropogenic emissions and other factors.

**2 Experimental Materials and methods**

**2.1 Sampling**

As shown in Fig. 1 shows the sampling site, which is situated at the top of a coastal hill (Baguanshan)

in Qingdao in northern China (36° 6' N, 120° 19' E, 77 m above sea level)and is approximately 1.0 km from the Yellow Sea tothe east.Ahigh-volume air sampler (Model KC-1000, Qingdao Laoshan

Electronic Instrument Complex Co., Ltd.) was set up on the roof of an two-story office buildingto collect total suspended particle (TSP) sampleson quartz microfiber filtersshowed the sampling site, which is situated at the top of a coastal hill(Baguanshan) in the northern China (36° 6' N, 120° 19' E, 77

m above sea level)and approximately 1.0 km from the Yellow Sea in the east., total suspended particles (TSP) were collected atthe Baguanshan site in the coastal region of the Yellow Sea.The samples were collected with quartz microfiber filters (Whatman QM-A) at a flow rate of 1 m$^3$/min. Prior to the sampling, the filters were heated at 450°C for 4.5 h to remove organic compounds. Our sample collection strategy involved collecting dust samples representing long-range transported particles. We followed the definition of dust events adopted in the regulations of surface meteorological observations of China (CMA, 2004;Wang et al., 2008) and identified dust events based on the meteorological records (Weather Phenomenon) of Qingdao from Meteorological Information Comprehensive Analysis and Process System (MICAPS) of the China Meteorological Administration.Each dust sample was collected over4 hrs,and the sampling started only when the PM$_{10}$ and dust mass concentration available on the website (http://www-cfors.nies.go.jp/~cfors/;http://www.qepb.gov.cn/m2/) hadincreased greatly.

Thisapproach made the dust sample more representative relative to urban background. For dust events with durations of less than one day, only one sample was collected. For dust events with durationsgreater thanone day, a4-hr dust sample was collected once perday. Table 1 lists the sampling information. Based on the forecast, we also collected aerosol particle samples immediately before orafter the dust event for comparison. These comparison samples were further classified into sunny day samples, cloudy day samples and post-dust samples. The post-dust samples were collected under clear and sunny weather conditions and low PM$_{10}$mass concentrations.

usinga Ahigh-volume air sampler (Model KC-1000, Qingdao Laoshan Electronic Instrument Complex

Co., Ltd.)with a flow rate of 1 m$^3$/minwas set up on the roof of an two-story office buildingforcollecting total suspended particles (TSP)on quartz microfiber filters (Whatman QM-A)with a flow rate of 1 m³/min(36° 6' N, 120° 19' E, 77 m above sea level) approximately 1.0 km from the shore.Prior to the sampling. The the filters were heated at 450°C for 4.5 h to remove organic compounds. Moreover, the concentration of Alreferring to the total Al concentration in TSPsamples were used to confirm the division of dust or comparison samples according to the criterion "geometric mean×2GSD" proposed by Hsu et al. (2008).

Since Asian dust events at the sampling site mostlyoccurin the spring, we collected samples duringevery spring, i.e., from March to May,from2008-2011. A smaller peak inAsian dust was observed in eastern China in 2008-2011, which followeda larger peakin 2000-2003 duringthis century (Fig. 6).Overall, atotal of 14 sets of dust samples and 8 sets of comparison samples were collected for this study.

To facilitate the coastal sampling data analysis, tThe sand samples were collected at the Zhurihe site (42°22'N, 112°58'E) in the Hunshandake Desert, one of main Chinese sand deserts, in April 2012. After sSand samples were packed in clean plastic sample bags, the samples and were stored below -20°C before the transfer. An ice-box was used to store the samples which wereduring transported to the lab for chemical analysis.

[Figure]

**Figure 1.**Location of the aerosol and dust sampling site.

**2.2 Analysis**

The aerosol samples were allowed to achieve equilibrium in a air-conditioned chamber at a constant relative humidity and temperature for 24hrs balanced in a air-condition relative humidity- and temperature-controlled chamberat a constant relative humidity and temperature for 24-hrs until the particlebefore weights remained constanting. The mass concentrations of TSPwere calculated according to the particlemassesand the sampling volume.The sample membranes were then cut into several portions for analysis.

One portion of each aerosol sample and 0.1 g of parallel sandsample were was ultrasonically extracted with ultra-pure water in an ice water bath, and the concentration of inorganic water-soluble ions was determined via using ICS-3000 ion chromatography (Qi et al., 2011). The parallel sand samples collected at the Zhurihe sitefrom the Hunshandake Desert were analyzed using the same procedure. We refer to dissolved inorganic nitrogen (DIN),thesum of nitrate and ammonium, in the later discussion due to the very low concentration of nitrite in the samples.

One portion of each aerosol filter was cut into 60 cm$^2$ pieces and digested with $HNO_3$+$HClO_4$+$HF$

(5:2:2 in volume) at 160°C using an electric heating plate. A blank membrane was also analyzed using the same procedure to ensure analytical precision. Cu, Zn, Cr, Sc and Pb were measured by using an inductively coupled plasma-mass spectrometry (Thermo X Series 2), while Al, Ca, Fe, Na and Mg were detected by using an inductively coupled plasma-atomic emission spectroscopy (IRIS Intrepid II XSP).

The membrane blanks have been corrected for in the calculation of the.The membrane blanks have beeb deducted to calculate tThe metal concentrations were determined by calibrating the measured concentrations of samples using mfor analysis below.embrane blanks.

Table 1.Detection limits, precisions and recoveries of water-soluble ions and metal elements.

| Component | Measurement method | Detection limit ($\mu g \cdot L^{-1}$) | Precision (RSD%) | Recovery (%) |
|---|---|---|---|---|
| $NO_3^-$ | | 2.72 | 1.54 | 97 |
| $SO_4^{2-}$ | | 1.62 | 1.55 | 98 |
| $NH_4^+$ | IC | 0.4 | 1.10 | 97 |
| $Ca^{2+}$ | | 0.44 | 0.79 | 94 |
| Cu | ICP-MS (Xin et al., 2012) | 0.006 | 4.0 | 106 |
| Zn | | 0.009 | 2.5 | 102 |
| Cr | | 0.004 | 3.0 | 95 |
| Sc | | 0.002 | 2.4 | 97 |
| Pb | | 0.008 | 3.9 | 104 |

| | | | | |
|---|---|---|---|---|
| Al | ICP-AES (Lin et | 7.9 | 0.6 | 103 |
| Ca | al., 1998) | 5.0 | 1.2 | 99 |
| Fe | | 2.6 | 0.7 | 104 |
| Na | | 3.0 | 0.6 | 99 |
| Mg | | 0.6 | 0.6 | 105 |
| Hg | CVAFS | 0.0001 | 6.6 | 105 |
| As | CVAFS | 0.1 | 5.0 | 98 |

One portion of each aerosol sample was digested by adding with HNO$_3$ solution (10% HNO$_3$, 1.6 M) at 160°C for 20 min in a microwave digestion system (CEM, U.S.). Hg and As in sample extracts were analyzed following the U.S. Environmental Protection Agency method 1631E (U.S. EPA, 2002) using cold vapor atomic fluorescence spectrometry (CVAFS).

The detection limits, precisions and recoveries of water-soluble ions and metal elements are listed in Table 12.

**2.3 Computational modeling**

The enrichment factor of metal elements was given by

$$EFi = \frac{\left(X_i / X_{\text{Re}}\right)_{aerosols}}{\left(X_i / X_{\text{Re}}\right)_{crust}} \tag{1}$$

[revised manuscript text omitted]

Before characterizing the inorganicTo understand the impact of dust events on atmospheric particulate nitrogen in atmospheric particles at Baguanshan site,downwind areas, we collected aerosol samples in the coastal region of the Yellow Sea in the spring from 2008 to 2011.Wwe first examined the mass concentrations of  TSP as well as concentrations of crustal and anthropogenic metals to compare samples collected on dust days and immediately before or after days.  The comparative results are highlighted below. For these comparison samples,  the TSP concentrations  rangeof from 94 to 275μg·m⁻³, with an average of 201 μg·m⁻³ (Fig. 2). The TSP mass concentration were increased substantially  to 501-3857μg·m⁻³-in dust day samples, with an average of 1140.2 μg·m⁻³. In each dust day-comparison day sample pair, the mass concentration of  TSPs increased by 0-14.01303% with the median value of 537%(mean: 5.9) .  A similar increase was present in the crustal elements in each pair of samples. For example, the mean concentrations of Sc, Al, Fe, Ca and Mg were increased by over a factor of four in dust day samples relative to the comparison samples. In addition,  the enrichment factors (EF) of Al, Fe, Ca, and Mg were less than three in dust day samples but  less than  14in each pair (Table 3).  lower  values are indicative of these elements from the the primari crustal origin~~s of these elements. We found that the mean concentrations of Sc, Al, Fe, Ca and Mgincreased by over a factor of four as compared to those on ND days. Al concentrations in dust weather increased 1.7 to 21.9-fold (mean: 6.9) on ND days. The Al concentration of the "geometric mean×2GSD" (where GSD is the geometric standard deviation) was used as a criterion to define major AD events in areas of East Asia (Hsu et al., 2008). Al concentrationswere higher than the criterion level in all dust samples, which indicated that the samples we collected on dust days were truly affected by dust events. Fe was 10.3 times higher on dust days than on ND days. Additionally, nss-Ca, a typical dust index, increased 3.6-fold on dust days (Fig. 2). The EF of the anthropogenic metal elements decreased on dust days. Cu, Pb, Zn, Cr, Hg and As had high EFs,much greater than 10,on ND days, which indicated that these metal elements were mainly from anthropogenic sources.Thetheseon-12-foldon averagerelative comparedon ND days.Additionally, the EFs of~~

these anthropogenic elements decreased on dust days. These data are consistent with the very low EFs of these elements in dust particles. Thus indicated a decreasing relative contribution of  anthropogenic sources to the total TSP mass in dust day samples.

Table 2. The average concentrations and EFs of metal elements on dust and non-dust days.

| Element | Concentration (ng/m³) | | EF* | |
| --- | --- | --- | --- | --- |
| | Non-dust days | Dust days | Non-dust days | Dust days |
| Sc | 1.11 | 13.90 | - | - |
| Al | $8.53\times10^{3}$ | $6.86\times10^{4}$ | 3.8 | 1.4 |
| Fe | $4.91\times10^{3}$ | $3.88\times10^{4}$ | 3. | 1.2 |
| Ca | $1.05\times10^{4}$ | $4.29\times10^{4}$ | 14.0 | 2.1 |
| Mg | $1.62\times10^{3}$ | $1.58\times10^{4}$ | 3.5 | 1.1 |
| Cu | 50.2 | 124.5 | 36.3 | 6.1 |
| Pb | 127.9 | 221.0 | 389.4 | 56.1 |
| Zn | 340.0 | 457.7 | 248.9 | 20.6 |
| Cr | 33.8 | 244.0 | 44.0 | 11.1 |
| Hg | 0.26 | 0.36 | 176.0 | 13.8 |
| As | 25.5 | 27.4 | 707.2 | 43.9 |

*The EF factor was calculated using Scandium as the reference element (Han et al., 2010).

**3.2 Concentrations of inorganic nitrogen in dust day samples**

 When the mass concentrations of  $NH_4^+$ and $NO_3^-$ in each pair of TSP samples were compared, the concentrations of  $NH_4^+$ increased by 20%-5.7in some dust day samples (20080301, 20080315, 20090316, 20100315, 20100320, 20100315. 20110415 and 20110418), but slightly or greatly  decreased by 40-85%

concentration (less than 20% of that on ND days) oin other dust day samples (Fig. 3, Table S2). The same was general true for the measured concentration of $NO_3^-$. Similar to ammonium, nitrate displayed two different concentration variationson dust days. Nitrate concentrationsincreased by a factor of 1.4-9.2on some dust daysand decreased on other dust days.The secondary inorganic ion$SO_4^{2-}$ exhibited concentration variations that were similar to those ofnitrate. Therefore, the influence of dust on these secondary ions was different from that on crustal metal elements, andthe effect of dust on inorganic nitrogen differed during different types ofdust events.

[Figure]

**Figure 34. Mass concentration of** $NH_4^+$, andNO$_3^-$and $SO_4^{2-}$in aerosol samples collected at the Baguanshan site on dust and comparison days during March – May in the coastal region of the Yellow Sea on non-dust and dust days from 2008 to 2011.

Considering the relative values of $NH_4^+$ and $NO_3^-$ in dust day samples relative to comparison samples, the dust day samples can be classified into three categories (Table 4). In Category 1, the mass concentrations of $NH_4^+$ and $NO_3^-$ were larger in dust day sample than in the comparison samples. In

Category 2, the reverse was true. In Category 3, the mass concentrations of $NO_3^-$ were lower in the dust samples than in the comparison samples, whereas the reverse was true for $NH_4^+$. Considering that the Yellow Sea was mainly affected by dust storms from the Hunshandake Desert (Zhang and Gao, 2007), we compared our observations with the sand particles collected from this desert (Table 5). The relative mass concentrations of nitrate and ammonium to the total mass of sand particles were very low, i.e., less than 81μg/g, approximately three orders of magnitude less than the corresponding values in our dust samples. Moreover, the values obtained from atmospheric aerosols at Duolun (Cui, 2009) and Alxa Right Banner (NiuandZhang, 2000) were also more than one order of magnitude lower than the corresponding values in this study (Table 5). This suggested that $NO_3^-$ and $NH_4^+$ observed in the dust day samples were very likely due to interactions and mixing between anthropogenic air pollutants and dust particles during transport from the source zone to the reception site (Cui et al., 2009,Wang et al., 2011; Wu et al., 2016). However, along the different transport paths of Asian dust, air pollutant emissions, meteorological conditions, chemical reactions, and other factors can affect the abundance of $NH_4^+$ and $NO_3^-$ in atmospheric particles. These factors canvary greatly among different dust events, hence leading to the three different categories.

Considered relative values of $NH_4^+$ and NO3- in dust day sample against in comparison sample, these dust day samples can be classified into three categories When we focused on inorganic nitrogen (IN), we found that IN concentrations could be grouped into three cases (Table 3). InN Category 1, the mass concentrations of $NH_4^+$ and $NO_3^-$ were larger in dust day sample than in comparison sample.In Category 2, the reverse was true. In Category 3, the massconcentrationof $NO_3^-$swere less in dust samplehigher on dust days than on    in comparison sampleND days for Case 1, while IN was loweron dust days for Case 2. For Case 3, nitrate concentrationson dust days were less than on ND days, while ammonium concentrationson dust days were slightly higher than those on ND days the reverse was true for $NH_4^+$.

Considering that To understand the influence of dust on the nitrogen concentration, we compared the IN content in aerosols from the coastal regionof the Yellow Sea with sand particles and atmospheric aerosols from Duolun, a site very close to the Zhurihe Sand Desert. The Yellow Sea is was mainly affected by dust storms from the Zhurihe Sand desertthis sand source (Zhang and Gao, 2007). ), we compared our observation with those sand particles collected at the desert (Table 4). The relative mass concentrations of nitrate and ammonium to the total mass of sand particles were much low, i.e., less than $81\mu g/g$, which were three orders of magnitude less than the corresponding values inherently with our dust samples. Moreover, the values obtained from atmospheric aerosols at Duolun, a site very close to the Zhurihe Sand Desert, in literature (Cui et al., 2009), were also more than one order of magnitude less than the corresponding values in this study (Table 4). This suggested that $NO_3^-$ and $NH_4^+$ observed in the dust day samples were related to interactions of anthropogenic air pollutants and dust particles during the transport from the source zone to the reception site (Zhang et al., 2016; Cui et al., 2014.). However, along the different transport paths of Asian dust, air pollutants' emissions, meteorological conditions, chemical reactions, etc., can affect the abundance of $NH_4^+$ and $NO_3^-$ in atmospheric particles. These factors could vary a lot in different dust events, leading to three different categories.

**3.3 Theoretical analysis of three categories**

[revised manuscript text omitted]

From Table 4, we found that nitrate and ammonium concentrations in the source sand particles were very low (less than 50 μg/g). Therefore, the dust particlesin this th source area that affectthe Yellow Seaarenutrient poor.Although the IN content in aerosols particles at Duolunin dust days was higher than that th in sand particles, the nitrate and ammonium concentrations were much lower than osein the coastal region of the Yellow Sea.Other researchers also found the nitrate concentration in dust particles in downwind area higher than that at source region (Zhang et al., 2016; Cui et al., 2014). Therefore, we believe that the dust particles from the source have a dilution effect on atmospheric particulate nitrogen because of the low IN concentration in sand particles. if the dust particles just mixing with anthropogenic particles and no effective heterogeneous reaction on dust particles during the transport. When dust events occurred and no effective formation of nitrate and ammonium reaction happen, the content of nitrogen per particle mass decreasedbecause of the dilution of particulate nitrogen resulting from theincreasednumber of nutrient-poor dust particles rapidlyleaving the source area, . And we really found some samples in Case 2, such as 20080425, 20080528, 20080529, 20110319 and 20110501, had a lower IN content in aerosols particle on dust days thanthatonND.The nitrate concentration decreased 30-95% and ammonium decreased 16-72% for these samples. We think these dilution effect resulted by mixing is related with dust intensity of dust events and the distance from the dust source. In the region close to the dust sources, the IN content was very low in aerosols in dust days due to IN poor dust particlesandno effective emission or absorption and reaction occur during such short transport time. However, during long range transport, the concentration of IN will increase by reacting with gas emitted into the air under appropriate reaction conditions or by mixing withanthropogenic aerosol particles from a local source. Therefore, IN concentrations will increase in aerosols in downwind areas because of reactions on the dust surfaces or mixing with anthropogenic particles along the transport path. Cui et al. (2009) observed one dust event at four sites on transport path (Duolun, Beijing, Taishan, Shanghai), and found the enhancement and complex of mixing of dust with anthropogenic particles during transport from source to downwind region. In addition, the dilution effect was affected by dust intensity. For our samples in Case 2, Sample 200805028corresponded to a severe dust storm; 080529 and 110501 was related a dust storm; 110319 was a blowing dust (Table 3). IN concentration decreased by 66-84%, 36-71% and 5-28% for samples collected on severe dust storm, dust storm andblowing dust, respectively. The results showed The the stronger a dust storm is and the closer to the source, the stronger the dilution effect is.

**Table 34.**Average concentrations of inorganic nitrogen, TSP, NOx, Relative Humidity (RH) and T for eachcaseinaerosolsamplesin the coastal region of the Yellow Sea

| | Sample numbers | TSP µg·m⁻³ | NO₂⁻ µg·m⁻³ | NH₄⁺ µg·m⁻³ | RH % | T °C | NOx µg·m⁻³ | Summary |
|---|---|---|---|---|---|---|---|---|
| Case 1 | 080301, 080315, 090316, 100321, 110415, 110502 | 696 | 24.1 | 14.9 | 46.6 | 13.8 | 62.7 | IN>ND |
| Case 2 | 080425, 080528, 080529, 110319, 110501 | 1199 | 3.1 | 2.6 | 29.2 | 19.8 | 52.3 | IN<ND |
| Case 3 | 100315, 100320, 110418 | 1639 | 4.9 | 4.7 | 10.1 | 10.1 | 70.7 | NO₃⁻<ND NH₄⁺≅ND |
| Non-dust | | 212 | 5.5 | 4.6 | 42.2 | 13.7 | 59.7 | |

Therefore, IN concentrations will increase in aerosols in downwind areasbecause of reactions on the
dust surfacesor mixing with anthropogenic particles along the transport path. If no effective emissionor absorption and reaction occur during transport, theIN content per particle mass (μg/g) will decrease in atmospheric aerosols in the downwind area. In Case 1, the particle concentration in the dust events wasless than 700 μg/m³, and the IN concentration in atmospheric aerosols increased by a factor of more than three in dust events in the coastal region of the Yellow Sea, which might be a result of slow transport or a weak dilution effect. High relative humidity (RH), low temperature and high $NO_x$transport over a long distance and at a low speed arebeneficial to the formation of nitrate and ammonium. In Cases 2 and 3, the particle concentrationswere very high, with an average higher than 1100μg m⁻³, which indicatedthat the samples were affected by a strong dust storm or that the dust might be transported quickly. Concentrations of IN in aerosols in dust events decreased at the downwind site in Case 2 because the low RH, high temperature and low NOxduring rapid transport were not advantageous to the formation of IN. In Case 3, the low IN content was a result of a strong dilution effect and low RH.

In addition, the transport path affected the IN content of aerosol particles in the downwind area, and this influence will be discussed in Section 3.3.

**Table 45.** Comparison of the IN content in dust particles according to the dust source region (unit: μg/g)

| Sands sampled in Zhurihe | | Aerosols in Duolun* | | Aerosols in the coastal region of the Yellow Sea | |
| --- | --- | --- | --- | --- | --- |
| $NO_3^-$ | $NH_4^+$ | $NO_3^-$ | $NH_4^+$ | $NO_3^-$ | $NH_4^+$ |
| 25.46±22.87 | 4.21±1.03 | 1200 | 900 | Non-dust: | Non-dust: |
| | | | | 28,200±24,819 | 24,063±21,515 |
| | | | | Case 1: | Case 1: |
| | | | | 34,892±9570 | 22,571±7016 |
| | | | | Case 2: | Case 2: |
| | | | | 5542±5117 | 4758±5698 |

|  | Case 3: | Case 3: |
|---|---|---|
|  | 6359±4697 | 7059±5591 |

\* Adapted from Cui (2009)

**3.3 4Influence of transport pathways on particulate inorganic nitrogenin dust samples**

The calculated air mass trajectoriesof 13 out of 14 samplesshowed that the air mass originatedfromInner Mongolia, China (Fig. 5), generally consistent with the results by Zhang and Gao (2007). Theremaining one originated from Northeast China. Figs. 6 and 7 show a few areas with high emissions of $NO_x$ and $NH_3$, e.g., Liaoning, Beijing-Tianjin-Hebei, Shandong, Henan and Jiangsu in China. The calculated trajectories showed that all the air mass passed over parts of these highly polluted regions and experienced different residence time in these regions. In Fig. 5, except for the one exterior sample, all trajectories in Category 1 showed that the air masses were transported from either the north or northwest over the continent. In Category 2,the air massescrossed over the sea for 94-255 km prior to arriving at the reception site. $NH_3$-poor conditions in the marine atmosphere disfavored the formation and existence of ammonium nitrate. On the other hand, the humid marine conditions might have enhanced particle-particle coagulation and might have led to the release of $NH_3$ via reactions between preexisting ammonium salts and carbonate salts. Moreover, we also examined the links among the measured concentrations of particulate ammonium and nitrate, the mixing layer along the back trajectories, and the residence time of air masses crossing over the highly polluted zones. The results supported our hypothesis, i.e., ammonium salts mostly co-existed with dust aerosolsexternally. For example, except for 20080425, all dust day samples mostly traveled at an altitude above the mixing layer before mixing down to ground level. For most sampling days in Category 1,the average mixing layer was less than 900 m, favoring the trapping of locally emitted anthropogenic air pollutants in the mixing layer. In addition, the air massesat this elevation apparently moved slowly and took over 10 hr to cross over the highly polluted area. Even lower speeds were expected for air masses atthe bottom of the mixing layer, as wind speed decreases with height. Except for exterior samples, the sampling days in Category 2 featured a mixing layer that was, on average, higher than 900 m. The air masses at this elevation took less than 10 hr to cross over the highly polluted areas and generally had higher speeds. Theoretically, a lower mixing layer and a lower wind speed favored the accumulation of air pollutants and the formation of ammonium nitrate to some extent. The transport of dust air masses above the mixing layer reduced the possibility for internal mixing of ammonium salts and reactionwith dust aerosols along the long transport path. The shorter time for dust air masses mixing down to ground level before arriving at the reception site also increased the possibility for external co-existence between ammonium salt aerosols and dust aerosols in Category 1. The reverse could be argued to explain the observations for Category 2. The correlation analysis results in Table S2 indirectly supportthese conclusions. In fact, previous studies proposed that nitrate is rarely formed on the surface of dust particles (Zhang and Iwasaka, 1999). Therefore, much lower nitrate concentrations were observed in Category 2. Noted that the exteriors with ID of 20110415 and 20110502 have not yet been explained.

The reported threshold of windspeed for dust mobilization in the Gobi Desert ranges from 10-12 m/s(Choi and Zhang, 2008).We estimated 40.5±9.9 km/has the average wind speed during a dust storm according to Asia dust observations (). If air masses were transported faster than40.5 km/h,we found that the INcontent in most atmospheric aerosol samples was lower on dust days than on ND daysbecause of the strong dilution effect. This effect was observed in samples 080528, 080529, 110319 and 100315 (Table 5). If an air mass was transported over the ocean for some distance (ratio of oversea to totaldistance of at least 10%), no matter how fast the transport velocity, the IN content decreased because of theinput of clean marine air, such as in samples080425, 100320, 110418and110501. If the air mass was transported slowly (less than 42.4 km/h) or transported onlya short distance over the sea, with an oversea to total distance ratio of less than 10%, the IN content increased in samples collected in the downwind area, such as in samples 080301, 090316, 100321 and 110502. Therefore, the IN content is related to not only the transport path and speed but also local emissions and reaction conditions during transport.

Table 56. IN content, RH, NOx, transport speed and transport distance over the sea for atmospheric aerosol samples on dust days

| Group | Sample number | TSP (μg/m³) | NO₃⁻ (μg/g) | NH₄⁺ (μg/g) | Speed (km/h) | Distance over the sea (km) | Ratio of the distance over the sea to the total distance (%) |
|---|---|---|---|---|---|---|---|
|   |  |  |  |  |  |  |  |
| |  |  |  |  |  |  |  |
| |  |  |  |  |  |  |  |
| |  |  |  |  |  |  |  |
| |  |  |  |  |  |  |  |
| |  |  |  |  |  |  |  |
|   |  |  |  |  |  |  |  |
| |  |  |  |  |  |  |  |
| |  |  |  |  |  |  |  |
| |  |  |  |  |  |  |  |
| |  |  |  |  |  |  |  |
|    |  |  |  |  |  |  |  |
| |  |  |  |  |  |  |  |
| |  |  |  |  |  |  |  |
* * *
[Figure]

**Figure 4.**The 72-h backward trajectories for non-dust and dust samples from 2008 to 2011

Fig. Daily mean emission of NOxdust

**3.4 5 Source apportionment of aerosols from dust and non-dust events**

[Figure]

**Figure 5.**

The sources of atmospheric aerosols on dust and ND days were determined by running the PMF

model (Paatero and Tapper, 1993; Paatero, 1997).  Fig. 8 showed that atmospheric aerosols on  comparison days  mainly  consisted of six sources: industry, soil dust, secondary aerosols, sea salt, biomass burning, and coal combustion  the other uncertain sources .

with 90% of the scaled residuals falling between -3 and +3; $r^2$=0.97. On dust days, the sources of aerosols differed from those on ND days, mainly including oil combustion, industry, soil dust, secondary aerosols, and coal combustion  other uncertain sources. These value are compared in

Table 7.

The contribution of soil dust increased from 23% to 36% on dust days relative to comparison days, consistent with the high concentrations of TSPs and crustal metals observed on dust days. Liu et al. (2014) also found an even larger increase inthe contribution of dust aerosols to PM$_{10}$, i.e.,

31%-40%, on dust days relative to non-dust days.

the contributions of local anthropogenic sources decreased on dust day samples, especially those of secondary aerosols,

The source profile for coal combustion in dust day samples showed a high percentage of K$^+$, Cl$^-$, Ca, Mg, Co, Ni, As, Al and Fe, indicating a mixture of coal combustion and other pollutants emitted along the transmission path on dust days. This source increased due to the coal combustion emission mixing with other uncertain source emitted to the air under strong wind.

The calculation results also showed that the contribution of dust aerosol mass(the sum of nitrate and ammonium associated with the dust source) to the total aerosol mass (the total nitrate and ammonium) greatly increased on dust days.

~~Therefore, the sources of aerosol particles changed on dust days. Dust events had a great impact on aerosol sources in the downwind area. The influence of soil dust on aerosols and IN-loaded particles was greater than that on local sourceson dust days. In fact, the contribution of soil dust to aerosols was related to the intensity of the dust storm and the transport path. However, we could notdetermine the contributions of dust to aerosols for the different dust casesbecause of the limited number of samples.~~

|  | |  | |
|---|---|---|---|
|  |  |  |  |
|  |  |  |  |
|  |  |  |  |
|  |  |  |  |
|  |  |  |  |
|  |  |  |  |
| | |  |  |

**3. 6 Dry deposition fluxes of TSP, particulate inorganic nitrogen and metals**

Dust events are known to  the concentration and deposition of aerosol particles during long-range transport along the transport path. For example, Fu et al. (2014) found that the long-range transport of dust particles increased the dry deposition of $PM_{10}$ in the Yangtze River Delta region by a factor of approximately 20%. Some studies  reported enhancements in oceanic chlorophyll α following  dust storm events (Tan and Wang, 2014; Banerjee and Kumar, 2014).

However,  the deposition  fluxes of dust varied greatly among different dust storms, and only a few  dust episodes were followed by increases in chlorophyll *a* (Banerjee and Kumar, 2014). In addition to those in high-nutrient and low-chlorophyll (HNLC)regions, the input of nitrogen input and other nutrients associated with dust deposition is expected to promote the growth of phytoplankton. However, the extent can vary greatly depending on the nutrient limitation conditions in the oceans. A similar principle holds for the occurrence or absence of algal blooms following dust events. Thus, we calculated the dry deposition fluxes of aerosols particles, $N_{NH4++NO3}$ and metal elements during dust and comparison periods using the measured component concentrations and modeled dry deposition velocities (Table 8). We then compared the dry deposition flux of TSP and $N_{NH4++NO3}$with the previous observations in literature.

~~The role of dust deposition as a nutrient source leading toan increase in algalblooms has not been adequately addressed. To understand the influence of dust weather on the nitrogen deposition flux, we calculated the dry deposition fluxes of aerosols particles,IN and metal elements during dust and ND periods using the measured component concentrations and modeled dry deposition velocities obtainedfrom Williams' model (Qi et al., 2005)(Table 7).~~

The dry deposition fluxes of atmospheric particulates increased on dust days relative to comparison days. All increases or decreases in this section reflected the value on dust days relative to comparison days, if not specified. For example,

 –the dry deposition flux of  TSP was 2,800±700 mg/m$^2$/month in the coastal region of the Yellow SeaThe particle flux varied over a wide range from 5,200-65,000 mg/m$^2$/month under different dust sampling days, with an average of 18,800  mg/m$^2$/month.  However, the dry deposition flux of $N_{NH4++NO3}$- did not follow this pattern. In  Category 1, the dry deposition flux of $N_{NH4++NO3}$-  increased by 285%, corresponding to the increase in theTSP flux of 86-252% (Table S3). In Categories 2 and 3, the dry deposition fluxes of TSP increased by 126% to 2226% compared to that on comparison days. Except for ammonium in Category 3, the dry deposition fluxes of particulate $N_{NH4++NO3}$, however, decreased by 41% (on average). A larger relative decrease was found for the concentration of nitrate, i.e., decreases of 73%

and 46% in Category 2 and 3, respectively. Note that the average ammonium deposition flux decreased by 47% in Category 2 but increase in Category 3.

~~Cases 2 and 3, the dry deposition flux increased 2.3 to 23.2-fold compared to that on ND days. Except for ammonium in Case 3, the dry deposition flux of particulate IN decreased by an average of 41% in the case of high particle concentrations. The concentration of nitrate decreased 63% and 46% in Cases 2 and 3, respectively. Additionally, the ammonium flux decreased by 14% in Case 2, while in Case 3, ammonium was higher thanthat on ND days.We found that dust events sometimes led to an increase in the nitrogen input to the ocean relative to that during ND events, but it did not always occur depending on the chemical composition of the dust particles. As discussed, dust particles may carry abundant reactive nitrogen when they travel through polluted continental atmosphere. However, the relatively pure dust particles may be transported when no air pollution occurs along the dust transport route to oceans.~~

The dry atmospheric deposition fluxes of Fe  increased by a factor of 124-370 on dust days . Atmospheric inputs of iron to the ocean  have been proposed to enhance primary production in  areas (Jickells et al., 2005). Moreover, except for Pb and Zn in  Category 2, the dry deposition fluxes of Cu, Pb and Zn increased with those of nitrogen and iron on dust days. Trace metals were found to have a toxic effect on marine phytoplankton and inhibit their growth (Bielmyer et al., 2006; Echeveste et al., 2012). Liu et al. (2013) found that this inhibition coexisted with the promotion of some phytoplankton species in incubation experiments involving the addition of Asian dust samples in the southern Yellow Sea in the spring of 2011.~~In Case 3, dust wasdeposited in the ocean, the atmospheric supply of nitrogen decreased, and the atmospheric inputs of Fe and some toxic metals increased. Moreover, phytoplankton growth was affected by the addition of nutrient elements andtoxic elements. The overall effect of dust deposition on primary productivity was a combinationof these two effects. This is likely the reason why inhibition coexisted with the promotion of some phytoplankton species in incubation experimentsusing additions of AD in the southern Yellow Sea in the spring of 2011 (Liu et al., 2013).~~

of a dust event.Dust subjected to long-range transportdoes not always increase the atmospheric input of nitrogen. Long-term observations of dust eventsmust be performed to evaluate the contributions of dust to the biogeochemistry of nitrogen and primary production in oceans.

**Table 7.**Dry deposition of aerosol particles (mg/m²/month), particulate inorganic nitrogen (mg N/m²/month) and some toxic trace metals (mg/m²/month) on dust and non-dust days

| | Dry deposition flux | | | | | | |
| | Particles | NO$_3^-$-N | NH$_4^+$-N | Fe | Cu | Pb | Zn |
| Case 1 IN>ND | 9600± 4300 | 87±53 | 25±13 | 650±340 | 2±1 | 0.3±0.2 | 6±3 |
| Case 2 IN<ND | 18000± 11,000 | 13±18 | 5±7 | 1300±1000 | 3±2 | 0.08±0.04 | 4±1 |
| Case 3 NO$_3^-$<ND NH$_4^+$≅ND | 29,000± 31,000 | 26±6 | 17±8 | 2100±2200 | 6±1 | 0.20±0.02 | 5±3 |
| Non-dust | 2800± 700 | 48±33 | 8±8 | 190±110 | 1±1 | 0.09±0.1 | 5±4 |

| | | | | | |
| This work | 2008-2011 | Qingdao, coastal region of the Yellow Sea | Non-dust day Dust day | 2800±700 10138±15940 | 63±39 66±61 |
| Shi et al., 2013 | 2007 | The Yellow Sea | Non-dust day | | 19.2 |

| Dust day | ▲ | 104.4 |
| Average of dust and non-dust | ▲ | |

**3.7 Potential impacts of nitrogen dry deposition flux associated with dust influenced by anthropogenic activity**

Due to anthropogenic activity and economic development, inorganic nitrogen emissionsincreasedin China from 1980 to 2010 (Fig.S5). Accordingly, the $N_{NH4++NO3}$ dry deposition flux should have theoretically increased with the increase in inorganic nitrogen emissions. However, from the limited data shown in Table 9, we did not find the expected increase in dry deposition flux of inorganic nitrogen during the dust days. Considering the uncertainty in dry deposition velocity, we normalized the dry deposition flux of $N_{NH4++NO3}$ using the concentration of nitrate and ammonium reported in the literature and the recommended dry deposition velocity of 1 cm/s for nitrate and 0.1 m/s for ammonium in coastal areas reported by Duce et al. (1991). We then found that dry deposition fluxes of $N_{NH4++NO3}$ over the Yellow Sea during the dust days increased greatly from 1999 to 2007. The fluxes of $N_{NH4++NO3}$ in Qingdao, including during the dust days, varied narrowly in a range of 94.75-99.65 mg N/m$^2$/month from 1997 to 2011(Table 8). The complicated results may reflect the combined effects of NOx and NH$_3$ emissions in northern China, the occurrence frequency and intensity of dust events and metrological conditions affecting the transport pathways and moving speeds of dust air masses and chemical reactions occurring therein. For example, dust events commonly exhibited a periodic variation from 2000 to 2011 (Fig.S5).

**4 Conclusion**

The concentrations of nitrate and ammonium in TSP samples varied greatly from event to event on dust days. Relative to non-dust day samples, the concentrations were both higher in some cases and lower in others. The observed ammonium in dust day samples was explained by ammonium salt aerosols co-existing externally with dust aerosols or the residual of incomplete reactions between ammonium salts and carbonate salts. NO$_3^-$ in the dust day samples was partially related to mixing and reactions between anthropogenic air pollutants and dust particles during the transport from the source zone to the reception site. However, this process was generally much less effective and led to a sharp decrease in nitrate in Category 2 TSP samples. The external co-existence of ammonium salt aerosols with dust aerosols and the extent of the reactions between ammonium salts and carbonate salts were apparently associated with the transport pathway, moving speeds and metrological conditions, among other factors.

The concentration of  nitrate and ammonium in TSP samples varied greatly from event to event on dust days. Relative to non-dust day samples, the concentrations were both higher in some cases and lower in others. The observed ammonium in dust day samples was explained by ammonium salt aerosols co-existing externally with dust aerosols or the residual of incomplete reactions between ammonium salts and carbonate salts. $NO_3^-$ in the dust day samples was partially related to mixing and reactions between anthropogenic air pollutants and dust particles during the transport from the source zone to the reception site. However, this process was generally much less effective and led to a sharp decrease in nitrate in Category 2 TSP samples. The external co-existence of ammonium salt aerosols with dust aerosols and the extent of the reactions between ammonium salts and carbonate salts were apparently associated with the transport pathway, moving speeds and metrological conditions, among other factors.

 Due to a sharp increase in dust loads on dust days, the contribution of soil dust to the total aerosol mass was higher on dust days than on comparison days, while the contributions from local anthropogenic sources were accordingly lower.

Overall, this study strongly suggested that atmospheric deposition of $N_{NH4++NO3-}$on dust days varied greatly and that no simple linear increase existed with increasing dust load. More observations at various locations are needed to obtain a statisticalrelationship between dust events andatmospheric deposition of $N_{NH4++NO3-}$. A simple assumption ofa linear increase in$N_{NH4++NO3-}$ with increasing dust load, like that in the literature, could lead to considerable overestimation ofthe dry deposition flux of nutrientsinto the oceans and the consequent primary productionassociated with dust events.

Dust events enhance the input of atmospheric particulates via dry deposition. However, the influence of dust events on the input of nitrogen to the ocean is still uncertain. The dry deposition flux of IN on dust days decreased when a strong dilution effect was present. The contribution of dust events to marine nitrogen inputs and primary production could be overestimated if the dry deposition flux of nutrients is estimated using onlyparticulateconcentrationson dust days.

*Acknowledgments.* This work was supported by the Department of Science and Technology of the P. R. China through the State Key Basic Research & Development Program under Grant No. 2014CB953701 and the National Natural Science Foundation of China (No. 41375143). We thank Prof. Yaqiang Wang and Jinhui Shi for the valuable discussion regarding this research. We also express our appreciation to Tianran Zhang for help with sand sampling and Qiang Zhang for data collection.We thank Prof. Xiaohong Yao for the valuable discussion regarding this research. We alsoexpress our appreciation to Tianran Zhang for help with sand sampling.

[revised manuscript text omitted]

[a]For the corresponding non-dustday for each dust event, see Table 1.

**Table 5.** Comparison of the inorganic nitrogen (DIN) content in sandand aerosol particles on dust days or close to the dust source region (unit: $\mu g/g$)

| Sands sampled in dust source regions | | | Aerosols in or close to dust source region on dust days | | | Aerosols in the coastal region of the Yellow Sea | |
|---|---|---|---|---|---|---|---|
| Study region and data source | Relative concentration[a] | | Study region and data source | Relative concentration[a] | | $NO_3^-$ | $NH_4^+$ |
| | $NO_3^-$ | $NH_4^+$ | | $NO_3^-$ | $NH_4^+$ | | |
| Zhurihe (This study) | 25.46± 22.87 | 4.21± 1.03 | Duolun (Cui, 2009) | 1200 | 900 | Non-dust: 28,200±24,819 | Non-dust: 24,063±21,515 |
| AlxaLeft Banner, Inner Mongolia (NiuandZhang, 2000) | 62.1±7.4 | 79.1±1.1 | AlxaRightBanner, Inner Mongolia (NiuandZhang, 2000) | 1975[b] | 4091[b] | Category 1: 34,892±9570 | Category 1: 22,571±7,016 |
| Yanchi, Ningxia (NiuandZhang, 2000) | 46.4±2.2 | 80.9±1.3 | Hinterland of theTaklimakan Desert, Xinjiang (Dai et al., 2016) | 142-233 | 2-15 | Category 2: 5,542±5,117 | Category 2: 4,758±5,698 |
| | | | Average of SonidYouqi, Huade (Inner Mongolia), Zhangbei (Hebei) (Mori et al., 2003) | 253 | 710 | Category 3: 6,359±4,697 | Category 3: 7,059±5,591 |

| | | |
|---|---|---|
| Yulin, the north edge of Loess Plateau (Wang et al., 2011) | 216.4 | 80.6 |
| Golmud, Qinghai(Sheng et al., 2016) | 892.9 | -[c] |
| Hohhot, Inner Mongolia (Yang et al., 1995) | 588.1 | No data |

[a]Relative concentration of DIN per aerosol particle mass

[b] Samples collected on a floating dust day (Horizontalvisibility less than 10000 m and very low wind speed)

[c] The ammonium concentration was lower than the detection limit of the analytical instrument.

**Table 6.**Concentrations of TSP, $NO_3^-$, and $NH_4^+$; transport speed; transport distance over the sea; transport distance; air temperature; RH; average mixed layer during transport and transport time in polluted region for atmospheric aerosol samples on dust days.

| Group | Sample number | TSP ($\mu g/m^3$) | $NO_3^-$ ($\mu g/g$) | $NH_4^+$ ($\mu g/g$) | Speed (km/h) | Distance over the sea (km) | Transport altitude (m) | Mixed layer depth (m) | Residence time[a] (h) | T[b] (℃) | RH (%) |
|---|---|---|---|---|---|---|---|---|---|---|---|
| | 080301 | 527 | 38,984 | 24,107 | 40.1 | 0 | 1,160±702 | 864±745 | 39 | -2.9±11.7 | 29±10 |
| Category 1 IN>ND | 080315 | 410 | 47,611 | 34,130 | 79.1 | 0 | 4,921±1,870 | 950±525 | 13 | -32.5± 16.4 | 34±16 |
| | 090316 | 688 | 23,050 | 25,012 | 86.2 | 0 | 3,739±1083 | 702±665 | 11 | -19.1±11.7 | 42±17 |
| | 100321 | 519 | 31,741 | 18,155 | 87.2 | 0 | 3,407±1,249 | 1,113±760 | 19 | -23.0±13.6 | 42±22 |
| | 110502 | 810 | 25,995 | 13,632 | 30.2 | 177 | 3,666±1,371 | 747±957 | 26 | -13.2±15.8 | 31±13 |
| | 080425 | 256 | 4,089 | 372 | 29.6 | 0 | 887±656 | 1,161±1,040 | 10 | -2.7±6.1 | 66±13 |
| Category 2 IN<ND | 080528 | 2579 | 232 | 72 | 88.2 | 244 | 4,336±1461 | 1,064±830 | 8 | -15.5±13.6 | 31±16 |
| | 080529 | 2314 | 26 | 166 | 63.7 | 94 | 2,148±1,725 | 1,194±816 | 43 | 3.6±18.4 | 25±17 |
| | 110319 | 939 | 13,088 | 10,067 | 70.6 | 132 | 4,271±1867 | 790±719 | 27 | -26.3±20.0 | 48±32 |
| | 110501 | 502 | 8,924 | 10,631 | 35.1 | 252 | 3,212±810 | 916±1,114 | 5 | -13.4±8.5 | 39±13 |

| | | | | | | | | | | | |
|---|---|---|---|---|---|---|---|---|---|---|---|
| Category 3 | 100315 | 501 | 10,767 | 8,515 | 57.3 | 0 | 5,009±1410 | 1,110±365 | 7 | -40.4±13.3 | 45±29 |
| $NO_3^-$<ND | 100320 | 3857 | 1,418 | 884 | 76.9 | 0 | 1,284±401 | 525±371 | 10 | -12.2±6.3 | 61±16 |
| $NH_4^+\cong$ND | 110418 | 558 | 6,891 | 11,778 | 35.6 | 931 | 1,344±780 | 695±672 | 2 | -0.1±8.2 | 52±28 |

**Table 7.** Sources and source contributions (expressed in%) calculated for aerosolsamples collected during dust and non-dust events

| Dust event | | Comparison days | |
|---|---|---|---|
| Source | % of TSP | Source | % of TSP |
| Soil dust | 36 | Soil dust | 23 |
| Industrial | 21 | Industrial | 24 |
| Secondary aerosol | 6 | Secondary aerosol | 23 |
| Oil combustion | 6 | Biomass burning | 16 |
| Coal combustion and other uncertain sources | 31 | Coal combustion | 5 |
| | | Sea salt | 9 |

**Table 8.** Dry deposition of TSP (mg/m$^2$/month), particulate inorganic nitrogen (mg N/m$^2$/month) and
some toxic trace metals (mg/m$^2$/month) on dust and non-dust days.

| | Dry deposition flux | | | | | | | |
|---|---|---|---|---|---|---|---|---|
| | TSP | $NO_3^-$-N | $NH_4^+$-N | $N_{NH4+ + NO3}$- | Fe | Cu | Pb | Zn |
| Category 1[a] | 8,000±1800 | 65±9 | 24±14 | 90±17 | 533±179 | 2±0.3 | 0.3±0.3 | 6±2 |
| Category 2[a] | 18000±11,000 | 13±18 | 8±4 | 21±22 | 1300±1000 | 3±2 | 0.08±0.04 | 4±1 |
| Category 3[a] | 29,000±31,000 | 26±6 | 17±8 | 42±12 | 2100±2200 | 6±1 | 0.20±0.02 | 5±3 |
| Non-dust | 2,800±700 | 48±33 | 15±8 | 63±39 | 190±110 | 1±1 | 0.09±0.1 | 5±4 |

[a]For the characterization of $N_{NH4+ + NO3}$- concentration and sample information of the category, see in Table 3.

**Table 9.** Comparison of dry deposition flux and normalized flux of TSP (mg/m$^2$/month) and $N_{NH4++NO3}$ (mg N/m$^2$/month) with observations from other studies(mg N/m$^2$/month)

| Source | Year | Area | | TSP | $N_{NH4++NO3}$ | Normalized average flux of $N_{NH4++NO3}$ [a] |
|---|---|---|---|---|---|---|
| This work | 2008-2011 | Qingdao, coastal region of the Yellow Sea | Non-dust day | 2,800±700 | 63±39 | 93.90 |
| | | | Dust day | 10,138±15,940 | 58±36 | 101.39 |
| | | | Average of dust and non-dust | | | 97.64 |
| Qi et al., 2013 | 2005-2006 | Qingdao, coastal region of the Yellow Sea | Average of nine months samples | 159.2 - 3,172.9 | 1.8-24.5 | 94.75 |
| Zhang et al., 2011 | 1997-2005 | Qingdao | Average of annual samples | | 132 | 99.65 |
| Zhang et al., 2007 | 1999-2003 | The Yellow Sea | | | 11.43 | 9.91 |
| Shi et al., 2013 | 2007 | The Yellow Sea | Non-dust day | | 19.2 | 132.17 |
| | | | Dust day | | 104.4 | 227.07 |

| | | |
|---|---|---|
| | Average of dust and non-dust | 179.62 |

[a] The calculation method of normalized flux of $N_{NH4+ +NO3-}$ was discussed in section 3.7.

[Figure]

**Figure 1.**Location of the aerosol and dust sampling sites.

[Figure]

**Figure 2.**Modeleddust concentrations over East Asia by CFORSmodel during each dust sampling day from 2008 to 2011(http://www-cfors.nies.go.jp/). (The figures showthe modeled dust concentration inthe middle of each sampling duration). Nodata are available forMar.19, 2011,because of the earthquake in Japan.Hourly PM10 concentrationsweremodeled by the WRF-CMAQ model for eachsampling day,and the resultsareshownin Fig. S3.

[Figure]

**Figure 3.**Mass concentrations of TSP、Al、Fe and nss-Ca in aerosol samples collected at the Baguanshan site on dust and comparison days from2008-2011.

[Figure]

**Figure 4**. Mass concentrations of $NH_4^+$ and $NO_3^-$ in aerosol samples collected at the Baguanshan site on dust and comparison days during March-May in 2008 to 2011.

[Figure]

[Figure]

**Figure 5.** The 72-h backward trajectories for non-dust (a) and dust (b) samples from 2008 to 2011(the yellow domains in the maps represent the dust source regions in China).

[Figure]

**Figure 6.** Daily mean emission of NOxover East Asia on the dust daysfrom 2008 to 2011.

[Figure]

**Figure 7.** Daily mean emission of NH₃ over East Asia on the dust days from 2008 to 2011.

[Figure]

[Figure]

**Figure 8.**Source profiles of atmospheric aerosol samples collected on non-dust (a) and dust (b) days
using the PMF model

|---|---|---|

字体: 10 磅

|---|---|---|

字体: 10 磅

|---|---|---|

字体: 10 磅

|---|---|---|

字体: 10 磅

|---|---|---|

字体: 10 磅

|---|---|---|

字体: 10 磅

|---|---|---|

字体: 10 磅

|---|---|---|

字体: 10 磅

|---|---|---|

字体: 10 磅

|---|---|---|

字体: 10 磅

|---|---|---|

字体: 10 磅

|---|---|---|

字体: 10 磅

|---|---|---|

字体: 10 磅

字体: 10 磅

字体: 10 磅

字体: 10 磅

字体: 10 磅

字体: 10 磅

字体: 10 磅

字体: 10 磅

字体: 10 磅

字体: 10 磅

字体: 10 磅

字体: 10 磅

字体: 10 磅

字体: 10 磅

字体: (默认) Times New Roman, 10 磅

字体: (默认) Times New Roman, 10 磅

字体: (默认) Times New Roman, 10 磅

字体: (默认) Times New Roman, 10 磅

字体: (默认) Times New Roman, 10 磅

字体: (默认) Times New Roman, 10 磅

字体: (默认) Times New Roman, 10 磅

字体: (默认) Times New Roman, 10 磅

字体: (默认) Times New Roman, 10 磅

字体: (默认) Times New Roman, 10 磅

字体: (默认) Times New Roman, 10 磅

字体: (默认) Times New Roman, 10 磅

字体: (默认) Times New Roman, 10 磅

字体: (默认) Times New Roman, 10 磅

字体: (默认) Times New Roman, 10 磅

字体: (默认) Times New Roman, 10 磅

字体: (默认) Times New Roman, 10 磅

---

## Referee Report (RR1)

General Comments: The manuscript titled 'The concentration, source apportionment and deposition flux of atmospheric particulate inorganic nitrogen during dust events' written by Jianhua Qi presented the dust impacts on particulate inorganic nitrogen by analyzing the aerosol samples collected at Qingdao, China. The authors divided dust pattern into three parts, and investigated the dry deposition flux. To estimate the source, PMF receptor model was also used. Based on the above approaches, the authors tried to answer the questions of 'dust event always increase the atmospheric input of nitrogen to the ocean?'. The topic is interested ones because the impact of dust as atmospheric input on ocean ecosystem has been still unclarified. However, throughout the manuscript, it is not well organized and hard to follow and understand. Overall, this manuscript will not be acceptable taking into account the high journal quality of Atmospheric Chemistry and Physics.

Reply: We will revise the manuscript according to the comments to improve the manuscript quality.

First of all, the space was not inserted appropriately in many parts, so it is hard to read and follow. Such a crude revision with low presentation quality should not be sent to reviewers.

I have reviewed again this manuscript and found some improvements on the manuscript; however, many replies have not been found in the revised manuscript and/or replied well to my concerns. I feel that the presentation quality is still low as to be published from the high quality journal of Atmospheric Chemistry and Physics. With regret, I have judged to reject this manuscript again.

Q1. Before the discussion, first, the definition of "dust events" cannot be understood well. In L99-101, the authors explained that 'Samples were collected on dust days and selected ND days in spring from March 2008 to May 2011, with sampling duration of 4h for each sample. We refer to the ND days as sunny and cloudy days before and after dust events in the following discussion'. The authors should add the appropriate reference of the Meteorological Information Comprehensive Analysis and Process System (MICAPS) which defined the weather conditions (and also, the subsection 2.4 should be reorganized partly into this explanation). What is the definition of "dust events" here? Visibility? More information of how the dust events are defined in this system should be announced in detail. Total of 14 samples (sample numbers in Table 3) during dust events were analyzed throughout this study. The sampling duration was 4 hrs, so which data are used in the corresponded date in Table 3? All samples in the day? Moreover, what is the sample numbers of ND? The current information in Section 2.1 is severely lacked in the information which the readers

can follow the authors methodology. Because this study discussed the dust impact, the explicit and detailed information regarding dust is required. In this sentence, I am worried about the explicit division of dust and non-dust samples. It is well known that some dust events are continued a few days. For example, the samples used in this study during 28-29 May 2008, 20-21 March 2010, 15 and 18 April 2011, and 1-2 May 2011 showed continuous dust events. In such cases, do the authors have confidence to the clear separation of dust and non-dust samples? How about the Al concentration definition (L171-172) of non-dust days samples? Why were other days samples not collected to clearly separate the dust impacts? The definition of ND is ambiguous. According to the definitions of dust and non-dust, the discussion on dust impact might be changed. The reconsideration of dust impact is needed based on the clear definitions of dust.

Reply: In this study, the dust event was defined according the definition adopted in regulations of surface meteorological observation of China (CMA, 2003; Wang et al., 2008) and identified based on the meteorological records information from Meteorological Information Comprehensive Analysis and Process System (MICAPS) of China Meteorological Administration. Each dust sample was collected for 4hrs duration and the sampling started only when the PM10 mass concentration available on the website (http://www-cfors.nies.go.jp/~cfors/; http://www.qepb.gov.cn/m2/) was increased greatly. The approach made the dust sample more representative relative to urban background. However, for dust event with duration less than one day, only one sample was collected; for dust event with longer duration, i.e. multiple days, the sample was collected once a day. The sampling information was listed in the Table S1. Based on the forecast, we also collected aerosol particle samples immediately before or after the dust event for comparison. These comparison samples were further classified into sunny day samples, cloudy day samples and post-dust samples. The post-dust samples were featured by collecting under a clear and sunny weather condition and lower mass concentration of PM10. Moreover, the concentration of Al referring to the total Al concentration in TSP samples were used to confirm the division of dust or comparison samples according to the criterion "geometric mean$x$2GSD" proposed by Hsu et al. (2008).

CMA: Regulations of Surface Meteorological Observation, China Meteorological Press, Beijing, 154–156, 2004.

Hsu, S. C., Liu, S. C., Huang, Y. T., Lung, S. C. C., Tsai, F., Tu, J. Y., and Kao, S. J.: A criterion for identifying Asian dust events based on Al concentration data collected from northern Taiwan

between 2002 and early 2007, Journal of Geophysical Research Atmospheres, 113, 1044-1044, 2008.

Wang Y. Q., Zhang X. Y., Gong S. L., Zhou C. H., Hu X. Q., Liu H. L., Niu T., Yang Y. Q.: Surface observation of sand and dust storm in East Asia and its application in CUACE/Dust, Atmos. Chem. Phys., 8, 545–553, 2008.

I have partly understood my concerning issue regarding the definition of dust event. Further concerning issue is the sampling duration of continuous dust event. Even the dust event continued multiple days, how should we consider the representativeness of the sampling? For instance, sample 20080528 and 20080529 (please note that the sampling time of 20080529 will have typo) had approximately one day interval. Was there large temporal variation of PM10 concentration during continuous dust days? If there was large change on PM10 concentration, why the authors collected on the listed time? The authors should state the reason, and should present the representativeness of 4 hrs sampling.

In the revised manuscript, it will be kind for readers to explicitly state that 'http://www-cfors.nies.go.jp/~cfors/' is for forecast model over Asia, and 'http://www.qepb.gov.cn/m2/' is for observed concentration at Qingdao.

Q2.The second concern is the "dilution effect" which the authors claimed as the key factor for the discussion of inorganic nitrogen. Again, without the explicit definition of dust and non-dust, the dilution effect cannot be understood well. In this discussion, although the authors introduced the air mass speed, there were no implications on the intensity of dust events itself. Why the upwind (i.e., near desert) information was not used here to describe the dust intensity? The dilution is not so simple, hence more information are required to reinforce the authors finding. The authors discussed the inorganic nitrogen behavior. In these cases, what is the counter ion of NH4+ and NO3-? Are the main counter ions metal elements? If NH4NO3 are formed, due to its chemical unstablity according to the temperature and relative humidity, it is not simple to discuss only the viewpoint of "dilution effect". In addition, the authors used NO2 data to investigate the inorganic nitrogen, but how about NH3? Only from NO2 data, it is insufficient to estimate the inorganic nitrogen variation. On the above reasons, the reconsideration is required to publish this manuscript from Atmospheric Chemistry and Physics.

Reply: In revision, the part reads as "Inorganic nitrogen (IN) concentrations highly varied in different

dust samples (Table 3). According to the concentrations relative to those in comparison samples, they can be classified into three categories, i.e., Category 1 in which higher IN concentrations were observed in dust samples, Category 2 in which lower IN concentrations were observed in dust samples, and Category 3 in which lower nitrate concentrations with slightly higher concentrations of ammonium in dust samples. Category 1 was usually associated with a lower moving speed of dust air mass or a longer distance over the ocean (Table 5) while the reverse was true for Category 2. The moving speed and distance over the ocean of dust air mass in Category 3 was generally between them. Theoretically, lower moving speed of dust air mass favors reactions between dust particles and anthropogenic gaseous precursors of IN due to a longer reaction time. Large moving speed of dust air mass was frequently associated with a large wind speed in the lower layer atmosphere (Gao et al., 2010; Gillette and Passi, 1988; Peng et al., 2007; Yue et al., 2008), leading to anthropogenic gaseous precursors therein to be better diluted. Shorter reaction time and reduced concentrations of anthropogenic gaseous precursors likely lowered IN in Category 2. Moreover, the relative concentration of IN per aerosol particle mass in $\mu$g/g was analyzed and compared with those values in literature. .."It is questionable for using NOx observed in Qingdao to argue the generation of IN in dust samples. We agree this because most of IN observed in dust samples should be derived from secondary reactions upwind of Qingdao by considering a low conversion rate of NOx to IN. The former study (Liu et al., 2010) showed that NOx and NH3 generally capture the spatial distribution patterns with high values over eastern China and relatively lower values over central and western China, where dust source regions are located (Fig. S1-S3). Thus, we will add modeling results using a 3-D air quality model to support our analysis in revision.

Gao, Q X., Ren Z H. et al. : Dust events and its impacts on atmospheric environment, Science press, Beijing, 2010.

Gillett e D A, Passi R.: Modeling dust emission caused by wind erosion, J G R., 1988, 93: 14234-14242.

Liu X. H., Zhang Y., Cheng S. H., Xing J., Zhang Q., Streets D. G., Jang C., Wang W. X., Hao J. M.: Understanding of regional air pollution over China using CMAQ, part I performance evaluation and seasonal variation, Atmospheric Environment , 44,2415-2426, 2010.

Peng, Z., Liu X. M., Hong Z. X., Wang B. L.: Characteristics of Atmospheric Boundary Layer Structure and Turbulent Flux Transfer during a Strong Dust Storm Weather Process over Beijing Area, Climatic and Environmental Research, 2007, 12(3): 268-276.

Qi J.H., Gao H.W., Yu L.M. , Qiao J.J.: Distribution of inorganic nitrogen-containing species in atmospheric particles from an island in the Yellow Sea, Atmospheric Research, 101,938-955, 2011.

Wang Y. Q., Zhang X. Y., Gong S. L., Zhou C. H., Hu X. Q., Liu H. L., Niu T., Yang Y. Q.: Surface observation of sand and dust storm in East Asia and its application in CUACE/Dust, Atmos. Chem. Phys., 8, 545–553, 2008.

Yue P., Niu S. J., Liu X. Y.: Dust Emission and Transmission during Spring Sand-dust Storm in Hunshandake Sand-land, Journal of Desert Research, 2008, 28(2): 227-230.

I have partly agreed, but I have further question on the application of a 3-D air quality model. First, what is the merit of the application of 3-D air quality model? In the revised manuscript, only the spatial distributions of PM10 were shown (Fig. 2 from CFORS model and Figs. S1-S3 with CFORS and WRF-CMA. Can such application reinforce the authors' discussion points? The behavior of IN were discussed in this manuscript, so what is the purpose to show PM10? The authors stated that 'The spatial distribution of PM10 concentrations for each dust event was consistent with the model results of dust by the Chemical Weather Forecast System (CFORS) by Uno et al. (2003)' (L199-201). If the consistency between other models is important, why the author calculated on your own model? I cannot follow this reason from the revised manuscript.

The following specific points also should be revised to clarify the model application.

L189: Centered point is needed because we cannot follow the modeling domain at the current description.

L193: On the INTEX-B emission inventory (Zhang et al., 2009), I suppose that NH3 emissions have not been provided. If so, this description should be changed.

L195, and Figures 6 and 8: So, all calculations were based on the emission level on 2008? Because the temporal resolution of INTEX-B emission inventory is month, I feel that there are no need to display all emissions on all dust samples. These emissions level should be differed only on month. Therefore, I suppose that the averaged (spring time) emissions of NOx and NH3 on each one figure is enough.

Figure S1: What is the purpose to show the difference between (b) and (c)? In this caption, what is 'WRF-CMA'?

Figure S3: In the main manuscript, it was stated that 'each dust sampling day are shown' in Fig. S3 (L218-219, L895). However, only the hourly concentration of PM10 concentration at 14:00 on 19 Mar 2011 were shown. Please confirm this supplemental figure.

Specific comments:

Q3. L35-36: This conclusion does not match to the manuscript contents. The authors stated that input of nitrogen to the ocean depends on the dust events.

Reply: We apologize for the confusion in the revision. We will revise the abstract sentence into "The atmospheric input of nitrogen into the ocean depends on the dust events; dust deposition was an uncertain source of nitrogen for the ocean".

I cannot find this revision.

Q4. L57-L67: In this paragraph, the authors used "ND days" simply. However, this wording should be used carefully; because the definition of non-dust days will be different in each study. Please consider to carefully define this wording.

Reply: Thank you for this suggestion. To avoid confusion, we will use "non-dust storm days" according to the original reference in L57-L67.

I have confirmed that the authors use the wording of 'dust storm' in the introduction.

Q5. L146: Some information should be replaced on Section 2.1 appropriately.

Reply: We will move this information to Section 2.1 in the revised version.

The section in the revised manuscript is well organized.

Q6. L162: "atmospheric particulate" is "TSP"?

Reply: We apologize for the confusion. The term "atmospheric particulate" will be revised to "total suspended particulates". Atmospheric particulate concentrations were obtained by weighting TSP samples. We will revise the sentence and the corresponding figures.

I have confirmed.

Q7. L165: I cannot follow the calculation of "1.8-14.0 times (mean: 5.9)". The mean concentration have not been stated for dust days.

Reply: Each sample on dust day had its corresponding non-dust sample (Table S2).The 1.8-14.0 times was calculated as a ratio of the TSP concentration on a given dust day to the values in the comparison samples. The concentration and the ratio of samples on dust days were listed in Table S2.

I have confirmed and understood the meaning. However, is this revision corresponded to L226-230?

If so, this revised sentence seems to contain many errors (NOT Table S2 but Table S1?). For example, we can find 410 µg/m3 on dust day sample on 20080315. What is the value of 80-1303%? These increased value were not corresponded to 'Ratio of DD to CS' shown in Table S1.

Q8. L167: The EF of Ca is 14.0 in Table 2. The statement of "decreased to less than three" cannot be followed from the valued listed in Table 2.

Reply: We apologize for this error and will revise the incorrect description to read "the enrichment factors (EFs) of Al, Fe, and Mg were lower than ten on ND days and decreased to less than three on dust days. These data are indicative of the primarily crustal origins of these elements. Furthermore, the EF of Ca was 14.0 on ND days, which indicated that Ca had a partially anthropogenic source on dust days".

I have confirmed this revision.

Q9.L171: Again, I cannot follow the calculation of "1.7-21.9 times (mean: 6.9)".

Reply: We apologize for the confusion. The calculation method is the same as that for TSP (see the reply to Q7). The correct concentrations and the ratios of samples on dust days are listed in Table S2.

Q11.L175: I cannot follow "10.3 times" for Fe. It can be calculated as 7.90 from the values in Table 2.

Reply: The calculation method is the same as that for TSP (see the reply to Q7). The concentrations and corrected mean ratios of samples on dust days are listed in Table S2.

Q13.L176: "3.6-fold" will not be followed from Fig. 2. It should be listed in Table 2.

Reply: The calculation method is the same as that for TSP (see the reply to Q7). The concentrations and the corrected mean ratios of samples on dust days are listed in Table S2.

So, in this revised manuscript, these statements of the increment ratio on dust-day compared to non-dust day have not been explicitly appeared. In L243, the authors stated 'Table S1', but Table S1 contained not only the information of inorganic nitrogen but also TSP, Al, Fe, and nss-Ca. So, it is appropriate to mention on Table S1 in Section 3.1.

Q10.L173-L174: To clarify the separation of dust and non-dust days, the information of criterion for samples on non-dust days will be needed.

Reply: As discussed above for Q1, the information will be supplemented in Section 2.1.

I have confirmed this revision.

Q12.L175: In Figure 2, nss-Ca was shown, but nss-Ca was not listed in Table 2. What is the authorsintention to introduce nss-Ca here?

Reply: Follow others' study, we calculated the EF of Ca in Table 2. The EFs of Ca on ND days indicated that Ca was affected by anthropogenic sources. nss-Ca usually was used as a typical dust index. Therefore we showed the nss-Cain Fig.2 and discussed the influence of dust on crustal elements using nss-Ca.

So, please state explicitly regarding this point to the readers. In the current form, nss-Ca was suddenly shown in Fig. 3 without any introduction.

Q14.L177: The EF of Ca on dust days is also greater that 10.

Reply: The EF of Ca was 14, not much greater than 10, indicating that the Ca was mainly from a natural source mixed with an anthropogenic source.

Excuse me for my previous misreading. I have confirmed.

Q15.L183: The increasing ratio of concentration between dust days and non-dust days will be helpful to understand the discussion on Section 3.1.

Reply: We will replace the times with ratios in our revised manuscript.

I have confirmed this revision.

Q16.L189: What is the comparison method on some dust days? The sample date are shown in Figure 3, so why the authors explicitly mention the date? I cannot follow the calculation of "a factor of 1.2-5.7".

Reply: It will be revised as "The concentrations of ammonium were increased by 20"

Q17.L190: What means "less than 20% of that on ND days"? Averaged data over ND days?

Reply: We apologize for the confusion. The sentence has been revised to read "The concentrations of ammonium were increased by 20.

Q18.L191: Again, what is the comparison method on some dust days? I cannot follow the calculation of "a factor of 1.4-9.2 ".

Reply: The calculation method is the same as that for ammonium (see the reply to Q16). The concentrations and the increasing factors of samples on dust days are listed in Table S2.

First of all, I cannot find the revision of 'The concentrations of ammonium were increased by 20' anywhere. Is this corresponded to Table S1? I suppose that the authors discussed regarding this point in L240-L244. Although ratio was shown in Table S1, percentages are discussed here. So it is hard to follow the manuscript. Why the discussion point have not been arranged on the uniformed unit?

Q19.L194-L195: In this sentence, the authors stated "the effect of dust on inorganic nitrogen differed during different types of dust events". Why the authors suddenly focused on inorganic nitrogen here? In L192-193, it was mentioned "inorganic ion $SO_4^{2-}$ exhibited concentration variations that were similar to those of nitrate". L197: The figures for inorganic nitrate will be helpful information here, if the authors focused on inorganic nitrogen.

Reply: The part will be revised as "Similar to ammonium, nitrate concentrations were sometimes increased by a factor of 1.4-9.2 relative to the comparison sample while they were decreased in others. Unlike substantially increased concentrations of crustal metal elements in dust samples, the concentrations of IN were likely determined by meteorological conditions as well as surface areas provided by dust particles."

I cannot find this revision.

Q20.L207: (respectively less than 50 ug/g and 6 ug/g) will be the correct expression for ammonium.

Reply: We have incorporated this suggestion.

I cannot find this revision.

Q21.L211: So what is the source of atmospheric particulate nitrogen? The location of Duolun and Zhurihe Sand Desert is very close.

Reply: Duolun and Zhurihe belong to the Hunshandake Desert in Inner Mongolia, one of the main Chinese sand deserts. According to studies, the Yellow Sea is mainly affected by dust storms from this sand source with a probability of 52

Zhang, Z K., and Gao, H.: The characteristics of Asian-dust storms during 2000–2002: From the source to the sea, Atmospheric Environment, 41, 9136-9145, 2007.

Gao, Q X., Ren Z H.: Dust events and its impacts on atmospheric environment, Science press, Beijing, 2010.

I am wondering that the differences of IN concentration between Duolun and Zhurihe. Both are Hunshandadke Desert, however, as is shown in Table 5, IN concentration was much higher in Duolun. Are there some emission source?

Q22.L214-L216: Without more information of the intensity of dust, the discussion on 'dilution effect' seems to be lacked in scientific understanding. This part should be fully revised based on not only dilution effect but also dust intensity.

Reply: As discussed above, we will add modeling results of dust distribution to support our analysis in revision.

Again, only from the dust spatial distribution, it is hard to state the dust intensity.

Q23.L217: Averaged information were listed here, however, will the each sample information be valuable? The equation shown in summary column cannot be understood form (e.g., IN and ND were not comparable index).

Reply: Thank you for the suggestion. According to the suggestion, we revised Table 3 and listed the sample information.

I have confirmed the revision.

Q24.L219: It seems that the discussion on this paragraph (e.g., "700 ug/m3 in Case 1" and "higher than 1100 ug/m3 in Cases 2 and 3") are based on Table 3. Please reorganize the paragraph, or please refer appropriate information here. It is hard to follow these values.

Reply: We will revise this paragraph and refer to the appropriate information in the revised manuscript according to revised Table 3.

I have confirmed the revision, but if the authors discussed on average (L300-303), the averaged values were also needed.

Q25.L219-L222: So what is the local source? What is the definition of the wording of "local" here? There was no information of the emissions here. It is hard to understand the "reaction" without the information of emissions intensity around dust source and downwind regions.

Reply: Local source refers to the gas or particle emissions from a local pollutant source, such as industry emission, coal burning, vehicle exhaust and agricultural activity, in the downwind region during the dust transport, which is not from the dust event itself. As we discussed above, the NOx and NH3 emissions increase greatly from the dust source region to the downwind region (see the reply to Q2). We have supplemented the modeled emissions intensity of NOx and NH3 in the revised manuscript.

I have confirmed this revision.

Q26.L224: "particle" is "TSP"?

Reply: We apologize for the confusion. We will revise "particle" to read "total suspended particles".

I have confirmed this revision.

Q27.L227-L228: The favorable condition to form ammonium cannot be discussed without the information of NH3. In addition, Table 3 indicated the aerosol samples in the coastal region of the Yellow Sea. How about the status over air mass path? Is it sufficient to conclude only from the downwind information to the formation of inorganic nitrogen?

Reply: We will add modeling results using a 3-D air quality model to support our analysis in revision.

Again, I cannot understand the model application results.

Q28. L230: "strong dust storm" cannot be discussed without any information on dust intensity here.

Reply: We will add modeling results of dust distribution to support our analysis in revision.

Again, from the additional information of CFORS, the spatial distribution pattern was found; however, how can we estimate the intensity?

Q29. L233-L234: But NOx concentration was high in Case 3. I cannot follow why the authors concluded "the strong dilution effect" on Case 3.

Reply: Among three cases, the NOx concentration was the highest with an average value of 70.7 for Case 3 and increased by 17.8

So where did the authors discussed the NOx concentration in the manuscript?

Q30. L244-L246: Because the Table 5 was lack in the information of ND days, we cannot follow the authors conclusion. The information of ND days on Table 5 will be required.

Reply: We have supplemented the information for ND days in Table S1 and S2.

I have confirmed the supplemental information on Tables S1 and S2.

Q31.L254-L255: The authors simply mentioned "local emissions" here. Because the samples were collected on downwind regions in the coastal region of the Yellow Sea, I guess that the discussion on emission characteristics of each (or, at least, some categorized) air mass should be discussed in detail. The inorganic nitrogen concentrations are highly related to the local conditions both on emissions strength and meteorological parameters, so the discussion only on air mass speed and air mass path over ocean are insufficient.

Reply: As discussed above (see the Reply to Q2), We will add modeling results using a 3-D air quality model to support our analysis in revision.

Again, model is used only for spatial distribution and not inform the chemical production process.

Q32. L256: RH and NOx information are not shown in Table 5.

Reply: We apologize for the mistake. We have revised the title of Table 5.

I have confirmed the information of RH and NOx on the revised Table 6.

Q33.L260: The colors are overlapped, hence we cannot distinguish each trajectory. Some paths (e.g., thick green color: 2008/5/22 or 2011/4/15) are apparently indicated the west or south part of China. Are these events really related to dust events?

Reply: We apologize for the confusion. We have provided all trajectories of samples collected on dust and non-dust days. Fig.4 has been redrawn to distinguish each trajectory for samples collected on dust and non-dust days.

I have rechecked the discussion of backward trajectories discussed on Section 3.4. There are many points should be clarified.

Figure 5: Please add the explanation of the trajectory of 20110415 was excluded based on the discussion on Fig. 2. Why the authors displayed "non-dust samples"? What were the differences between non-dust and dust samples trajectories? I feel that these were similar.

L314: What is the 'remaining one'? Please specify the trajectory data. In my opinion, two trajectories

of 20110418 and 20110501 originated from northeast China.

L317: What is the 'one exterior sample'? Please specify.

L319: I cannot see 'the air masses crossed over the sea for 94-255km' from Figure 5, because Fig. 5 showed the whole view of trajectories across China. More detailed figure or explanation will be required.

L328-329: What is the definition of the 'average mixing layer'. I suppose that the altitude of backward trajectories were so high because most of trajectories were originated outside China on 72 hrs. So, where is the averaged region to calculate '900m' in this sentence?

Q34.L278-L280: The source of coal combustion have increased compared to non-dust days. Short explanation will be needed here.

Reply: The source of coal combustion on dust days became complex. The source profile showed high percentages of K+, Cl-, Ca, Mg, Co, Ni, As, Al and Fe, indicating a mixture of coal combustion and other pollutants emitted along the transmission path on dust days, such as industry and building dust. This source increased due to the coal combustion emissions mixing with other uncertain sources emitted into the air in strong winds.

I cannot still understand the authors' conclusion here. As was discussed on L355-361, Fig. 8, and Table 7, although the coal combustion have increased on dust days, the contributions of local anthropogenic sources (especially secondary aerosols) have decreased on dust days. According to the discussion on Section 3.3, the authors concluded that ammonium salts were externally co-exist with dust aerosols in Category 1. So, why the contribution of secondary aerosols were decreased from PMF analysis. I feel that these results have contradicted. More careful discussion is required for this conclusion.

Q35.L305: If the authors discuss the dry deposition flux of "IN", the information should be inserted in Table 7. Table 7 only contained NO3- and NH4+ independently.

Reply: We inserted the flux of IN in Table 7 and corrected several mistakes.

I have checked this revision.

Q37.L306: I cannot follow the calculation of "a factor of 1.1-5.8" and "a factor of 1.8-6.3".

Reply: These factors were the flux ratio of each dust sample in Case 1 to the ND average. The flux

and ratio of each sample are listed in Table S3. We recalculated the increasing factors according to the revised values. The sentence was revised to read "Compared with the average flux on ND days, the dry deposition flux of IN increased by a factor of 1.1-3.9, and the flux of atmospheric particles (TSP) increased by a factor of 1.8-6.3 in Case 1"

I cannot find this revision.

Q38.L307: "the dry deposition flux" of what?

Reply: We apologize for the mistake in the revision. The passage has been revised to read "the dry deposition flux of atmospheric particles (TSP)".

I have checked this revision on L380.

Q39. L309: What is the calculation method of "63%" and "46%"?

Reply: We apologize for the mistake. The sentence has been revised to read "Compared with the average dry deposition flux on ND days, the average nitrate flux of samples in Cases 2 and 3 decreased by 73

I cannot find this revision.

Q40. L310: What is the calculation method of "14%" ?

Reply: We corrected the calculation error and revised this sentence to read "Additionally, the average ammonium flux decreased by 47"

I cannot find this revision.

Q41. L317: I cannot follow the calculation of "a factor of 2-25".

Reply: The factor was calculated by comparing the flux of the sample on dust days with the average Fe flux on ND days (see Table S3).

For Fe, it seems that the increased ratio were 2.81-11.08 from Table S3.

Q42.L339: "aerosol particles" is "TSP"? In Table 7, please confirm the significant digits for each specie.

Reply: We apologize for the confusion. "aerosol particles" was revised to read "TSP". The former digits were revised according to the editor's suggestion. We will consider revising again to confirm

the significant digits.

The revision for TSP was found in Table 9.

Technical Corrections:

Q43. L31: Comma is needed on '2800'.

Reply: We have added a comma according to the suggestion.

Q44. L199: 'IN' should be defined in L194.

Reply: Due to the very low concentration of nitrite, in this manuscript, IN represents inorganic nitrogen, mainly including nitrate and ammonium. We have provided this definition in L194.

Q45.L236: Need appropriate comma for all numbers. L301: Comma is needed on '2800±700'.

Reply: We have added a comma according to the suggestion.

I have checked these technical corrections.

Specific comments:

Table 6: Missing the note of a and b.

Table S2: Please align the right-column, it is hard to follow. What is the meaning of *?

---

## Referee Report (RR2)

This manuscript focuses on an important issue, specifically the relationship between ammonium and nitrate with mineral dust in China. The variability in previously reported relative concentration trends is worth exploring in detail. There are a number of areas where the paper could use improvement prior to publication, particularly with connecting to the literature and placing the work in context.

Comments:

- Throughout the manuscript there are odd spacing issues, where two words are together without a space. As an example in the abstract "For these two groups, NH4+in dust day samples waspresent in the form of ammonium salts externally co-existing with dust aerosols or the residual of incomplete reactions between ammonium salt and carbonate salts."
- Line 51-54: The authors state the Asian dust has been transported as far as the north pacific, but this understates what has been observed for Asian Dust. Uno et al 2009 Nat Geosci showed that Asian dust can circumnagivate the globe. VanCuren and Cahill 2002 showed Asian dust impacting California air quality, while Ault et al. 2011 JGR and Creamean 2013 Science showed impacts of Asian dust on orographic precipitation in the Sierra Nevada (in California). Pratt et al. 2009 Nat Geo showed Asian dust influencing clouds over Wyoming.
- Though the authors note that a native English speaker was utilized for the revision, a considerable improvement in the grammar and proofreading are needed before the writing is at a publishable level.
- Line 233 insert comma after "samples"
- Line 266: Is it really a safe assumption that gas aerosol thermodynamic equilibrium is met for inorganic ions during a dust storm? It would seem that many non-aqueous (i.e. solid) aerosol would be present that would not have normal equilibrium partitioning. It would be nice to see some evidence of this. This would also help support the conclusion that $Ca(NO_3)_2$ and $CaSO_4$ are negligible.
- Line 282: The presence of Cu, brings to mind the question of transition metal ions and industrial sources of metal containing particles. How were these accounted for? Particularly since they often have different properties and propensity for generating ROS as Weber and company at Georgia Tech have shown.
- It should be noted that there is a great deal of uncertainty regarding aerosol pH, particularly in North China, with estimates ranging from 3-7 pH units. This of course will affect nitrate. The authors could comment on this with respect to their data, though keeping in mind Hennigan et al showing the proxy methods such as $NH_4^+/(NO_3^-+SO_4^{2-})$ are qualitative at best.
- What is the mineralogy of the Hunshandake Desert? Is it rich in $CaCO_3$? Based on a few assumptions made, documentation of the presence of this mineral from aerosols in the region would be helpful. Perhaps some of Ro and co-workers analysis of transported dust with SEM-EDX?
- Line 321 some evidence for "humid marine conditions might have enhanced particle-particle coagulation" would be helpful. The number concentrations in the marine boundary layer are unlikely to be > $10^5$ #/cm$^3$ where coagulation is prevalent, more likely in the $10^2-10^3$ #/cm$^3$. Are the authors referring to fog-processing? That would seem to be the primary way this could happen in a marine environment.
- Line 326 The line "ammonium salts mostly co-existed with dust aerosols externally" is confusing as written. Is the population externally mixed with respect to ammonium nitrate and dust? Or are the salts co-existing with dust, but not other particle types? Please rephrase for clarity.

- Overall many of the conclusions on page 12 appear to mostly be speculation with little data to support it. I would recommend sticking to conclusions with more support from the data in the paper.
- Line 357: The source profile for coal, could it have dust mixed in? When the author's say that there is a "mixture of coal combustion and other pollutants" are they saying that they are internally mixed or simply present contemporaneously? Clarifying that point would be helpful.
- Overall the Figures could use improvement as portions are hard to read and the take home point of each is not always clear. It seems at times as if the authors are simply showing everything they can, as opposed to targeting their figure to the main points of the paper.

---

## Referee Report (RR3)

General Comments: First of all, the space was not inserted appropriately in many parts, so it is hard to read and follow. Such a crude revision with low presentation quality should not be sent to reviewers. I have reviewed again this manuscript and found some improvements on the manuscript; however, many replies have not been found in the revised manuscript and/or replied well to my concerns. I feel that the presentation quality is still low as to be published from the high quality journal of Atmospheric Chemistry and Physics. With regret, I have judged to reject this manuscript again.

Response:We are sorry for the space missing problem. We have a double check for our submitted version, but not find this problem. We therefore re-install our software and avoid the problem in the new submission. In the new version, we further improve the quality of the manuscript according to two reviewers' comments. We are confident that it is ready for publishing in a high quality journal.

The authors have made fundamental revisions in this revision, and I have partly understood and agreed the responses. Although this manuscript was improved compared to the first stage, I feel that the current presentation quality and scientific promotion is remained not so high. I have finally judged that this manuscript will not be accepted.

Q1. I have partly understood my concerning issue regarding the definition of dust event. Further concerning issue is the sampling duration of continuous dust event. Even the dust event continued multiple days, how should we consider the representativeness of the sampling? For instance, sample 20080528 and 20080529 (please note that the sampling time of 20080529 will have typo) had approximately one day interval. Was there large temporal variation of PM10 concentration during continuous dust days? If there was large change on PM10 concentration, why the authors collected on the listed time? The authors should state the reason, and should present the representativeness of 4 hrs sampling. In the revised manuscript, it will be kind for readers to explicitly state that 'http://www-cfors.nies.go.jp/~cfors/' is for forecast model over Asia, and 'http://www.qepb.gov.cn/m2/' is for observed concentration at Qingdao.

Response: Due to no dust events lasting over 12 hrs (Lee et al., 2015; Su et al., 2017; Zhang et al., 2007), we collected one dust sample with a 4-hr duration in a day. The sampling for dust particles started only when the measured PM10 mass concentration in Qingdao (http://www.qepb.gov.cn/m2/) and the forecasted dust mass over Asia (http://www-cfors.nies.go.jp/~cfors/) had greatly increased. The samples with ID of 20080528 and 20080529 were subject to two different dust events occurring in two days instead of continuous samples for one dust event. On March 20-21, 2010, two dust events subsequently swept Qingdao. The 4 hr dust samples with IDs of 20100320 and 20100321 may not capture the entirety of the two events. However, the on-line data can allow adequate separation of the two dust event samples.

The same was true for the dust samples with IDs of 20110501, 20110502. The link illustration for these two links has been also added properly in the new version.

From this response, I can partly understand the reason for the representativeness of 4 hrs sampling. However, it seems very hard to follow the revised manuscript on L90-93. What is the evidence of "adequate separation"? The explicit reason is ambiguous here. Are these based on the three sampling category defined in this study (Section 3.2)? If so, it is not appropriate to mention here, because this is the section for methodology.

I have further questioned on the separation of the three categories used in this study really divide the dust samples. This is because three categories were based on the concentration of inorganic nitrogen, which easily decomposed into gas-phase depending on the atmospheric circumstances.

Q2. I have partly agreed, but I have further question on the application of a 3-D air quality model. First, what is the merit of the application of 3-D air quality model? In the revised manuscript, only the spatial distributions of PM10 were shown (Fig. 2 from CFORS model and Figs. S1-S3 with CFORS and WRF-CMA. Can such application reinforce the authors' discussion points? The behavior of IN were discussed in this manuscript, so what is the purpose to show PM10? The authors stated that 'The spatial distribution of PM10 concentrations for each dust event was consistent with the model results of dust by the Chemical Weather Forecast System (CFORS) by Uno et al. (2003)' (L199-201). If the consistency between other models is important, why the author calculated on your own model? I cannot follow this reason from the revised manuscript.

Response: Thank you for the suggestion. We have deleted the results by CFORS. The CMAQ model (v5.0.2) was applied to simulate the concentration of PM10, NOx, NH3, NO3- and NH4+ over the East Asia area for aerosol samples on dust and comparison days. We have revised the discussion on model results. Distribution of PM10 was used to characterize the dust events. Spatial distributions of PM10 during each dust events were consistent with the records in the "Sand-dust Weather Almanac" (CMA, 2009; 2010; 2012; 2013). The model results indicated that CMAQ results reasonably reproduce the mass concentrations of NO3- (Fig. S6). Simulated NH4+ concentrations in dust samples were severely under-predicted with NMB values at -71%. For reference samples, simulated NH4+ concentrations sometimes can well reproduce the observational values, but sometimes totally off. The external mixing mechanism proposed in this study is urgently needed to be included in the model for accurately predicting the concentrations during dust events.

First of all, I feel that it is not appropriate to use supporting materials frequently in the main text. The main and supporting texts should be understood with itself as a standalone. Without any speciation for CMAQ modeling results, to mention NMB only seems to be not good.

I am not sure the exact reason can be really attributed the external mixing mechanism in the model.

How about the evaluation for the precursors of NOx and NH3? The discussions on L331-L340 were only depends on the model results. Moreover, how can the authors' consider the model assumption of anthropogenic emission status on 2008? As the author's showed in Fig. S3, NOx and NH3 emissions in China will largely change from 2008 status to 2011. Again, if the authors' conclude that the external mixing, I have still questioned on the importance of intensity of dust event. In my opinion, Ca concentration will highly depends on the dust intensity, and decide the external status.

Q3. The following specific points also should be revised to clarify the model application.

L189: Centered point is needed because we cannot follow the modeling domain at the current description.

L193: On the INTEX-B emission inventory (Zhang et al., 2009), I suppose that NH3 emissions have not been provided. If so, this description should be changed.

L195, and Figures 6 and 8: So, all calculations were based on the emission level on 2008? Because the temporal resolution of INTEX-B emission inventory is month, I feel that there are no need to display all emissions on all dust samples. These emissions level should be differed only on month. Therefore, I suppose that the averaged (spring time) emissions of NOx and NH3 on each one figure is enough.

Response: We have supplemented the centered point (110°E, 34°N) in the new version. According to the publications of INTEX-B and TRACE-P Asia emission inventories (Zhang Q et al., 2009; Streets et al., 2003.), INTEX-B inventory was developed based on TRACE-P inventory with NH3 emission considered (the annual emission amount of NH3 in China was 13.6 Tg). However, due to the low priority and low variability of NH3 emission during 2000-2006, NH3 emission was not updated in INTEX-B inventory, and the NH3 emission in INTEX-B inventory was consistent with TRACE-P.

Agree and revised.

Although I have confirmed this revision, please see Q2 for questions on this assumption.

Q4. Figure S1: What is the purpose to show the difference between (b) and (c)? In this caption, what is 'WRF-CMA'?

Response: We have indicated that one exterior dust sample was collected on 1 March when no dust was recorded in Qingdao by MICAPS. However, the MICPAS information over the whole country indeed showed the dust events in China on 1 March. And the modeled spatial distribution of PM10 and TSP mass concentration for this dust event on 1 March implied that the sample should be classified into dust sample. Therefore we listed all the supporting figures in Fig. S1. Fig. S1 (b) was the weather information from the MICAPS at 8:00 on Mar.2, 2008 and (c) was hourly PM10

concentration modeled by the WRF-CMAQ model at 15:00 on Mar.1, 2008. We guessed that the reviewer maybe refer to the difference between (c) and (d), therefore we deleted (d).

We have revised the caption.

I have confirmed this revision for Fig. S1.

Q5. Figure S3: In the main manuscript, it was stated that 'each dust sampling day are shown' in Fig. S3 (L218-219, L895). However, only the hourly concentration of PM10 concentration at 14:00 on 19 Mar 2011 were shown. Please confirm this supplemental figure.

Response: We really modeled the PM10 concentration on each dust sampling day, but only showed the PM10 concentration at the middle time of the sampling in Fig.S3 (Now Fig.S5 in the new version) due to too many figures. We have revised this section and the sentence has been revised into "The concentrations of PM10 and its major components NO3- and NH4+ over East Asia on dust days and comparison days were modeled using the WRF-CMAQ model (Fig. S5-6)" in L341-342 in the new version.

I have confirmed this revision for new Fig. S5.

Specific comments:

Q6. L35-36: This conclusion does not match to the manuscript contents. The authors stated that input of nitrogen to the ocean depends on the dust events.

Re-comment: cannot find this revision.

Response: The revision in the last round was prepared after the quick response. After a careful consideration, we agreed with the comment and delete the part in the last revised version.

I have confirmed this point. However, when I have read the abstract at the current version, the readers cannot distinguish the category 2 and 3, but only the category 2 was mentioned on last sentence. The abstract should be re-organized.

Q7. I have confirmed and understood the meaning. However, is this revision corresponded to L226-230? If so, this revised sentence seems to contain many errors (NOT Table S2 but Table S1?). For example, we can find 410 µg/m3 on dust day sample on 20080315. What is the value of 80-1303%? These increased value were not corresponded to 'Ratio of DD to CS' shown in Table S1.

Response: Yes, this revision corresponded to L226-230 in last version. And the times of dust to non-dust day samples were replaced by the ratio according to the former suggestion. To avoid the confusion, we have revised Table S1 to give the increased ratio.

I have confirmed this revision for Table S1.

Q8.L171: Again, I cannot follow the calculation of "1.7-21.9 times (mean: 6.9)".

L175: I cannot follow "10.3 times" for Fe. It can be calculated as 7.90 from the values in Table 2.

L176: "3.6-fold" will not be followed from Fig. 2. It should be listed in Table 2.

Re-comment: So, in this revised manuscript, these statements of the increment ratio on dust-day compared to non-dust day have not been explicitly appeared. In L243, the authors stated 'Table S1', but Table S1 contained not only the information of inorganic nitrogen but also TSP, Al, Fe, and nss-Ca. So, it is appropriate to mention on Table S1 in Section 3.1.

Response: Agree and revised.

In the revised Table S1, the averaged value was not shown; hence it is partly hard to follow L202 and L204. In the last part of Table S1, it is explained as "Mean ratio of all samples on dust days", however, in L205, it is stated that "a median value of 403%". In the analysis, mean and median will cause important differences. This should be clearly used.

Q9.L175: So, please state explicitly regarding this point to the readers. In the current form, nss-Ca was suddenly shown in Fig. 3 without any introduction.

Response: Agree and revised.

I have confirmed this point in new Fig. 2.

Q10. First of all, I cannot find the revision of 'The concentrations of ammonium were increased by 20' anywhere. Is this corresponded to Table S1? I suppose that the authors discussed regarding this point in L240-L244. Although ratio was shown in Table S1. percentages are discussed here. So it is hard to follow the manuscript. Why the discussion point have not been arranged on the uniformed unit?

Response: Agree and revised.

I have confirmed in Table S1.

Q11.L194-L195: In this sentence, the authors stated "the effect of dust on inorganic nitrogen differed during different types of dust events". Why the authors suddenly focused on inorganic nitrogen here? In L192-193, it was mentioned "inorganic ion $SO_4^{2-}$ exhibited concentration variations that were similar to those of nitrate".

L197: The figures for inorganic nitrate will be helpful information here, if the authors focused on inorganic nitrogen.

Re-comments: I cannot find this revision.

Response: The revision in the last round was prepared after the quick response. After a careful

consideration, we completely rewrote the part to avoid confusion in the last revised version.

I still cannot find the explicit reason to focus on the inorganic nitrogen in the revised manuscript. Why the inorganic nitrogen was focused in this study? This will be an important statement.

Q12. L207: (respectively less than 50 ug/g and 6 ug/g) will be the correct expression for ammonium. Re-comments: I cannot find this revision.

Response: The revision in the last round was prepared after the quick response. After a careful consideration, we completely rewrote the part into "The ratios of mass concentrations of nitrate and ammonium to the total mass of sand particles were very low, i.e., less than 81μg/g, which are approximately three orders of magnitude less than the corresponding values in our dust samples." at L230-232.

I have confirmed.

Q13.I am wondering that the differences of IN concentration between Duolun and Zhurihe. Both are Hunshandadke Desert, however, as is shown in Table 5, IN concentration was much higher in Duolun. Are there some emission source?

Response: Sand samples were collected at a remote site in Zhurihe desert. Little anthropogenic influence is expected. Atmospheric aerosol samples were collected at an urban site in Duolun on dust days for comparison. It is not surprised for a strong signal for anthropogenic sources. This has been clarified in the new version.

I have confirmed in Table 5.

Q14. Again, only from the dust spatial distribution, it is hard to state the dust intensity.

Response: We had made a substantial revision on the part in the last round revision and didn't consider dust intensity as an important factor for our unique results.

Q15. L214-L216: Without more information of the intensity of dust, the discussion on 'dilution effect' seems to be lacked in scientific understanding. This part should be fully revised based on not only dilution effect but also dust intensity.

Re-comments: Again, only from the dust spatial distribution, it is hard to state the dust intensity.

Response: We had made a substantial revision on the part in the last round revision and didn't consider dust intensity as an important factor for our unique results.

I have checked that the discussion on dust intensity was fully removed. Please see Q2.

Q16. I have confirmed the revision, but if the authors discussed on average (L300-303), the averaged values were also needed.

Response: We really had given the average of TSP in form of average±standard deviations at L300-303 in last revision. Now we had made a substantial revision on this part and didn't discuss TSP average concentration.

I have checked that the statement on TSP with Table 4 was fully removed.

Q17. L227-L228: The favorable condition to form ammonium cannot be discussed without the information of NH3. In addition, Table 3 indicated the aerosol samples in the coastal region of the Yellow Sea. How about the status over air mass path? Is it sufficient to conclude only from the downwind information to the formation of inorganic nitrogen?

Re-comments: Again, I cannot understand the model application results.

Response: We modeled the emission and concentration of NOx and NH3 over East Asia on the dust and comparison days. The model results showed that the calculated trajectories of the entire dust air mass passed over those highly polluted regions with strong emissions of NOx and NH3 shown in Fig 6 and experienced different residence times therein. The average concentration of NOx and NH3 during transport were calculated and discussed according to Categories 1 and 2. The air masses in Category 1 took over 11-39 hrs to cross over the highly polluted area with appreciable concentrations of NOx (5.7±1.4 ppb) and NH3 (7.6±3.3 ppb). Except for the exterior samples, air masses in Category 2 took less than 10 hrs to cross over the polluted areas with lower concentrations of NOx (3.6±3.4 ppb) and NH3 (4.7±4.7 ppb) and the mixing layer height along the route was 916-1194 m (on average) for each dust event. This further led to the external mixing of anthropogenic particulate matters and dust.

Although I can partly agree these kinds of analysis, please see Q2.

Q18. L230: "strong dust storm" cannot be discussed without any information on dust intensity here.

Re-comments: Again, from the additional information of CFORS, the spatial distribution pattern was found; however, how can we estimate the intensity?

Response: We had made a substantial revision on the part in the last round revision and didn't consider dust intensity as an important factor for our unique results.

Why the dust intensity is not an important factor? What results indicates this conclusion? Please see Q2.

Q19. L233-L234: But NOx concentration was high in Case 3. I cannot follow why the authors concluded on Case 3.

Re-comments: So where did the authors discussed the NOx concentration in the manuscript?

Response: We had made a substantial revision on the part in the last round revision after the quick response. The NOx concentration was discussed in Section 4.3 in the new revision.

I have confirmed this revision in Section 4.3.

Q20. L254-L255: The authors simply mentioned "local emissions" here. Because the samples were collected on downwind regions in the coastal region of the Yellow Sea, I guess that the discussion on emission characteristics of each (or, at least, some categorized) air mass should be discussed in detail. The inorganic nitrogen concentrations are highly related to the local conditions both on emissions strength and meteorological parameters, so the discussion only on air mass speed and air mass path over ocean are insufficient.

Re-comments: Again, model is used only for spatial distribution and not inform the chemical production process.

Response: We had made a substantial revision according to the suggestion. The chemical production process was discussed in Section 4.1 "Theoretical analysis of the three categories". In Category 1, ammonium salt aerosols may externally exist with dust aerosols in these dust day samples and $NO_3^-$ and $SO_4^{2-}$ were almost completely associated with $NH_4^+$ in these dust day samples; whereas a larger fraction of $NO_3^- + SO_4^{2-}$ may exist as metal salts due to reactions of their precursors with dust aerosols in Category 2. The simulated $NO_3^-$ and $NH_4^+$ concentrations was compared with the observation in Qingdao, and the results indicated that the external mixing mechanism proposed in this study is urgently needed to be included in the model for accurately predicting the concentrations during dust events.

I have confirmed this revision in Section 4.1.

Q21. I have rechecked the discussion of backward trajectories discussed on Section 3.4. There are many points should be clarified.

Figure 5: Please add the explanation of the trajectory of 20110415 was excluded based on the discussion on Fig. 2. Why the authors displayed "non-dust samples"? What were the differences between non-dust and dust samples trajectories? I feel that these were similar.

Response: Agree and revised.

I have suggested that the reason or short notice is needed in Fig. 5.

Q22. L314: What is the 'remaining one'? Please specify the trajectory data. In my opinion, two trajectories of 20110418 and 20110501 originated from northeast China.

Response: Yes, trajectory 20110501 was really from northeast China, however it then passed over the Inner Mongolia, and arrived at Qingdao from north, just like 20110502. Therefore, we grouped the trajectory into the air mass originated from Inner Mongolia, China. However, it was really easy to mislead the readers. Therefore, we accepted the suggestion, and revised the sentences into "The calculated air mass trajectories for 13 out of 14 samples showed that the air mass originated from North and Inner Mongolia, China (Fig. 5), generally consistent with the results of Zhang and Gao (2007). The remaining one, with ID of 20110418 originated from Northeast China.".

Q23. L317: What is the 'one exterior sample'? Please specify.

Response: Agree and revised.

I have confirmed this revision.

Q24. L319: I cannot see 'the air masses crossed over the sea for 94-255km' from Figure 5, because Fig. 5 showed the whole view of trajectories across China. More detailed figure or explanation will be required.

Response: The distance over sea of the air mass for each sample was measured from the trajectory using TrajStat software (Wang et al., 2009). We have added the explanation in Section 2.3.

Wang, Y. Q., Zhang, X. Y., and Draxler, R. R.: TrajStat: GIS-based software that uses various trajectory statistical analysis methods to identify potential sources from long-term air pollution measurement data, Environ. Modell. Softw., 24, 938-939, 2009

I have understood this methodology.

Q25. L328-329: What is the definition of the 'average mixing layer'. I suppose that the altitude of backward trajectories were so high because most of trajectories were originated outside China on 72 hrs. So, where is the averaged region to calculate '900m' in this sentence?

Response: The average mixing layer was calculated as an average of all points on the air mass back trajectory of each sample. This has been clarified in section 2.4 in the new version.

I have understood this methodology.

Q26. I cannot still understand the authors' conclusion here. As was discussed on L355-361, Fig. 8, and Table 7, although the coal combustion have increased on dust days, the contributions of local anthropogenic sources (especially secondary aerosols) have decreased on dust days. According to the discussion on Section 3.3, the authors concluded that ammonium salts were externally co-exist with dust aerosols in Category 1. So, why the contribution of secondary aerosols were decreased from PMF analysis. I feel that these results have contradicted. More careful discussion is required

for this conclusion.

Response: The source of coal combustion on dust days became complicated. "mixture of coal combustion and other pollutants" means these compounds present contemporaneously, because that PMF model can't show the mixing or existing state. We have revised the sentence into "The source profile for coal combustion in dust day samples showed a high percentage of K+, Cl-, Ca, Mg, Co, Ni, As, Al and Fe, indicating coal combustion presenting contemporaneously with other pollutants emitted along the transport path on dust days.". Ammonium salts were externally co-exist with dust aerosols in Category 1, but showed lower concentrations in Category 2 likely due to unfavorable conditions for forming ammonium salts. Here the conclusion was a result of source appointment for all dust samples including Category 1 to 3. And we have revised the sentence "In these dust samples, including Categories 1-3, oil combustion, industry, soil dust, secondary aerosols, and coal combustion/other sources were identified as five major sources (Table 6).

I have confirmed this revision.

Q27.L306: I cannot follow the calculation of "a factor of 1.1-5.8" and "a factor of 1.8-6.3".

Re-Comments: I cannot find this revision.

Response: We had made a revision on these sentences in the last round revision after the quick response. According to the former suggestion, we changed the factor to ratios. And this sentence was revised to "In Category 1, the dry deposition fluxes of NNH4++NO3- increased by 9-75% with increased TSP flux by 86-252% (Table S3)" at L371-372 in the new revision. And we also revised Table S3 to give increased proportion and the calculation method.

I have confirmed this revision in Table S3.

Q28. L309: What is the calculation method of "63%" and "46%"?

L310: What is the calculation method of "14%" ?

Re-Comments: I cannot find this revision.

Response: We had made a revision on these sentences in the last round revision after the quick response. We have revised Table S3 to give increased proportion and the calculation method.

I have confirmed this revision in Table S3.

Q29. L317: For Fe, it seems that the increased ratio were 2.81-11.08 from Table S3.

Response: This sentence has been revised to "However, the dry atmospheric deposition fluxes of Fe increased by a factor of 124-2370% in dust day samples." at L383-384 in new revision.

I have confirmed this revision in Table S3.

Specific comments:

Q30.Table 6: Missing the note of a and b.

Response: Done.

I have confirmed this revision in new Table 7.

Q31. Table S2: Please align the right-column, it is hard to follow. What is the meaning of *?

Response: Agree and revised.

I have confirmed this revision in Table S3.

---

## Editor Decision (ED1)

**Editor:**

In the copy of the response to the review by Referee #1, I replaced all comments that have been addressed satisfactorily by *[…]*
It seems to me that multiple authors might have worked on the revision of the manuscript and that the response to the referees is based on an early revised version. Several of your responses do not match the submitted revised manuscript, as pointed out at several places by Referee #1.

In addition to addressing all scientific comments below, please make sure that in the next revised version:
- spaces are inserted where needed (e.g. l. 99, 128, 139)
- response to reviews and changes in manuscript are consistent
- proofread carefully the manuscript for typos and grammar errors

**Referee #1 – Re-Review:**
**Responses to revision in red**

General Comments:
The manuscript titled 'The concentration, source apportionment and deposition flux of atmospheric particulate inorganic nitrogen during dust events' written by Jianhua Qi presented the dust impacts on particulate inorganic nitrogen by analyzing the aerosol samples collected at Qingdao, China. The authors divided dust pattern into three parts, and investigated the dry deposition flux. To estimate the source, PMF receptor model was also used. Based on the above approaches, the authors tried to answer the questions of 'dust event always increase the atmospheric input of nitrogen to the ocean?'. The topic is interested ones because the impact of dust as atmospheric input on ocean ecosystem has been still unclarified. However, throughout the manuscript, it is not well organized and hard to follow and understand. Overall, this manuscript will not be acceptable taking into account the high journal quality of Atmospheric Chemistry and Physics.
Reply: We will revise the manuscript according to the comments to improve the manuscript quality.
First of all, the space was not inserted appropriately in many parts, so it is hard to read and follow. Such a crude revision with low presentation quality should not be sent to reviewers.
I have reviewed again this manuscript and found some improvements on the manuscript; however, many replies have not been found in the revised manuscript and/or replied well to my concerns. I feel that the presentation quality is still low as to be published from the high quality journal of Atmospheric Chemistry and Physics. With regret, I have judged to reject this manuscript again.

Q1. Before the discussion, first, the definition of "dust events" cannot be understood well. In L99-101, the authors explained that 'Samples were collected on dust days and selected ND days in spring from March 2008 to May 2011, with sampling duration of 4h for each sample. We refer to the ND days as sunny and cloudy days before and after dust events in the following discussion'. The authors should add the appropriate reference of the Meteorological Information Comprehensive Analysis and Process System (MICAPS) which defined the weather conditions (and also, the subsection 2.4 should be reorganized partly into this explanation). What is the

definition of "dust events" here? Visibility? More information of how the dust events are defined in this system should be announced in detail. Total of 14 samples (sample numbers in Table 3) during dust events were analyzed throughout this study. The sampling duration was 4 hrs, so which data are used in the corresponded date in Table 3? All samples in the day? Moreover, what is the sample numbers of ND? The current information in Section 2.1 is severely lacked in the information which the readers can follow the authors methodology. Because this study discussed the dust impact, the explicit and detailed information regarding dust is required. In this sentence, I am worried about the explicit division of dust and non-dust samples. It is well known that some dust events are continued a few days. For example, the samples used in this study during 28-29 May 2008, 20-21 March 2010, 15 and 18 April 2011, and 1-2 May 2011 showed continuous dust events. In such cases, do the authors have confidence to the clear separation of dust and non-dust samples? How about the Al concentration definition (L171-172) of non-dust days samples? Why were other days samples not collected to clearly separate the dust impacts? The definition of ND is ambiguous. According to the definitions of dust and non-dust, the discussion on dust impact might be changed. The reconsideration of dust impact is needed based on the clear definitions of dust.

Reply: In this study, the dust event was defined according the definition adopted in regulations of surface meteorological observation of China (CMA, 2003; Wang et al., 2008) and identified based on the meteorological records information from Meteorological Information Comprehensive Analysis and Process System (MICAPS) of China Meteorological Administration. Each dust sample was collected for 4hrs duration and the sampling started only when the PM10 mass concentration available on the website (http://www-cfors.nies.go.jp/~cfors/; http://www.qepb.gov.cn/m2/) was increased greatly. The approach made the dust sample more representative relative to urban background. However, for dust event with duration less than one day, only one sample was collected; for dust event with longer duration, i.e. multiple days, the sample was collected once a day. The sampling information was listed in the Table S1. Based on the forecast, we also collected aerosol particle samples immediately before or after the dust event for comparison. These comparison samples were further classified into sunny day samples, cloudy day samples and post-dust samples. The post-dust samples were featured by collecting under a clear and sunny weather condition and lower mass concentration of PM10. Moreover, the concentration of Al referring to the total Al concentration in TSP samples were used to confirm the division of dust or comparison samples according to the criterion "geometric mean$\times$2GSD" proposed by Hsu et al. (2008). CMA: Regulations of Surface Meteorological Observation, China Meteorological Press, Beijing, 154–156, 2004. Hsu, S. C., Liu, S. C., Huang, Y. T., Lung, S. C. C., Tsai, F., Tu, J. Y., and Kao, S. J.: A criterion for identifying Asian dust events based on Al concentration data collected from northern Taiwan between 2002 and early 2007, Journal of Geophysical Research Atmospheres, 113, 1044-1044, 2008. Wang Y. Q., Zhang X. Y., Gong S. L., Zhou C. H., Hu X. Q., Liu H. L., Niu T., Yang Y. Q.: Surface observation of sand and dust storm in East Asia and its application in CUACE/Dust, Atmos. Chem. Phys., 8, 545–553, 2008.

I have partly understood my concerning issue regarding the definition of dust event. Further concerning issue is the sampling duration of continuous dust event. Even the dust event continued multiple days, how should we consider the representativeness of the sampling? For instance, sample 20080528 and 20080529 (please note that the sampling time of 20080529 will have typo) had approximately one day interval. Was there large temporal variation of PM10 concentration

during continuous dust days? If there was large change on PM10 concentration, why the authors collected on the listed time? The authors should state the reason, and should present the representativeness of 4 hrs sampling. In the revised manuscript, it will be kind for readers to explicitly state that 'http://www-cfors.nies.go.jp/~cfors/' is for forecast model over Asia, and 'http://www.qepb.gov.cn/m2/' is for observed concentration at Qingdao.

Q2.The second concern is the "dilution effect" which the authors claimed as the key factor for the discussion of inorganic nitrogen. Again, without the explicit definition of dust and non-dust, the dilution effect cannot be understood well. In this discussion, although the authors introduced the air mass speed, there were no implications on the intensity of dust events itself. Why the upwind (i.e., near desert) information was not used here to describe the dust intensity? The dilution is not so simple, hence more information are required to reinforce the authors finding. The authors discussed the inorganic nitrogen behavior. In these cases, what is the counter ion of $NH_4^+$ and $NO_3^-$? Are the main counter ions metal elements? If $NH_4NO_3$ are formed, due to its chemical unstablity according to the temperature and relative humidity, it is not simple to discuss only the viewpoint of "dilution effect". In addition, the authors used $NO_2$ data to investigate the inorganic nitrogen, but how about $NH_3$? Only from $NO_2$ data, it is insufficient to estimate the inorganic nitrogen variation. On the above reasons, the reconsideration is required to publish this manuscript from Atmospheric Chemistry and Physics.

Reply: In revision, the part reads as "Inorganic nitrogen (IN) concentrations highly varied in different dust samples (Table 3). According to the concentrations relative to those in comparison samples, they can be classified into three categories, i.e., Category 1 in which higher IN concentrations were observed in dust samples, Category 2 in which lower IN concentrations were observed in dust samples, and Category 3 in which lower nitrate concentrations with slightly higher concentrations of ammonium in dust samples. Category 1 was usually associated with a lower moving speed of dust air mass or a longer distance over the ocean (Table 5) while the reverse was true for Category 2. The moving speed and distance over the ocean of dust air mass in Category 3 was generally between them. Theoretically, lower moving speed of dust air mass favors reactions between dust particles and anthropogenic gaseous precursors of IN due to a longer reaction time. Large moving speed of dust air mass was frequently associated with a large wind speed in the lower layer atmosphere (Gao et al., 2010; Gillette and Passi, 1988; Peng et al., 2007; Yue et al., 2008), leading to anthropogenic gaseous precursors therein to be better diluted. Shorter reaction time and reduced concentrations of anthropogenic gaseous precursors likely lowered IN in Category 2. Moreover, the relative concentration of IN per aerosol particle mass in μg/g was analyzed and compared with those values in literature. .."It is questionable for using NOx observed in Qingdao to argue the generation of IN in dust samples. We agree this because most of IN observed in dust samples should be derived from secondary reactions upwind of Qingdao by considering a low conversion rate of NOx to IN. The former study (Liu et al., 2010) showed that NOx and $NH_3$ generally capture the spatial distribution patterns with high values over eastern China and relatively lower values over central and western China, where dust source regions are

located (Fig. S1-S3). Thus, we will add modeling results using a 3-D air quality model to support our analysis in revision.

Gao, Q X., Ren Z H. et al. : Dust events and its impacts on atmospheric environment, Science press, Beijing, 2010.

Gillett e D A, Passi R.: Modeling dust emission caused by wind erosion, J G R., 1988, 93: 14234-14242.

Liu X. H., Zhang Y., Cheng S. H., Xing J., Zhang Q., Streets D. G., Jang C., Wang W. X., Hao J. M.: Understanding of regional air pollution over China using CMAQ, part I performance evaluation and seasonal variation, Atmospheric Environment , 44,2415-2426, 2010.

Peng, Z., Liu X. M., Hong Z. X., Wang B. L.: Characteristics of Atmospheric Boundary Layer Structure and Turbulent Flux Transfer during a Strong Dust Storm Weather Process over Beijing Area, Climatic and Environmental Research, 2007, 12(3): 268-276.

Qi J.H., Gao H.W., Yu L.M. , Qiao J.J.: Distribution of inorganic nitrogen-containing species in atmospheric particles from an island in the Yellow Sea, Atmospheric Research, 101,938-955, 2011.

Wang Y. Q., Zhang X. Y., Gong S. L., Zhou C. H., Hu X. Q., Liu H. L., Niu T., Yang Y. Q.: Surface observation of sand and dust storm in East Asia and its application in CUACE/Dust, Atmos. Chem. Phys., 8, 545–553, 2008.

Yue P., Niu S. J., Liu X. Y.: Dust Emission and Transmission during Spring Sand-dust Storm in Hunshandake Sand-land, Journal of Desert Research, 2008, 28(2): 227-230.

I have partly agreed, but I have further question on the application of a 3-D air quality model. First, what is the merit of the application of 3-D air quality model? In the revised manuscript, only the spatial distributions of PM10 were shown (Fig. 2 from CFORS model and Figs. S1-S3 with CFORS and WRF-CMA. Can such application reinforce the authors' discussion points? The behavior of IN were discussed in this manuscript, so what is the purpose to show PM10? The authors stated that 'The spatial distribution of PM10 concentrations for each dust event was consistent with the model results of dust by the Chemical Weather Forecast System (CFORS) by Uno et al. (2003)' (L199-201). If the consistency between other models is important, why the author calculated on your own model? I cannot follow this reason from the revised manuscript. The following specific points also should be revised to clarify the model application. L189: Centered point is needed because we cannot follow the modeling domain at the current description. L193: On the INTEX-B emission inventory (Zhang et al., 2009), I suppose that NH3 emissions have not been provided. If so, this description should be changed. L195, and Figures 6 and 8: So, all calculations were based on the emission level on 2008? Because the temporal resolution of INTEX-B emission inventory is month, I feel that there are no need to display all emissions on all dust samples. These emissions level should be differed only on month. Therefore, I suppose that the averaged (spring time) emissions of NOx and NH3 on each one figure is enough. Figure S1:

What is the purpose to show the difference between (b) and (c)? In this caption, what is 'WRF-CMA'? Figure S3: In the main manuscript, it was stated that 'each dust sampling day are shown' in Fig. S3 (L218-219, L895). However, only the hourly concentration of PM10 concentration at 14:00 on 19 Mar 2011 were shown. Please confirm this supplemental figure.

Specific comments:

Q3. L35-36: This conclusion does not match to the manuscript contents. The authors stated that input of nitrogen to the ocean depends on the dust events.

Reply: We apologize for the confusion in the revision. We will revise the abstract sentence into "The atmospheric input of nitrogen into the ocean depends on the dust events; dust deposition was an uncertain source of nitrogen for the ocean".

I cannot find this revision.

Q4. L57-L67: In this paragraph, the authors used "ND days" simply. However, this wording should be used carefully; because the definition of non-dust days will be different in each study. Please consider to carefully define this wording.

Reply: Thank you for this suggestion. To avoid confusion, we will use "non-dust storm days" according to the original reference in L57-L67.

I have confirmed that the authors use the wording of 'dust storm' in the introduction.

*[…]*

Q7. L165: I cannot follow the calculation of "1.8-14.0 times (mean: 5.9)". The mean concentration have not been stated for dust days.

Reply: Each sample on dust day had its corresponding non-dust sample (Table S2).The 1.8-14.0 times was calculated as a ratio of the TSP concentration on a given dust day to the values in the comparison samples. The concentration and the ratio of samples on dust days were listed in Table S2.

I have confirmed and understood the meaning. However, is this revision corresponded to L226-230? If so, this revised sentence seems to contain many errors (NOT Table S2 but Table S1?). For example, we can find 410 µg/m3 on dust day sample on 20080315. What is the value of 80-1303%? These increased value were not corresponded to 'Ratio of DD to CS' shown in Table S1.

[…]

Q9.L171: Again, I cannot follow the calculation of "1.7-21.9 times (mean: 6.9)".

Reply: We apologize for the confusion. The calculation method is the same as that for TSP (see the reply to Q7). The correct concentrations and the ratios of samples on dust days are listed in Table S2.

Q11.L175: I cannot follow "10.3 times" for Fe. It can be calculated as 7.90 from the values in Table 2.

Reply: The calculation method is the same as that for TSP (see the reply to Q7). The concentrations and corrected mean ratios of samples on dust days are listed in Table S2.

Q13.L176: "3.6-fold" will not be followed from Fig. 2. It should be listed in Table 2.

Reply: The calculation method is the same as that for TSP (see the reply to Q7). The concentrations and the corrected mean ratios of samples on dust days are listed in Table S2. So, in this revised manuscript, these statements of the increment ratio on dust-day compared to non-dust day have not been explicitly appeared. In L243, the authors stated 'Table S1', but Table S1 contained not only the information of inorganic nitrogen but also TSP, Al, Fe, and nss-Ca. So, it is appropriate to mention on Table S1 in Section 3.1.

*[...]*

Q12.L175: In Figure 2, nss-Ca was shown, but nss-Ca was not listed in Table 2. What is the authorsintention to introduce nss-Ca here?

Reply: Follow others' study, we calculated the EF of Ca in Table 2. The EFs of Ca on ND days indicated that Ca was affected by anthropogenic sources. nss-Ca usually was used as a typical dust index. Therefore we showed the nss-Cain Fig.2 and discussed the influence of dust on crustal elements using nss-Ca. So, please state explicitly regarding this point to the readers. In the current form, nss-Ca was suddenly shown in Fig. 3 without any introduction.

*[...]*

Q16.L189: What is the comparison method on some dust days? The sample date are shown in Figure 3, so why the authors explicitly mention the date? I cannot follow the calculation of "a factor of 1.2-5.7".

Reply: It will be revised as "The concentrations of ammonium were increased by 20"

Q17.L190: What means "less than 20% of that on ND days"? Averaged data over ND days? Reply: We apologize for the confusion. The sentence has been revised to read "The concentrations of ammonium were increased by 20.

Q18.L191: Again, what is the comparison method on some dust days? I cannot follow the calculation of "a factor of 1.4-9.2 ".

Reply: The calculation method is the same as that for ammonium (see the reply to Q16). The concentrations and the increasing factors of samples on dust days are listed in Table S2.

First of all, I cannot find the revision of 'The concentrations of ammonium were increased by 20' anywhere. Is this corresponded to Table S1? I suppose that the authors discussed regarding this point in L240-L244. Although ratio was shown in Table S1, percentages are discussed here. So it is hard to follow the manuscript. Why the discussion point have not been arranged on the uniformed unit?

Q19.L194-L195: In this sentence, the authors stated "the effect of dust on inorganic nitrogen differed during different types of dust events". Why the authors suddenly focused on inorganic nitrogen here? In L192-193, it was mentioned "inorganic ion $SO_4^{2-}$ exhibited concentration variations that were similar to those of nitrate". L197: The figures for inorganic nitrate will be helpful information here, if the authors focused on inorganic nitrogen.

Reply: The part will be revised as "Similar to ammonium, nitrate concentrations were sometimes increased by a factor of 1.4-9.2 relative to the comparison sample while they were decreased in others. Unlike substantially increased concentrations of crustal metal elements in dust samples, the concentrations of IN were likely determined by meteorological conditions as well as surface areas provided by dust particles."

I cannot find this revision.

Q20.L207: (respectively less than 50 ug/g and 6 ug/g) will be the correct expression for ammonium.

Reply: We have incorporated this suggestion.

I cannot find this revision.

Q21.L211: So what is the source of atmospheric particulate nitrogen? The location of Duolun and Zhurihe Sand Desert is very close.

Reply: Duolun and Zhurihe belong to the Hunshandake Desert in Inner Mongolia, one of the main Chinese sand deserts. According to studies, the Yellow Sea is mainly affected by dust storms from this sand source with a probability of 52 Zhang, Z K., and Gao, H.: The characteristics of Asian-dust storms during 2000–2002: From the source to the sea, Atmospheric Environment, 41, 9136-9145, 2007. Gao, Q X., Ren Z H.: Dust events and its impacts on atmospheric environment, Science press, Beijing, 2010.

I am wondering that the differences of IN concentration between Duolun and Zhurihe. Both are Hunshandadke Desert, however, as is shown in Table 5, IN concentration was much higher in Duolun. Are there some emission source?

Q22.L214-L216: Without more information of the intensity of dust, the discussion on 'dilution effect' seems to be lacked in scientific understanding. This part should be fully revised based on not only dilution effect but also dust intensity.

Reply: As discussed above, we will add modeling results of dust distribution to support our analysis in revision.

Again, only from the dust spatial distribution, it is hard to state the dust intensity.

[...]

Q24.L219: It seems that the discussion on this paragraph (e.g., "700 ug/m3 in Case 1" and "higher than 1100 ug/m3 in Cases 2 and 3") are based on Table 3. Please reorganize the paragraph, or please refer appropriate information here. It is hard to follow these values.

Reply: We will revise this paragraph and refer to the appropriate information in the revised manuscript according to revised Table 3.

I have confirmed the revision, but if the authors discussed on average (L300-303), the averaged values were also needed.

*[…]*

Q27.L227-L228: The favorable condition to form ammonium cannot be discussed without the information of NH3. In addition, Table 3 indicated the aerosol samples in the coastal region of the Yellow Sea. How about the status over air mass path? Is it sufficient to conclude only from the downwind information to the formation of inorganic nitrogen?

Reply: We will add modeling results using a 3-D air quality model to support our analysis in revision.

Again, I cannot understand the model application results.

Q28. L230: "strong dust storm" cannot be discussed without any information on dust intensity here.

Reply: We will add modeling results of dust distribution to support our analysis in revision. Again, from the additional information of CFORS, the spatial distribution pattern was found; however, how can we estimate the intensity?

Q29. L233-L234: But NOx concentration was high in Case 3. I cannot follow why the authors concluded "the strong dilution effect" on Case 3.

Reply: Among three cases, the NOx concentration was the highest with an average value of 70.7 for Case 3 and increased by 17.8

So where did the authors discussed the NOx concentration in the manuscript?

*[…]*

Q31.L254-L255: The authors simply mentioned "local emissions" here. Because the samples were collected on downwind regions in the coastal region of the Yellow Sea, I guess that the discussion on emission characteristics of each (or, at least, some categorized) air mass should be discussed in detail. The inorganic nitrogen concentrations are highly related to the local conditions both on emissions strength and meteorological parameters, so the discussion only on air mass speed and air mass path over ocean are insufficient.

Reply: As discussed above (see the Reply to Q2), We will add modeling results using a 3-D air quality model to support our analysis in revision.

Again, model is used only for spatial distribution and not inform the chemical production process.

Q32. L256: RH and NOx information are not shown in Table 5.

Reply: We apologize for the mistake. We have revised the title of Table 5.

I have confirmed the information of RH and NOx on the revised Table 6.

Q33.L260: The colors are overlapped, hence we cannot distinguish each trajectory. Some paths (e.g., thick green color: 2008/5/22 or 2011/4/15) are apparently indicated the west or south part of China. Are these events really related to dust events?

Reply: We apologize for the confusion. We have provided all trajectories of samples collected on dust and non-dust days. Fig.4 has been redrawn to distinguish each trajectory for samples collected on dust and non-dust days.

I have rechecked the discussion of backward trajectories discussed on Section 3.4. There are many points should be clarified. Figure 5: Please add the explanation of the trajectory of 20110415 was excluded based on the discussion on Fig. 2. Why the authors displayed "non-dust samples"? What were the differences between non-dust and dust samples trajectories? I feel that these were similar. L314: What is the 'remaining one'? Please specify the trajectory data. In my opinion, two trajectories of 20110418 and 20110501 originated from northeast China. L317: What is the 'one exterior sample'? Please specify. L319: I cannot see 'the air masses crossed over the sea for 94-255km' from Figure 5, because Fig. 5 showed the whole view of trajectories across China. More detailed figure or explanation will be required. L328-329: What is the definition of the 'average mixing layer'. I suppose that the altitude of backward trajectories were so high because most of trajectories were originated outside China on 72 hrs. So, where is the averaged region to calculate '900m' in this sentence?

Q34.L278-L280: The source of coal combustion have increased compared to non-dust days. Short explanation will be needed here.

Reply: The source of coal combustion on dust days became complex. The source profile showed high percentages of K+, Cl-, Ca, Mg, Co, Ni, As, Al and Fe, indicating a mixture of coal combustion and other pollutants emitted along the transmission path on dust days, such as industry and building dust. This source increased due to the coal combustion emissions mixing with other uncertain sources emitted into the air in strong winds.

I cannot still understand the authors' conclusion here. As was discussed on L355-361, Fig. 8, and Table 7, although the coal combustion have increased on dust days, the contributions of local anthropogenic sources (especially secondary aerosols) have decreased on dust days. According to the discussion on Section 3.3, the authors concluded that ammonium salts were externally co-exist with dust aerosols in Category 1. So, why the contribution of secondary aerosols were decreased from PMF analysis. I feel that these results have contradicted. More careful discussion is required for this conclusion.

*[…]*

Q37.L306: I cannot follow the calculation of "a factor of 1.1-5.8" and "a factor of 1.8-6.3".

Reply: These factors were the flux ratio of each dust sample in Case 1 to the ND average. The flux and ratio of each sample are listed in Table S3. We recalculated the increasing factors according to the revised values. The sentence was revised to read "Compared with the average flux on ND

days, the dry deposition flux of IN increased by a factor of 1.1-3.9, and the flux of atmospheric particles (TSP) increased by a factor of 1.8-6.3 in Case 1"

I cannot find this revision.

*[…]*

Q39. L309: What is the calculation method of "63%" and "46%"?

Reply: We apologize for the mistake. The sentence has been revised to read "Compared with the average dry deposition flux on ND days, the average nitrate flux of samples in Cases 2 and 3 decreased by 73

I cannot find this revision.

Q40. L310: What is the calculation method of "14%" ?

Reply: We corrected the calculation error and revised this sentence to read "Additionally, the average ammonium flux decreased by 47"

I cannot find this revision.

Q41. L317: I cannot follow the calculation of "a factor of 2-25".

Reply: The factor was calculated by comparing the flux of the sample on dust days with the average Fe flux on ND days (see Table S3).

For Fe, it seems that the increased ratio were 2.81-11.08 from Table S3.

Q42.L339: "aerosol particles" is "TSP"? In Table 7, please confirm the significant digits for each specie.

Reply: We apologize for the confusion. "aerosol particles" was revised to read "TSP". The former digits were revised according to the editor's suggestion. We will consider revising again to confirm the significant digits.

The revision for TSP was found in Table 9.

Technical Corrections:

*[…]*

Specific comments: Table 6: Missing the note of a and b. Table S2: Please align the right-column, it is hard to follow. What is the meaning of *?
* * *
**Referee #3:**

This manuscript focuses on an important issue, specifically the relationship between ammonium and nitrate with mineral dust in China. The variability in previously reported relative concentration trends is worth exploring in detail. There are a number of areas where the paper could use improvement prior to publication, particularly with connecting to the literature and placing the work in context.

Comments:
- Throughout the manuscript there are odd spacing issues, where two words are together without a space. As an example in the abstract *"For these two groups, NH4+in dust day samples waspresent in the form of ammonium salts externally co-existing with dust aerosols or the residual of incomplete reactions between ammonium salt and carbonate salts."*

- Line 51-54: The authors state the Asian dust has been transported as far as the north pacific, but this understates what has been observed for Asian Dust. Uno et al 2009 Nat Geosci showed that Asian dust can circumnagivate the globe. VanCuren and Cahill 2002 showed Asian dust impacting California air quality, while Ault et al. 2011 JGR and Creamean 2013 Science showed impacts of Asian dust on orographic precipitation in the Sierra Nevada (in California). Pratt et al. 2009 Nat Geo showed Asian dust influencing clouds over Wyoming.

- Though the authors note that a native English speaker was utilized for the revision, a considerable improvement in the grammar and proofreading are needed before the writing is at a publishable level.

- Line 233 insert comma after "samples"
- Line 266: Is it really a safe assumption that gas aerosol thermodynamic equilibrium is met for inorganic ions during a dust storm? It would seem that many non-aqueous (i.e. solid) aerosol would be present that would not have normal equilibrium partitioning. It would be nice to see some evidence of this. This would also help support the conclusion that Ca(NO3)2 and CaSO4 are negligible.

- Line 282: The presence of Cu, brings to mind the question of transition metal ions and industrial sources of metal containing particles. How were these accounted for? Particularly since they often have different properties and propensity for generating ROS as Weber and company at Georgia Tech have shown.

- It should be noted that there is a great deal of uncertainty regarding aerosol pH, particularly in North China, with estimates ranging from 3-7 pH units. This of course will affect nitrate. The authors could comment on this with respect to their data, though keeping in mind Hennigan et al showing the proxy methods such as NH4 +/(NO3 -+SO42-) are qualitative at best.

- What is the mineralogy of the Hunshandake Desert? Is it rich in CaCO3? Based on a few assumptions made, documentation of the presence of this mineral from aerosols in the region would be helpful.
Perhaps some of Ro and co-workers analysis of transported dust with SEM-EDX?

- Line 321 some evidence for "humid marine conditions might have enhanced particle-particle coagulation" would be helpful. The number concentrations in the marine boundary layer are unlikely to be > 105 #/cm3 where coagulation is prevalent, more likely in the 102—103 #/cm3. Are the authors referring to fog-processing? That would seem to be the primary way this could happen in a marine environment.

- Line 326 The line "ammonium salts mostly co-existed with dust aerosols externally" is confusing as written. Is the population externally mixed with respect to ammonium nitrate and dust? Or are the salts co-existing with dust, but not other particle types? Please rephrase for clarity.

- Overall many of the conclusions on page 12 appear to mostly be speculation with little data to support it. I would recommend sticking to conclusions with more support from the data in the paper.

- Line 357: The source profile for coal, could it have dust mixed in? When the author's say that there is a "mixture of coal combustion and other pollutants" are they saying that they are internally mixed or simply present contemporaneously? Clarifying that point would be helpful.

- Overall the Figures could use improvement as portions are hard to read and the take home point of each is not always clear. It seems at times as if the authors are simply showing everything they can, as opposed to targeting their figure to the main points of the paper.

---

## Author Response (AR2)

Dear Dr. Barbara Ervens,

Thank you very much for your patience and help! Thank you for permission for an extension of the resubmission of the revised manuscript (Ms. Ref. No.:acp-2016-1183). Based on the two reviewers' comments, we have made a substantial revision again to our manuscript, including reanalyzing the model results, enhancing our discussion and changing the figures. Additionally, this resubmitted version has been polished by English editor. We are confident that it is ready for publishing in a high quality journal. Detailed item-by-item responses to the comments are listed below.

Best regards,

Yours sincerely,

Jianhua Qi

Response to Referee report 1

General Comments: First of all, the space was not inserted appropriately in many parts, so it is hard to read and follow. Such a crude revision with low presentation quality should not be sent to reviewers. I have reviewed again this manuscript and found some improvements on the manuscript; however, many replies have not been found in the revised manuscript and/or replied well to my concerns. I feel that the presentation quality is still low as to be published from the high quality journal of Atmospheric Chemistry and Physics. With regret, I have judged to reject this manuscript again.

Response:We are sorry for the space missing problem. We have a double check for our submitted version, but not find this problem. We therefore re-install our software and avoid the problem in the new submission. In the new version, we further improve the quality of the manuscript according to two reviewers' comments. We are confident that it is ready for publishing in a high quality journal.

Q1. I have partly understood my concerning issue regarding the definition of dust event. Further concerning issue is the sampling duration of continuous dust event. Even the dust event continued multiple days, how should we consider the representativeness of the sampling? For instance, sample 20080528 and 20080529 (please note that the sampling time of 20080529 will have typo) had approximately one day interval. Was there large temporal variation of PM10 concentration during continuous dust days? If there was large change on PM10 concentration, why the authors collected on the listed time? The authors should state the reason, and should present the representativeness of 4 hrs sampling. In the revised manuscript, it will be kind for readers to explicitly state that 'http://www-cfors.nies.go.jp/~cfors/' is for forecast model over Asia, and 'http://www.qepb.gov.cn/m2/' is for observed concentration at Qingdao.

Response: Due to no dust events lasting over 12 hrs (Lee et al., 2015; Su et al., 2017; Zhang et al., 2007), we collected one dust sample with a 4-hr duration in a day. The sampling for dust particles started only when the measured PM10 mass concentration in Qingdao (http://www.qepb.gov.cn/m2/) and the forecasted dust mass over Asia (http://www-cfors.nies.go.jp/~cfors/) had greatly increased.

The samples with ID of 20080528 and 20080529 were subject to two different dust events occurring in two days instead of continuous samples for one dust event. On March 20-21, 2010, two dust events subsequently swept Qingdao. The 4 hr dust samples with IDs of 20100320 and 20100321 may not capture the entirety of the two events. However, the on-line data can allow adequate separation of the two dust event samples. The same was true for the dust samples with IDs of 20110501, 20110502.

The link illustration for these two links has been also added properly in the new version.

Lee, Y. G., Ho, C., Kim, J., and Kim, J.: Quiescence of Asian dust events in South Korea and Japan during 2012 spring: Dust outbreaks and transports, Atmos. Environ., 114, 92-101, 2015.

Su X., Wang Q., Li Z., Calvello M., Esposito F., Pavese G., Lin M., Cao J., Zhou C., Li D., Xu H. Regional transport of anthropogenic pollution and dust aerosols in spring to Tianjin — A coastal Su, X., Wang, Q., Li, Z., Calvello, M., Esposito, F., Pavese, G., Lin, M., Cao, J., Zhou, C., Li, D., and Xu, H.: Regional transport of anthropogenic pollution and dust aerosols in spring to Tianjin — A coastal megacity in China, Sci. Total. Environ., 584–585, 381–392, 2017.

Zhang, K., and Gao, H. W.: The characteristics of Asian-dust storms during 2000–2002: From the source to the sea, Atmos. Environ., 41, 9136-9145, 2007.

Q2. I have partly agreed, but I have further question on the application of a 3-D air quality model. First, what is the merit of the application of 3-D air quality model? In the revised manuscript, only the spatial distributions of PM10 were shown (Fig. 2 from CFORS model and Figs. S1-S3 with CFORS and WRF-CMA. Can such application reinforce the authors' discussion points? The behavior of IN were discussed in this manuscript, so what is the purpose to show PM10? The authors stated that 'The spatial distribution of PM10 concentrations for each dust event was consistent with the model results of dust by the Chemical Weather Forecast System (CFORS) by Uno et al. (2003)' (L199-201). If the consistency between other models is important, why the author calculated on your own model? I cannot follow this reason from the revised manuscript.

Response: Thank you for the suggestion. We have deleted the results by CFORS. The CMAQ model (v5.0.2) was applied to simulate the concentration of PM10, NOx, NH$_3$, NO$_3^-$ and NH$_4^+$ over the East Asia area for aerosol samples on dust and comparison days. We have revised the discussion on model results. Distribution of PM10 was used to characterize the dust events. Spatial distributions of PM10 during each dust events were consistent with the records in the "Sand-dust Weather Almanac" (CMA, 2009; 2010; 2012; 2013). The model results indicated that CMAQ results reasonably reproduce the mass concentrations of NO$_3^-$ (Fig. S6). Simulated NH$_4^+$ concentrations in dust samples were severely under-predicted with NMB values at -71%. For reference samples, simulated NH$_4^+$ concentrations sometimes can well reproduce the observational values, but sometimes totally off. The external mixing mechanism proposed in this study is urgently needed to be included in the model for accurately predicting the concentrations during dust events.

CMA: Sand-dust weather almanac 2008, China Meteorological Press, Beijing, 10-64, 2009.
CMA: Sand-dust weather almanac 2009, China Meteorological Press, Beijing, 11-59, 2010.
CMA: Sand-dust weather almanac 2010, China Meteorological Press, Beijing, 11-79, 2012.
CMA: Sand-dust weather almanac 2011, China Meteorological Press, Beijing, 10-53, 2013.

Q3. The following specific points also should be revised to clarify the model application.
L189: Centered point is needed because we cannot follow the modeling domain at the current description.

L193: On the INTEX-B emission inventory (Zhang et al., 2009), I suppose that NH3 emissions have not been provided. If so, this description should be changed.

L195, and Figures 6 and 8: So, all calculations were based on the emission level on 2008? Because the temporal resolution of INTEX-B emission inventory is month, I feel that there are no need to display all emissions on all dust samples. These emissions level should be differed only on month. Therefore, I suppose that the averaged (spring time) emissions of NOx and NH3 on each one figure is enough.

Response: We have supplemented the centered point (110 E, 34 N) in the new version.

According to the publications of INTEX-B and TRACE-P Asia emission inventories (Zhang Q et al., 2009; Streets et al., 2003.), INTEX-B inventory was developed based on TRACE-P inventory with $NH_3$ emission considered (the annual emission amount of $NH_3$ in China was 13.6 Tg). However, due to the low priority and low variability of $NH_3$ emission during 2000-2006, $NH_3$ emission was not updated in INTEX-B inventory, and the $NH_3$ emission in INTEX-B inventory was consistent with TRACE-P.

Agree and revised.

Zhang Q et al., Asian emissions in 2006 for the NASA INTEX-B mission; Atmos. Chem. Phys., 9, 5131-5153, 2009.

Streets D.G. et al., An inventory of gaseous and primary aerosol emissions in Asia in the year 2000, 108 (D21), DOI: 10.1029/2002JD003093, 2003.

Q4. Figure S1: What is the purpose to show the difference between (b) and (c)? In this caption, what is 'WRF-CMA'?

Response: We have indicated that one exterior dust sample was collected on 1 March when no dust was recorded in Qingdao by MICAPS. However, the MICPAS information over the whole country indeed showed the dust events in China on 1 March. And the modeled spatial distribution of PM10 and TSP mass concentration for this dust event on 1 March implied that the sample should be classified into dust sample. Therefore we listed all the supporting figures in Fig. S1. Fig. S1 (b) was the weather information from the MICAPS at 8:00 on Mar.2, 2008 and (c) was hourly PM10 concentration modeled by the WRF-CMAQ model at 15:00 on Mar.1, 2008. We guessed that the reviewer maybe refer to the difference between (c) and (d), therefore we deleted (d).

We have revised the caption.

Q5. Figure S3: In the main manuscript, it was stated that 'each dust sampling day are shown' in Fig. S3 (L218-219, L895). However, only the hourly concentration of PM10 concentration at 14:00 on 19 Mar 2011 were shown. Please confirm this supplemental figure.

Response: We really modeled the PM10 concentration on each dust sampling day, but only showed the PM10 concentration at the middle time of the sampling in Fig.S3 (Now Fig.S5 in the new version) due to too many figures. We have revised this section and the sentence has been revised into "The concentrations of PM10 and its major components $NO_3^-$ and $NH_4^+$ over East Asia on dust days and comparison days were modeled using the WRF-CMAQ model (Fig. S5-6)" in L341-342 in the new version.

Specific comments:
Q6. L35-36: This conclusion does not match to the manuscript contents. The authors stated that input of nitrogen to the ocean depends on the dust events.
Re-comment: cannot find this revision.

Response: The revision in the last round was prepared after the quick response. After a careful consideration, we agreed with the comment and delete the part in the last revised version.

Q7. I have confirmed and understood the meaning. However, is this revision corresponded to L226-230? If so, this revised sentence seems to contain many errors (NOT Table S2 but Table S1?). For example, we can find 410 μg/m3 on dust day sample on 20080315. What is the value of 80-1303%? These increased value were not corresponded to 'Ratio of DD to CS' shown in Table S1.

Response: Yes, this revision corresponded to L226-230 in last version. And the times of dust to non-dust day samples were replaced by the ratio according to the former suggestion. To avoid the confusion, we have revised Table S1 to give the increased ratio.

Q8.L171: Again, I cannot follow the calculation of "1.7-21.9 times (mean: 6.9) ".
L175: I cannot follow "10.3 times" for Fe. It can be calculated as 7.90 from the values in Table 2.
L176: "3.6-fold" will not be followed from Fig. 2. It should be listed in Table 2.
Re-comment: So, in this revised manuscript, these statements of the increment ratio on dust-day compared to non-dust day have not been explicitly appeared. In L243, the authors stated 'Table S1', but Table S1 contained not only the information of inorganic nitrogen but also TSP, Al, Fe, and nss-Ca. So, it is appropriate to mention on Table S1 in Section 3.1.

Response: Agree and revised.

Q9.L175: So, please state explicitly regarding this point to the readers. In the current form, nss-Ca was suddenly shown in Fig. 3 without any introduction.

Response: Agree and revised.

Q10. First of all, I cannot find the revision of 'The concentrations of ammonium were increased by 20' anywhere. Is this corresponded to Table S1? I suppose that the authors discussed regarding this point in L240-L244. Although ratio was shown in Table S1, percentages are discussed here. So it is hard to follow the manuscript. Why the discussion point have not been arranged on the uniformed unit?

Response: Agree and revised.

Q11.L194-L195: In this sentence, the authors stated "the effect of dust on inorganic nitrogen differed during different types of dust events". Why the authors suddenly focused on inorganic nitrogen here? In L192-193, it was mentioned "inorganic ion $SO_4^{2-}$ exhibited concentration variations that were similar to those of nitrate". L197: The figures for inorganic nitrate will be helpful information here, if the authors focused on inorganic nitrogen.
Re-comments: I cannot find this revision.

Response: The revision in the last round was prepared after the quick response. After a careful consideration, we completely rewrote the part to avoid confusion in the last revised version.
.
Q12. L207: (respectively less than 50 ug/g and 6 ug/g) will be the correct expression for ammonium.
Re-comments: I cannot find this revision.

Response: The revision in the last round was prepared after the quick response. After a careful consideration, we completely rewrote the part into "The ratios of mass concentrations of nitrate and ammonium to the total mass of sand particles were very low, i.e., less than 81 μg/g, which are approximately three orders of magnitude less than the corresponding values in our dust samples." at L230-232.

Q13.I am wondering that the differences of IN concentration between Duolun and Zhurihe. Both are Hunshandadke Desert, however, as is shown in Table 5, IN concentration was much higher in Duolun. Are there some emission source?

Response: Sand samples were collected at a remote site in Zhurihe desert. Little anthropogenic influence is expected. Atmospheric aerosol samples were collected at an urban site in Duolun on dust days for comparison. It is not surprised for a strong signal for anthropogenic sources. This has been clarified in the new version.

Q14. Again, only from the dust spatial distribution, it is hard to state the dust intensity.

Response: We had made a substantial revision on the part in the last round revision and didn't consider dust intensity as an important factor for our unique results.

Q15. L214-L216: Without more information of the intensity of dust, the discussion on 'dilution effect' seems to be lacked in scientific understanding. This part should be fully revised based on not only dilution effect but also dust intensity.
Re-comments: Again, only from the dust spatial distribution, it is hard to state the dust intensity.

Response: We had made a substantial revision on the part in the last round revision and didn't consider dust intensity as an important factor for our unique results.

Q16. I have confirmed the revision, but if the authors discussed on average (L300-303), the averaged values were also needed.

Response: We really had given the average of TSP in form of average±standard deviations at L300-303 in last revision. Now we had made a substantial revision on this part and didn't discuss TSP average concentration.

Q17. L227-L228: The favorable condition to form ammonium cannot be discussed without the information of NH3. In addition, Table 3 indicated the aerosol samples in the coastal region of the Yellow Sea. How about the status over air mass path? Is it sufficient to conclude only from the downwind information to the formation of inorganic nitrogen?
Re-comments: Again, I cannot understand the model application results.

Response: We modeled the emission and concentration of NOx and $NH_3$ over East Asia on the dust and comparison days. The model results showed that the calculated trajectories of the entire dust air mass passed over those highly polluted regions with strong emissions of $NO_x$ and $NH_3$ shown in Fig 6 and experienced different residence times therein. The average concentration of $NO_x$ and $NH_3$ during transport were calculated and discussed according to Categories 1 and 2. The air masses in Category 1 took over 11-39 hrs to cross over the highly polluted area with appreciable concentrations of NOx (5.7±1.4 ppb) and $NH_3$ (7.6±3.3 ppb). Except for the exterior samples, air masses in Category 2 took less than 10 hrs to cross over the polluted areas with lower concentrations of NOx (3.6±3.4 ppb) and $NH_3$ (4.7±4.7 ppb) and the mixing layer height along the route was 916-1194 m (on average) for each dust event. This further led to the external mixing of anthropogenic particulate matters and dust.

Q18. L230: "strong dust storm" cannot be discussed without any information on dust intensity here.
Re-comments: Again, from the additional information of CFORS, the spatial distribution pattern was found; however, how can we estimate the intensity?

Response: We had made a substantial revision on the part in the last round revision and didn't consider dust intensity as an important factor for our unique results.

Q19. L233-L234: But NOx concentration was high in Case 3. I cannot follow why the authors concluded   on Case 3.
Re-comments: So where did the authors discussed the NOx concentration in the manuscript?

Response: We had made a substantial revision on the part in the last round revision after the quick response. The $NO_x$ concentration was discussed in Section 4.3 in the new revision.

Q20. L254-L255: The authors simply mentioned "local emissions" here. Because the samples were collected on downwind regions in the coastal region of the Yellow Sea, I guess that the discussion on emission characteristics of each (or, at least, some categorized) air mass should be discussed in detail. The inorganic nitrogen concentrations are highly related to the local conditions both on emissions strength and meteorological parameters, so the discussion only on air mass speed and air mass path over ocean are insufficient.

Re-comments: Again, model is used only for spatial distribution and not inform the chemical production process.

Response: We had made a substantial revision according to the suggestion. The chemical production process was discussed in Section 4.1 "Theoretical analysis of the three categories". In Category 1, ammonium salt aerosols may externally exist with dust aerosols in these dust day samples and $NO_3^-$ and $SO_4^{2-}$ were almost completely associated with $NH_4^+$ in these dust day samples; whereas a larger fraction of $NO_3^-+SO_4^{2-}$ may exist as metal salts due to reactions of their precursors with dust aerosols in Category 2. The simulated $NO_3^-$ and $NH_4^+$ concentrations was compared with the observation in Qingdao, and the results indicated that the external mixing mechanism proposed in this study is urgently needed to be included in the model for accurately predicting the concentrations during dust events.

Q21. I have rechecked the discussion of backward trajectories discussed on Section 3.4. There are many points should be clarified.
Figure 5: Please add the explanation of the trajectory of 20110415 was excluded based on the discussion on Fig. 2. Why the authors displayed "non-dust samples"? What were the differences between non-dust and dust samples trajectories? I feel that these were similar.

Response: Agree and revised.

Q22. L314: What is the 'remaining one'? Please specify the trajectory data. In my opinion, two trajectories of 20110418 and 20110501 originated from northeast China.

Response: Yes, trajectory 20110501 was really from northeast China, however it then passed over the Inner Mongolia, and arrived at Qingdao from north, just like 20110502. Therefore, we grouped the trajectory into the air mass originated from Inner Mongolia, China. However, it was really easy to mislead the readers. Therefore, we accepted the suggestion, and revised the sentences into "The calculated air mass trajectories for 13 out of 14 samples showed that the air mass originated from North and Inner Mongolia, China (Fig. 5), generally consistent with the results of Zhang and Gao (2007). The remaining one, with ID of 20110418 originated from Northeast China.".

Q23. L317: What is the 'one exterior sample'? Please specify.

Response: Agree and revised.

Q24. L319: I cannot see 'the air masses crossed over the sea for 94-255km' from Figure 5, because Fig. 5 showed the whole view of trajectories across China. More detailed figure or explanation will be required.

Response: The distance over sea of the air mass for each sample was measured from the trajectory using TrajStat software (Wang et al., 2009).We have added the explanation in Section 2.3.

Wang, Y. Q., Zhang, X. Y., and Draxler, R. R.: TrajStat: GIS-based software that uses various trajectory statistical analysis methods to identify potential sources from long-term air pollution measurement data, Environ. Modell. Softw., 24, 938-939, 2009.

Q25. L328-329: What is the definition of the 'average mixing layer'. I suppose that the altitude of backward trajectories were so high because most of trajectories were originated outside China on 72 hrs. So, where is the averaged region to calculate '900m' in this sentence?

Response: The average mixing layer was calculated as an average of all points on the air mass back trajectory of each sample. This has been clarified in section 2.4 in the new version.

Q26. I cannot still understand the authors' conclusion here. As was discussed on L355-361, Fig. 8, and Table 7, although the coal combustion have increased on dust days, the contributions of local anthropogenic sources (especially secondary aerosols) have decreased on dust days. According to the discussion on Section 3.3, the authors concluded that ammonium salts were externally co-exist with dust aerosols in Category 1. So, why the contribution of secondary aerosols were decreased from PMF analysis. I feel that these results have contradicted. More careful discussion is required for this conclusion.

Response: The source of coal combustion on dust days became complicated. "mixture of coal combustion and other pollutants" means these compounds present contemporaneously, because that PMF model can't show the mixing or existing state. We have revised the sentence into "The source profile for coal combustion in dust day samples showed a high percentage of K+, Cl-, Ca, Mg, Co, Ni, As, Al and Fe, indicating coal combustion presenting contemporaneously with other pollutants emitted along the transport path on dust days.". Ammonium salts were externally co-exist with dust aerosols in Category 1, but showed lower concentrations in Category 2 likely due to unfavorable conditions for forming ammonium salts. Here the conclusion was a result of source appointment for all dust samples including Category 1 to 3. And we have revised the sentence "In these dust samples, including Categories 1-3, oil combustion, industry, soil dust, secondary aerosols, and coal combustion/other sources were identified as five major sources (Table 6).

Q27.L306: I cannot follow the calculation of "a factor of 1.1-5.8" and "a factor of 1.8-6.3".
Re-Comments: I cannot find this revision.

Response: We had made a revision on these sentences in the last round revision after the quick response. According to the former suggestion, we changed the factor to ratios. And this sentence was revised to "In Category 1, the dry deposition fluxes of $N_{NH_4^++NO_3^-}$ increased by 9-75% with increased TSP flux by 86-252% (Table S3)" at L371-372 in the new revision. And we also revised Table S3 to give increased proportion and the calculation method.

Q28. L309: What is the calculation method of "63%" and "46%"?
L310: What is the calculation method of "14%"?
Re-Comments: I cannot find this revision.

Response: We had made a revision on these sentences in the last round revision after the quick response. We have revised Table S3 to give increased proportion and the calculation method.

Q29. L317: For Fe, it seems that the increased ratio were 2.81-11.08 from Table S3.

Response: This sentence has been revised to "However, the dry atmospheric deposition fluxes of Fe increased by a factor of 124-2370% in dust day samples." at L383-384 in new revision.

Specific comments:
Q30.Table 6: Missing the note of a and b.

Response: Done.

Q31. Table S2: Please align the right-column, it is hard to follow. What is the meaning of *?

Response: Agree and revised.

Response to Referee report 2

This manuscript focuses on an important issue, specifically the relationship between ammonium and nitrate with mineral dust in China. The variability in previously reported relative concentration trends is worth exploring in detail. There are a number of areas where the paper could use improvement prior to publication, particularly with connecting to the literature and placing the work in context.

Response:We thank the reviewer's constructive comments and revise our manuscript accordingly. We are confident that it is ready for publishing in a high quality journal.

Comments:
Q1‐ Throughout the manuscript there are odd spacing issues, where two words are together without a space. As an example in the abstract "For these two groups, NH4+in dust day samples was present in the form of ammonium salts externally co‐existing with dust aerosols or the residual of incomplete reactions between ammonium salt and carbonate salts."

Response: We are sorry for the space missing problem. We have a double check for our submitted version, but not find this problem. We therefore re-install our software and avoid the problem in the new submission.

Q2‐ Line 51‐54: The authors state the Asian dust has been transported as far as the north pacific, but this understates what has been observed for Asian Dust. Uno et al 2009 Nat Geosci showed that Asian dust can circumnagivate the globe. showed Asian dust impacting California air quality, while Ault et al. 2011 JGR and Creamean 2013 Science showed impacts of Asian dust on orographic precipitation in the Sierra Nevada (in California). Pratt et al. 2009 Nat Geo showed Asian dust influencing clouds over Wyoming.

Response: Thank you very much for the suggestion. We revised the sentence to "Asian dust has been reported to not only    frequently cross over the mainland and the China Seas, but also to occasionally reach the remote northern Pacific Ocean or North America (Creamean 2013; Tan and Wang, 2014; Van Curen and Cahill, 2002; Zhang and Gao, 2007). In an extreme case, Asian dust was found to be transported more than one full circuit around the globe in approximately 13 days (Uno et al 2009)." at Line 45-49.

Creamean, J. M., Suski, K. J., Rosenfeld, D., Cazorla, A., DeMott, P. J., Sullivan, R. C., White, A. B., Ralph, F. M., Minnis, P., Comstock, J. M., Tomlinson, J. M., Prather, K. A.: Dust and Biological Aerosols from the Sahara and Asia Influence Precipitation in the Western U.S., Science, 339, 1572-1578, 2013.

Tan, S. C., and Wang, H.: The transport and deposition of dust and its impact on phytoplankton growth in the Yellow Sea, Atmos. Environ., 99, 491-499, 2014.

VanCuren, R., and Cahill, T.: Asian aerosols in North America: Frequency and concentration of fine dust, J. Geophys. Res., 107(D24), 4804, doi:10.1029/2002JD002204, 2002.

Zhang, K., and Gao, H. W.: The characteristics of Asian-dust storms during 2000–2002: From the source to the sea, Atmos. Environ., 41, 9136-9145, 2007.

Q3 Though the authors note that a native English speaker was utilized for the revision, a considerable improvement in the grammar and proofreading are needed before the writing is at a publishable level.

Response: This resubmitted version has been polished by English editor.

Q4  Line 233 insert comma after "samples"

Response: Done.

Q5  Line 266: Is it really a safe assumption that gas aerosol thermodynamic equilibrium is met for inorganic ions during a dust storm? It would seem that many non‐aqueous (i.e. solid) aerosol would be present that would not have normal equilibrium partitioning. It would be nice to see some evidence of this. This would also help support the conclusion that Ca(NO3)2 and CaSO4 are negligible.

Response: According to this reviewer's suggestion, the progresses on this issue in literature have been summarized and cited to support our analysis.

Q6 Line 282: The presence of Cu, brings to mind the question of transition metal ions and industrial sources of metal containing particles. How were these accounted for? Particularly since they often have different properties and propensity for generating ROS as Weber and company at Georgia Tech have shown.

Response: Cu was once used as an effective marker of diesel and biodiesel-blend exhaust (Gangwar et al., 2012), while it can also be derived from copper pyrites ($CuFeS_2$) in Inner Mongolia mines (Huang et al., 2010). The increase of Cu in the mass concentration in dust samples implied dust particles mixed with anthropogenic particles, particularly from industrial emissions, during transport.

Gangwar, J. N., Gupta, T., and Agarwal, A.K.: Composition and comparative toxicity of particulate matter emitted from a diesel and biodiesel fuelled CRDI engine, Atmos. Environ., 46, 472-481, 2012.

Huang, K., Zhuang, G., Li, J., Wang, Q., Sun, Y., Lin Y., and Fu J. S.: Mixing of Asian dust with pollution aerosol and the transformation of aerosol components during the dust storm over China in spring 2007, J. Geophys. Res-Atmos, 115, D00k13, Doi:10.1029/2009jd013145, 2010.

Q7.‐ It should be noted that there is a great deal of uncertainty regarding aerosol pH, particularly in North China, with estimates ranging from 3‐7 pH units. This of course will affect nitrate. The authors could comment on this with respect to their data, though keeping in mind Hennigan et al showing the proxy methods such as NH4+/(NO3‐+SO42‐) are qualitative at best.

Response: We thank the comments. We are not sure whether the estimated aerosols pH from 3‐7 pH units in north China were valid or not, by considering three types of aerosols, i.e., ammonium salt aerosol, $K_2SO_4$ or $KNO_3$ aerosol, and $CaCO_3$. TSP was collected in this study while PM2.5 was used for analysis by Hennigan et al (2015). It is not surprised parts of $NO_3^-$ and $SO_4^{2-}$ to be associated with metals in TSP samples, but we agree that $NO_3^-$ and $SO_4^{2-}$ may overwhelmingly associated with $NH_4^+$ in PM2.5 as found by Hennigan et al (2015). However, we tried our best to properly interpret the formation of nitrate and sulfate in different Categories in the revised version.

Hennigan, C. J., J. Izumi, A. P. Sullivan, R. J. Weber, and Nenes, A.: A critical evaluation of proxy methods used to estimate the acidity of atmospheric particles, Atmos. Chem. Phys., 15(5), 2775–2790, 2015.

Q8. What is the mineralogy of the Hunshandake Desert? Is it rich in CaCO3? Based on a few assumptions made, documentation of the presence of this mineral from aerosols in the region would be helpful. Perhaps some of Ro and co‑workers analysis of transported dust with SEM‑EDX?

Response: Thank you for the suggestion. The references results show that mineral dust is relatively enriched with Calcite (Matsuki, et al., 2005; Formenti et al., 2011; Nie et al., 2012) and the carbonate content generally de-creases from west to east with exception of the Gurbantunggut desert in China desert (Fig.RS1, Formenti et al., 2011). Except for 20080502, the remaining dust samples in Category 2 were transported from the desert relatively enriched with $CaCO_3$ (1-25% in Wt%) (Formenti et al., 2011). And Huang et al. (2010) found that calcite was one of the main species in the aerosol over Duolun and the high content of $CaCO_3$ (~80% in dust storm days) in the total soluble part of Duolun aerosol. A positive correlation between $NO_3^-$ and $SO_4^{2-}$ in Category 2 against a negative correlation in Category 1 also implied that the dust particles enriched with $CaCO_3$ in Category 2 might play an important role to form $SO_4^{2-}$ and $NO_3^-$. Ca-rich dust particles coated with highly soluble nitrate were observed at Kanazawa in Japan during Asian dust storm periods using SEM/EDX (scanning electron microscopy equipped with an energy dispersive X-ray spectrometer) (Tobo et al.,2010). The single-particle observation conducted by Hwang and Ro (2006) showed that $CaCO_3$ in dust particles was almost completely consumed to produce mainly $Ca(NO_3)_2$ species.

(a)

[Figure]

[Figure]

[Figure]

**Fig. 2.** Potential source areas in Eastern Asia based on work by Xuan et al. (2004), Laurent et al. (2006), Shao and Dong (2006), Kim et al. (2007), Wang et al. (2008), and Zhang et al. (2003e). Outlines of potential source areas (shaded areas) are drawn by hand. PSA EAS-1: Taklamakan; PSA EAS-2: Gurbantunggut; PSA EAS-3: Kumtaq, Qaidam, Hexi corridor; PSA EAS-4: Mongolian (Northern Gobi) deserts; PSA EAS-5: Inner Mongolian (Southern Gobi) deserts: Badain Jaran and Tengger (PSA EAS-5a), Ulan Buh, Hobq, Mu Us (PSA EAS-5b); PSA EAS-6: north-eastern deserts (Otindag Sandy Land, Horquin Sandy Land, Hulun Buir Sandy Land).

Figure RS1. Potential source areas in Eastern Asia adapted from Formenti et al., 2011(a) and The 72-h backward trajectories for samples in Category 2 (b).

Formenti, P., Sch¨utz, L., Balkanski, Y., Desboeufs, K., Ebert, M., Kandler, K., Petzold, A., Scheuvens, D., Weinbruch, S., and Zhang, D.: Recent progress in understanding physical and chemical properties of African and Asian mineral dust, Atmos. Chem. Phys., 11, 8231–8256, doi:10.5194/acp-11-8231-2011, 2011.

Huang K., Zhuang G., Li J., Wang Q., Sun Y., Lin Y., and Fu J. S.Mixing of Asian dust with pollution aerosol and the transformation of aerosol components during the dust storm over China in spring 2007. Journal Of Geophysical Research, Vol. 115, D00k13, Doi:10.1029/2009jd013145, 2010

Hwang, H. and Ro, C. U.: Direct observation of nitrate and sulfate formations from mineral dust and sea-salts using low-Z particle electron probe X-ray microanalysis, Atmos. Environ., 40, 3869-3880, 2006.

Matsuki, A., Iwasaka, Y., Shi, G. Y., Chen, H. B., Osada, K., Zhang, D., Kido, M., Inomata,Y., Kim, Y. S., Trochkine, D., Nishita, C., Yamada, M., Nagatani, T., Nagatani, M., and Nakata, H.: Heterogeneous sulfate formation on dust surface and its dependence on mineralogy: balloon-borne observations from ballon-borne measurements in the surface of Beijing, China, Water Air Soil Poll., 5, 101–132, 2005.

Nie,W., Wang, T., Xue, L. K., Ding, A. J., Wang, X. F., Gao, X. M., Xu, Z., Yu, Y. C., Yuan, C., Zhou, Z. S., Gao, R., Liu, X. H., Wang, Y., Fan, S. J., Poon, S., Zhang, Q. Z., and Wang, W. X.: Asian dust storm observed at a rural mountain site in southern China: chemical evolution and heterogeneous photochemistry, Atmos. Chem. Phys., 12, 11985-11995.

Tobo, Y., Zhang, D. Z., Matsuki, A., and Iwasaka, Y.: Asian dust particles converted into aqueous droplets under remote marine atmospheric conditions, PNAS Proceedings of the National Academy of Sciences of the United States of America, 107, 17905–17910, 2010.

Q8‐ Line 321 some evidence for "humid marine conditions might have enhanced particle‐particle coagulation" would be helpful. The number concentrations in the marine boundary layer are unlikely to be > 105 #/cm3 where coagulation is prevalent, more likely in the 102—103 #/cm3. Are the authors referring to fog‐processing? That would seem to be the primary way this could happen in a marine environment.

Response: We are sorry for the confusion. The sentence has been revised to "On the other hand, the humid marine conditions (the average RH ranged in 50-75% over the Bohai and Yellow Seas in 2006-2012) might have enhanced hetero-coagulation between dust and smaller anthropogenic particles, leading to the release of $NH_3$ via reactions between preexisting ammonium salts and carbonate salts.".

Tobo Y., Zhang D., Matsuki A., Iwasaka Y. Asian dust particles converted into aqueous droplets under remote marine atmospheric conditions, PNAS, 2010, 107: 17905–17910.

Q9‑ Line 326 The line "ammonium salts mostly co‑existed with dust aerosols externally" is confusing as written. Is the population externally mixed with respect to ammonium nitrate and dust? Or are the salts co‑existing with dust, but not other particle types? Please rephrase for clarity.

Response: Thank you for the suggestion. We have revised to "ammonium salt aerosols may externally exist with dust aerosols".

Q10‑ Overall many of the conclusions on page 12 appear to mostly be speculation with little data to support it. I would recommend sticking to conclusions with more support from the data in the paper.

Response: We have made major revision on this section in the new revision.

Q11‑ Line 357: The source profile for coal, could it have dust mixed in? When the author's say that there is a "mixture of coal combustion and other pollutants" are they saying that they are internally mixed or simply present contemporaneously? Clarifying that point would be helpful.

Response: "mixture of coal combustion and other pollutants" means these compounds present contemporaneously, because that PMF model can't show the mixing or existing state. We have revised the sentence into "The source profile for coal combustion in the dust day samples showed a high percentage of $K^+$, $Cl^-$, Ca, Mg, Co, Ni, As, Al and Fe, indicating that coal combustion particles may exist contemporaneously with other anthropogenic pollutants emitted along the transport path.".

Q12‑ Overall the Figures could use improvement as portions are hard to read and the take home point of each is not always clear. It seems at times as if the authors are simply showing everything they can, as opposed to targeting their figure to the main points of the paper.

Response: Agree and revised.

[revised manuscript text omitted]

Category 2 (1.35±2.45 μg/m³) were lower or comparable to those in Category 1 (1.51±2.16 μg/m³).

The potential formation of nitrate metal salts was expected to be similar between the two categories, while unfavorable formation conditions for ammonium nitrate greatly increased decreased the mass concentrations of nitrate and the contributions to the TSPs in Category 12. Note that the $NO_3$

concentrations in Category 2 (1.35±2.45 μg/m³) were lower or comparable to those in Category 1

(1.51±2.16 μg/m³).

Overall, the higher ammonium concentrations observed in the dust day samples in Category 1 were likely associated with external co-existence of ammonium salt aerosols and dust particles in some dust days. However, the lower concentrations in Category 2 were likely due to unfavorable conditions for forming ammonium salts in some dust days. The observed ammonium was just the residual of incomplete reactions between preexisting ammonium salt and carbonate salts depending on dust event, atmospheric chemical state, etc. More discussion on this issue will be presented in Section 3.4.

**3.4 Influence of transport pathways on particulate inorganic nitrogenin dust samples**

The calculated air mass trajectoriesof 13 out of 14 samples showed that the air mass originated from north and Inner Mongolia, China (Fig. 54), generally consistent with the results by Zhang and Gao (2007). The remaining one 20110418, originated from Northeast China. Figs. 6 and 75 (taking emission in 2008 as an example) show a few areas with high emissions of $NO_x$ and $NH_3$, e.g., Liaoning,

Beijing Tianjin Hebei, Shandong, Henan and Jiangsu in China. The calculated trajectories showed that all the air mass passed over parts of these highly polluted regions and experienced different residence time in these regions. In Fig. 54, except for the one exterior sample 20110502, all trajectories in

Category 1 showed that the air masses were transported from either the north or northwest over the continent. In Category 2, the air masses crossed over the sea for 94 255 km prior to arriving at the reception site. NH$_3$-poor conditions in the marine atmosphere disfavored the formation and existence of ammonium nitrate. On the other hand, the humid marine conditions (the average RH beingranged in 50 75% over the Bohai and Yellow Sea in Qingdaoin 2006 2012) might have enhanced hetero coagulation between dust and smaller anthropogenic particlesparticle particle coagulation and might have led to the release of NH$_3$ via reactions between preexisting ammonium salts and carbonate salts. Tobo et al. (2010) suggested that the conversion of insoluble CaCO$_3$ to Ca(NO$_3$)$_2$ tends to be dominated over urban and industrialized areas of the Asian continent, while relatively moist conditions in the marine boundary layer (usually, RH >60%), it is highly likely that the production of CaCl$_2$ exceeds that of Ca(NO$_3$)$_2$ by modifying Ca rich particles in dust storms. Therefore we think the input of marine air during the transport was one reason for the low concentration of NH$_4^+$+NO$_3^-$ in Category 2.

Moreover, we also examined the links among the measured concentrations of particulate ammonium and nitrate, the mixing layer along the back trajectories, and the residence time of air masses crossing over the highly polluted zones. The results supported our hypothesis, i.e., ammonium salts mostly co-existed with dust aerosols externally. For example, except for 20080425, most of the time all dust day samples mostly traveled at an altitude above the mixing layer before mixing down to ground level. The transport of dust air masses above the mixing layer reduced the possibility for internal mixing of ammonium salts and reaction with dust aerosols along the long transport path. For most sampling days in Category 1, the average mixing layer was less than 900 m, favoring the trapping of locally emitted anthropogenic air pollutants in the mixing layer. In addition, the air masses in Category 1 at this elevation apparently moved slowly and took over 10 11 39 hr to cross over the highly polluted area. Even lower speeds were expected for air masses at the bottom of the mixing layer, as wind speed decreases with height. Except for exterior samples, the sampling days in Category 2 featured a mixing layer that was higher than 900 m on average (916 1194m), higher than 900 m. The air masses in this Category at this elevation took less than 10 hr to cross over the highly polluted areas and generally had higher speeds. Theoretically, a lower mixing layer and a lower wind speed favored the accumulation of air pollutants and the formation of ammonium nitrate to some extent. The transport of dust air masses above the mixing layer reduced the possibility for internal mixing of ammonium salts and reaction with dust aerosols along the long transport path. The shorter time for dust air masses mixing down to ground level before arriving at the reception site and lower wind speed (mean of 2.8m/s at sampling site) also increased the possibility for external co-existence between ammonium salt aerosols and dust aerosols in

Category 1. The reverse could be argued to explain the observations for Category 2 (average wind speed being 6.2 m/s at sampling site). The single particle characterization also showed that the Asian dust particles collected in Korea were mixed with sea salts entrained over the Yellow Sea, as well as air pollutants from the eastern China coastal areas for a slow-moving, low-altitude air mass (Hwang et al.,

[revised manuscript text omitted]

All these dust events, observed in Qingdao, had a characteristic of large influence range and relatively strong intensity. Though the dust intensity was judged by visibility in China (CMA, 2004), the PM10 concentration during a dust event can also show the dust intensity indirectly. Generally, strong dust storm had a high $PM_{10}$ value, such as 2010320, consistent with the dust records (CMA, 2009). The dust intensity and influence range of these dust events varied greatly, depending on the dust event. The different dust events were expected to had different impact on the composition of aerosols in downwind area.

Moreover, we also examined the links among the measured concentrations of particulate ammonium and nitrate, the mixing layer along the back trajectories, concentration of $NO_x$ and $NH_3$ along the back trajectories, and the residence time of air masses crossing over the highly polluted zones. The results supported our hypothesis, i.e., ammonium salts mostly existed with dust aerosols externally. For example, except for 20080425, most of the time all dust day samples traveled at an altitude above the mixing layer before mixing down to ground level. The transport of dust air masses above the mixing layer reduced the possibility for internal mixing of ammonium salts and reaction with dust aerosols along the long transport path. For most sampling days in Category 1, the average mixing layer was less than 900 m with high concentration of $NO_x$ and $NH_3$ in range of 13,000-48,000 (Table 5), favoring the trapping of locally emitted anthropogenic air pollutants in the mixing layer. In addition, the air masses in Category 1 took over 11-39 hr to cross over the highly polluted area. Except for exterior samples, the sampling days in Category 2 featured a mixing layer that was higher than 900 m on average (916-1194m) and much low concentration of $NO_x$ and $NH_3$ in range of 26-13,100 (Table 5). The air masses in this Category took less than 10 hr to cross over the highly polluted areas. Theoretically, a lower mixing layer, high concentration of $NO_x$ and $NH_3$, and a lower wind speed favored the accumulation of air pollutants and the formation of ammonium nitrate to some extent. The shortlonger residence time over the highly polluted zones for dust air masses mixing down to ground level before arriving at the reception site and lower wind speed (mean of 2.8m/s at sampling site) also increased the possibility for external existence between ammonium salt aerosols and dust aerosols in Category 1. The reverse could be argued to explain the observations for Category 2 (average wind speed being 6.2 m/s at sampling site). The single particle characterization also showed that the Asian dust particles collected in Korea were mixed with sea salts entrained over the Yellow Sea, as well as air pollutants from the eastern China coastal areas for a slow moving, low altitude air mass (Hwang et al., 2008). 
[revised manuscript text omitted]

---

## Editor Decision (ED2)

**Comments by Reviewer #1, including some remarks by the editor.**

1) Distinguishing the three categories

    a) The reviewer still questions the representativeness of the three categories. As the main characteristics is the ammonia and nitrate content, evaporation of these compounds might bias the conclusion on which category the samples belong to.

    Can you comment on this possible bias?

    b) The reviewer criticizes that the expression 'adequate separation' in l. 92 is too vague. Please clarify.

2) Assumptions on emissions

The reviewer asks why emissions for the year 2008 were used as they may not be representative for the following years. As you explain that data from the year 2006 were extrapolated to the year 2008, I (the editor) wonder if the same methodology could be applied in order to obtain emission scenarios for the subsequent years, too.

3) Importance of dust intensity

The reviewer still questions the statement of the low importance of the dust intensity. Can you comment on the statement that $Ca2+$ should be dependent on the dust event intensity?

4) Category 3

I agree with the reviewer that Category 3 should be mentioned in the abstract. In addition, I am missing a discussion of Category 3 in Section 4.1.

5) Inorganic nitrogen

a) The reviewer asks why the focus of the study was inorganic nitrogen. Can you estimate any possible contribution of organic nitrogen in the particles?

b) I think 'inorganic nitrogen' is a too broad term. Only at one place in the manuscript it is mentioned that nitrite is excluded. I suggest being explicit and replacing 'inorganic nitrogen' by '$NH4+$ and $NO3-$' throughout the manuscript.

6) Median vs Mean

Please correct the contradiction of l. 205 and Table S1 (cf reviewer comment)

**Additional editor comments**

l. 22: Do you mean 'externally mixed', i.e. in separate particles?

l. 26: What does $< 3$ refer to here?

l. 57/8: This sentence is not clear. Please reword.

l. 113: remove 'the'

l. 173: Are the emissions modeled, i.e. predicted based on assumptions of sources or are they an input to the model?

l. 204/5: This is ambiguous. As it is written, the text suggests that each individual sample pair exhibited a net increase of 82-1303%. Is this true? Or was this large range the range that was determined based on all samples?

l. 239: This sentence needs to be improved. Do you mean 'the absolute increase…'? I don't understand what is meant by 'complex for the interactions'. It is very vague and grammatically wrong.

l. 259 and throughout the manuscript: What is meant by 'exterior sample'? Do you mean an outlier? How was this determined?

l. 262: 'It was commonly believed' should be changed here. What evidence was this assumption based on? Could it be concluded based on more than one study?

l. 267: Do you mean 'may be externally mixed'?

l. 270/1: I don't understand this. How does the dilution effect affect particle composition and what chemical reaction(s) is/are referred to here?

l. 275 and throughout the manuscript: Is there any evidence in previous studies that metal ions form stable salts in particles? References? Are these all salts or would also metal-sulfato-complexes be possible?

Section 4.3: I got a bit lost in this Section? Which part is based on measurements and which based on model results? Please clarify.

l. 351: 'totally off' is very colloquial. Is there any explanation for this discrepancy?

l. 373: I cannot follow here. Why is ammonium excluded in Category 3? Isn't that a contradiction as you mention in the following sentence that you discuss here $N(NH_4^+ + NO_3^-)$?

l. 374: 'A larger decrease' than what? Please clarify.

l. 378 ff: Again, it is not clear whether the following text is based on observations or measurements. Please clarify.

Final comment: What are the main conclusions of your study? They should be summarized in a separate conclusion section after Section 4.4.

---

## Author Response (AR3)

Dear Dr. Barbara Ervens,

Thank you very much for your patience and help! Thank you for the opportunity to resubmit our manuscript (acp-2016-1183) to your journal. Based on the reviewers' and editor's comments, we have carefully revised our manuscript, including supplementing the reference, enhancing our discussion and supplementing the conclusion. We are confident that it is ready for publishing in a high quality journal. Detailed item-by-item responses to the comments are listed below.

Best regards,

Yours sincerely,

Jianhua Qi

Response to comments

**Comments by Reviewer #1, including some remarks by the editor.**

1) Distinguishing the three categories
a) The reviewer still questions the representativeness of the three categories. As the main characteristics is the ammonia and nitrate content, evaporation of these compounds might bias the conclusion on which category the samples belong to.
Can you comment on this possible bias?

**Response:** As reviewer suggested, gas-particle interactions, particle-particle interactions, and dissociation of semi volatile species can lead to the sampling loss of ammonia and nitrate (Dougle and Ten Brink, 1996; Pathak et al., 2004; Wang and John, 1988), especially at high temperature (e.g., higher than 30℃) and low relative humidity (e.g. less than 40%)(Pathak et al., 2004). Pathak et al. (2004) studied the sampling loss of ammonium and nitrate in PM2.5 using a speciation sampler equipped with two denuders and a filter-pack system, and found that ammonium loss was low with a ratio less than 11% but the nitrate loss from the Teflon filter were significant. Chang et al. (2000) found that sampling loss is rather small for undenuded filter sampling and the higher loss observed for the denuded filter because the denuder removes all of the nitric acid, a condition that enhances volatilization. In addition, Wang and John (1988) found ammonium nitrate evaporative loss was less than 10% when the loaded mass is abundant (more than 2500 μg) in Teflon filters. Our samples were collected on quartz microfiber filters only for 4 hrs using high-volume air sampler without denuder at a flow rate of 1 $m^3$/min, corresponding to a high mass on filter (more than 2500 μg for all samples). The evaporation of ammonia and nitrate were very likely negligible under such samplings.

Pathak, R.. K., X. H., Yao, and C. K., Chan.: Sampling artifacts of acidity and ionic species in PM2.5, Environ. Sci. Technol., 38, 254-259, 2004.

Dougle, P. G., H. M., Ten Brink.: Evaporative losses of ammonium nitrate in nephelometry and impactor measurements, J. Aerosol Sci., 27(S1), S511-512, 1996.

Chang, M. C., C., Sioutas, S., Kim, H., Gong Jr., and Linn W. S.: Reduction of nitrate losses from filter and impactor samplers by means of concentration enrichment, Atmos. Environ., 34, 85-98, 2000.

Wang, H.C., and John, W.: Characteristics of the Berner Impactor for sampling inorganic ions. Aerosol. Sci. Tech., 8, 157-172, 1988.

b) The reviewer criticizes that the expression 'adequate separation' in l. 92 is too vague. Please clarify.

**Response:** The sentence has been revised into "The on-line data in high time-resolution can allow identifying two dust events accurately from the start to the end." in l. 96-97 in the revised manuscript.

2) Assumptions on emissions
The reviewer asks why emissions for the year 2008 were used as they may not be representative for the following years. As you explain that data from the year 2006 were extrapolated to the year 2008, I (the editor) wonder if the same methodology could be applied in order to obtain emission scenarios for the subsequent years, too.

**Response:** Zhang et al. (2009) generated the emissions of air pollutants in 2006 including NOX and NH3 over East Asia, and they updated the emission inventory in 2008 for us being used in this study, using technology-based approach with detailed activity and technology information. Thus it is only available for emission of 2008 but not every year during 2008-2011. The annual variation of air pollutant emission was likely small at those periods, e.g., NOx shown in Fig. R1, especially the spatial distribution over China.

[Figure]

**Figure** R1. The EDGAR global anthropogenic emission inventory of NOx during 2008-2010 (From http://edgar.jrc.ec.europa.eu/overview.php?v=431) .

3) Importance of dust intensity

The reviewer still questions the statement of the low importance of the dust intensity. Can you comment on the statement that $Ca^{2+}$ should be dependent on the dust event intensity?

**Response:** According to Sand-dust weather almanac (CMA, 2009; 2010; 2012; 2013), half of the events were recorded as sand and dust storm (visibility less than 1000 m) and the rest was the severe sand and dust storm (visibility less than 500 m) in the source regions in Category 1 and 2, with exception of two events recorded as blowing dust (visibility reduce to 1000–10000 m) in Category 3. When the dust arrived at Qingdao after the long-range transport, we only observed floating dust (horizontal visibility less than 10000 m) in Qingdao. Therefore we didn't consider dust intensity as an important factor for $N_{NH4++NO3-}$ in dust events we studied in the last revised manuscript.

Just as Formenti et al. (2011) reviewed, studies on the mineralogical composition of unpolluted aerosol in dust source region in China are limited. The published references indicated the carbonate content and Dolomite ($CaMg(CO_3)_2$) generally can be used as a source tracer for Asia dust (Formenti et al. , 2011; Jong et al., 2008; Li et al., 2007; Shen et al., 2005). The carbonate content and Ca/Al ratio exhibits a geographical dependence with decreasing value from west to east (with exception of the Gurbantunggut desert), following the carbonate distribution in soil (Formenti et al., 2011). Thus, Ca concentration should highly depend on the dust source more than dust intensity.

CMA: Sand-dust weather almanac 2008, China Meteorological Press, Beijing, 10-64, 2009.

CMA: Sand-dust weather almanac 2009, China Meteorological Press, Beijing, 11-59, 2010.

CMA: Sand-dust weather almanac 2010, China Meteorological Press, Beijing, 11-79, 2012.

CMA: Sand-dust weather almanac 2011, China Meteorological Press, Beijing, 10-53, 2013.

Formenti, P., Sch¨utz, L., Balkanski, Y., Desboeufs, K., Ebert, M., Kandler, K., Petzold, A., Scheuvens, D., Weinbruch, S., and Zhang, D.: Recent progress in understanding physical and chemical properties of African and Asian mineral dust, Atmos. Chem. Phys., 11, 8231–8256, doi:10.5194/acp-11-8231-2011, 2011.

Jeong, G. Y.: Bulk and single-particle mineralogy of Asian dust and a comparison with its source soils, J. Geophys. Res., 113, D02208, doi:10.1029/2007jd008606, 2008.

Li, G., Chen, J., Chen, Y., Yang, J., Ji, J., and Liu, L.: Dolomite as a tracer for the source regions of Asian dust, J. Geophys. Res., 112, D17201, doi:10.1029/2007jd008676, 2007.

Shen, Z. X., Li, X., Cao, J., Caquineau, S., Wang, Y., and Zhang, X.: Characteristics of clay minerals in Asian dust and their environmental significance, China Particuology, 3, 260–264, 2005.

4) Category 3

I agree with the reviewer that Category 3 should be mentioned in the abstract. In addition, I am missing a discussion of Category 3 in Section 4.1.

**Response:** We are very sorry for the missing. We have supplemented the flux results of Category 3 in abstract in l.29-31. And we have supplemented the discussion of Category 3 in l. 310-319 in Section 4.1. However, the unique changes in $NH_4^+$ and $NO_3^-$, different from Category 1 and 2, need further investigation.

5) Inorganic nitrogen a) The reviewer asks why the focus of the study was inorganic nitrogen. Can you estimate any possible contribution of organic nitrogen in the particles?

Response: Inorganic nitrogen reportedly contributed to ~80% of the total water-soluble nitrogen (TDN) in atmospheric particles collected over the Yellow Sea and in Qingdao (Shi et al., 2010). In the region, the dry deposition flux of the inorganic nitrogen accounted for more than 75% for the TDN (Qi et al., 2013). When deposited to the ocean via atmospheric dry deposition, inorganic nitrogen has great impact on marine productivity due to its bioavailability. To update and improve our knowledge on reactive nitrogen carried by dust particles, we focused on nitrate and ammonium by excluding nitrite because of its very low concentration.

We have supplemented the contribution of organic nitrogen in the particles in l. 65-69 in *Introduction* Section.

Shi, J. H., Gao, H. W., Zhang, J., Tan, S. C., Ren, J. L., Liu, C. G., Liu, Y., and Yao, X. H.: Examination of causative link between a spring bloom and dry/wet deposition of Asian dust in the Yellow Sea, China, J. Geophys. Res-Atmos., 117, 127-135, 2012.

Qi, J. H., Shi, J. H., Gao, H. W., and Sun, Z.: Atmospheric dry and wet deposition of nitrogen species and its implication for primary productivity in coastal region of the Yellow Sea, China, Atmos. Environ., 81, 600-608, 2013.

b) I think 'inorganic nitrogen' is a too broad term. Only at one place in the manuscript it is mentioned that nitrite is excluded. I suggest being explicit and replacing 'inorganic nitrogen' by '$NH_4^+$ and $NO_3^-$' throughout the manuscript.

**Response:** We have replaced "inorganic nitrogen" by "$NH_4^+$ and $NO_3^-$" throughout the manuscript.

6) Median vs Mean
Please correct the contradiction of l. 205 and Table S1 (cf reviewer comment)

**Response:** We are sorry for the confusion. We have supplemented the average concentration in Table S1. And we have revised the "median" into "mean" (Now l. 212) in the revised manuscript.

Additional editor comments
l. 22: Do you mean 'externally mixed', i.e. in separate particles?

**Response:** "externally mixed" has been revised into "existing separately" throughout the manuscript.

l. 26: What does < 3 refer to here?

**Response:** It referred to "Our modeled results satisfied the reasonable fit criteria, i.e. 90% of the scaled residuals were located between the range −3 and +3 for each species.", which has been moved to Section 2.3 in l. 158-160.

l. 57/8: This sentence is not clear. Please reword.

**Response:** We have reworded the sentence to "However, Zhang et al. (2010a) reported an interesting result, i.e., the concentrations of $NO_3^-$ and $NH_4^+$ were lower during strong dust storm

events than weak dust events. A high uncertainty appeared to exist for carrying amount of reactive nitrogen by dust particles." in l. 56-58.

l. 113: remove 'the'

**Response:** Revised.

l. 173: Are the emissions modeled, i.e. predicted based on assumptions of sources or are they an input to the model?

**Response:** We are sorry for the confusion. Zhang et al. (2009) generated the emissions of air pollutants in 2006 including $NO_X$ and $NH_3$ over East Asia and they updated the emission inventory in 2008 for us being used in this study. And the sentence was supplemented in l. 180-182.

l. 204/5: This is ambiguous. As it is written, the text suggests that each individual sample pair exhibited a net increase of 82-1303%. Is this true? Or was this large range the range that was determined based on all samples?

**Response:** Yes, it's true. The sentence has been revised into "In each individual pair of dust day sample against reference sample, a net increase in the mass concentration of TSPs was observed. The percentages varied from 82 to 1,303% on basis of events, with a mean value of 403% (Table S1)" in l.210-212.

l. 239: This sentence needs to be improved. Do you mean 'the absolute increase…'? I don't understand what is meant by 'complex for the interactions'. It is very vague and grammatically wrong.

**Response:** The sentence has been revised into "Since air pollutant emissions, meteorological conditions, chemical reactions, and others can affect the concentrations of $NH_4^+$ and $NO_3^-$ in atmospheric particles collected in dust days, the observed increase or decrease in the mass concentration of nitrate and ammonium in different dust samples against the reference implied the combined effect of those factors." in l. 245-248.

l. 259 and throughout the manuscript: What is meant by 'exterior sample'? Do you mean an outlier? How was this determined?

**Response:** We have revised "exterior sample" into "except for…" throughout the manuscript.

l. 262: 'It was commonly believed' should be changed here. What evidence was this assumption based on? Could it be concluded based on more than one study?

**Response:** We have rewritten the sentence into "Anthropogenic ammonium nitrate and ammonium sulfate were thought to be produced by gas, aqueous phase reaction and thermodynamic equilibrium processes and they usually internally mixed (Seinfeld and Pandis, 1998)." in l. 269-271.

Seinfeld, J. H., and Pandis, S. N.: Atmospheric Chemistry and Physics: From Air Pollution to Climate Change, 2nd Edition, Wiley, New York, 1191 pp., 1998.

l. 267: Do you mean 'may be externally mixed'?

**Response:** "may externally exist with dust aerosols" has been revised into "very likely existed separately" in l. 273.

l. 270/1: I don't understand this. How does the dilution effect affect particle composition and what chemical reaction(s) is/are referred to here?

Response: We are very sorry for the confusion. According to the reference, Huang et al. reported that the higher concentration of $SO_4^{2-}$ was observed in dust samples due to the heterogeneous reaction on the alkaline dust during dust storms, while the concentrations of $NO_3^-$ and $NH_4^+$ decreased due to the dilution of the local pollution by a strong wind associated with the invaded dust. Because the low temperature, relative humidity, strong wind and the low pollution gases did not favor the chemical conversion, and the NOR (nitrogen oxidation ratio) was low and even less than 1% during the dust storm of Beijing (Yuan et al., 2008). We have revised the sentence to into "The observed $NO_3^-$ and $NH_4^+$ in Asia dust samples were argued due to physically mixing two types of particles rather than the heterogeneous formation of nitrate and ammonium (Huang et al., 2010)." in l.276-278 to clarify this question.

Huang, K., Zhuang, G., Li, J., Wang, Q., Sun, Y., Lin Y., and Fu J. S.: Mixing of Asian dust with pollution aerosol and the transformation of aerosol components during the dust storm over China in spring 2007, J. Geophys. Res-Atmos, 115, D00k13, Doi:10.1029/2009jd013145, 2010.
Yuan, H., Zhuang, G., Li, J., Wang, Z. and Li, J.: Mixing of mineral with pollution aerosols in dust season in Beijing: Revealed by source apportionment study, Atmos. Environ., 42, 2141–2157, 2008.

l. 275 and throughout the manuscript: Is there any evidence in previous studies that metal ions form stable salts in particles? References? Are these all salts or would also metal-sulfato-complexes be possible?

**Response:** We have supplemented the reference to support our hypothesis, i.e. metal ions can form stable salts in particles. Cu existed in form of salts and organic complexes (Scheinhardt et al., 2013; Wang et al., 2016; Zhang et al., 2015). Sulfate can exist in many forms of metal salts in atmospheric particles, such as $Na_2SO_4$, $K_2SO_4$, $K_2Ca(SO_4)_2 \cdot H_2O$, $Na_2Ca(SO_4)_2$, $Na_2Mg(SO_4)_2 \cdot 4H_2O$, $(NH_4)_2Mg(SO_4)_2 \cdot 6H_2O$, $Na_3(NO_3)(SO_4) \cdot H_2O$ (Chabas and Lefèvre, 2000; Sobanska et al., 2012; Xie et al., 2005). in l. 283-293.

Chabas, A., and Lefèvre, R. A.: Chemistry and microscopy of atmospheric particulates at Delos (Cyclades–Greece), Atmos. Environ., 34, 225–238, 2000.
Scheinhardt, S., Müller, K., Spindler, G., and Herrmann, H.: Complexation of trace metals in size-segregated aerosol particles at nine sites in Germany, Atmos. Environ.,74,102-109, 2013.
Sobanska, S., Hwang, H., Choël, M., Jung, H., Eom, H., Kim, H., Barbillat, J., and Ro C.: Investigation of the Chemical Mixing State of Individual Asian Dust Particles by the Combined Use of Electron Probe X-ray Microanalysis and Raman Microspectrometry, Anal. Chem., 84 (7), 3145–3154, 2012.
Wang, H., An, J., Shen, L., Zhu, B., Xia, L., Duan, Q., and Zou, J.: Mixing state of ambient aerosols in Nanjing city by single particle mass spectrometry, Atmos. Environ., 132, 123-132, 2016.
Xie, R. K., Seip, H. M., Leinum, J. R., Winje, T., and Xiao, J. S.: Chemical characterization of, individual particles

(PM10) from ambient air in Guiyang City, Sci. Total. Environ., 343(1-3).261-271, 2005.

Zhang, G., Han, B., Bi, X., Dai, S., Huang, W., Chen, D., Wang, X., Sheng, G., Fu, J., and Zhou, Z.: Characteristics of individual particles in the atmosphere of Guangzhou by single particle mass spectrometry, Atmos. Res., 153, 286-295, 2015.

Section 4.3: I got a bit lost in this Section? Which part is based on measurements and which based on model results? Please clarify.

**Response:** The emission of $NO_x$ and $NH_3$, concentration of PM10 and its major components $NO_3^-$ and $NH_4^+$ over East Asia were model results. We have clarified the modeled results in this section. In addition, the definition and method for transport distance over the sea, air temperature, RH, and average mixed layer for samples were shown in Section 2.4.

l. 351: 'totally off' is very colloquial. Is there any explanation for this discrepancy?

**Response:** The sentence has been revised into "For reference samples, simulated $NH_4^+$ concentrations sometimes can well reproduce the observational values, but the simulation was sometimes severely deviated from the observation. The deviation could be related to many factors which were out of scope of this study." in l.372-374.

l. 373: I cannot follow here. Why is ammonium excluded in Category 3? Isn't that a contradiction as you mention in the following sentence that you discuss here $N(NH_4^+ + NO_3^-)$?

**Response:** We are sorry for the confusion. The sentence has been revised into "The dry deposition fluxes of particulate $N_{NH4++NO3-}$ decreased by 50%, on average, in Categories 2 and 3, although the fluxes of ammonium of two samples in Category 3 increased." in l. 397-399.

l. 374: 'A larger decrease' than what? Please clarify.

**Response:** The sentence has been revised into "A larger decrease against the reference in the flux of nitrate was present in Categories 2 and 3, i.e., decreases of 73% and 46%, respectively." in l. 399-400.

l. 378 ff: Again, it is not clear whether the following text is based on observations or measurements. Please clarify.

**Response:** In the beginning of this section, we have stated that we calculated the dry deposition fluxes of aerosols particles, $N_{NH4++NO3-}$ and metal elements during dust and reference periods using the measured component concentrations and modeled dry deposition velocities (l. 385-387). And we have clarified the calculated values in this section.

Final comment: What are the main conclusions of your study? They should be summarized in a separate conclusion section after Section 5.

**Response:** We have summarized a separate conclusion in Section 5.

[revised manuscript text omitted]

[a]Residence time of the air mass passing over parts of highly polluted regions according to the trajectories of samples.

[b]Average air temperature with the definition in Section 2.4.

[c]Average relative humidity with the definition in Section 2.4.

[d]Reference samples collected on days immediately before or after dust event

**Table 8.** Dry deposition of TSP (mg/m$^2$/month), $N_{NH4++NO3-}$  (mg N/m$^2$/month) and some toxic trace metals (mg/m$^2$/month) on dust and reference days.

| | Dry deposition flux | | | | | | | |
|---|---|---|---|---|---|---|---|---|
| | TSP | $NO_3^-$ -N | $NH_4^+$-N | $N_{NH4++NO3^-}$ | Fe | Cu | Pb | Zn |
| Category 1[a] | 8,000± 1800 | 65±9 | 24±14 | 90±17 | 533±179 | 2±0.3 | 0.3±0.3 | 6±2 |
| Category 2[a] | 18000± 11,000 | 13±18 | 8±4 | 21±22 | 1300±1000 | 3±2 | 0.08±0.04 | 4±1 |
| Category 3[a] | 29,000± 31,000 | 26±6 | 17±8 | 42±12 | 2100±2200 | 6±1 | 0.20±0.02 | 5±3 |
| Non-dust | 2,800± 700 | 48±33 | 15±8 | 63±39 | 190±110 | 1±1 | 0.09±0.1 | 5±4 |

[a]For the characterization of $N_{NH4++NO3}$- concentration and sample information of the category, see Table 3.

**Table 9.** Comparison of dry deposition flux and normalized flux of TSP (mg/m$^2$/month) and N$_{NH4++NO3-}$
(mg N/m$^2$/month) with observations from other studies (mg N/m$^2$/month)

| Source | Year | Area | | TSP | N$_{NH4++NO3-}$ | Normalized average flux of N$_{NH4++NO3-}$[a] |
|---|---|---|---|---|---|---|
| This work | 2008-2011 | Qingdao, coastal region of the Yellow Sea | Reference day | 2,800±700 | 63±39 | 93.90 |
| | | | Dust day | 10,138±15,940 | 58±36 | 101.39 |
| | | | Average of dust and reference | | | 97.64 |
| Qi et al., 2013 | 2005-2006 | Qingdao, coastal region of the Yellow Sea | Average of nine months samples | 159.2 - 3,172.9 | 1.8-24.5 | 94.75 |
| Zhang et al., 2011 | 1997-2005 | Qingdao | Average of annual samples | | 132 | 99.65 |
| Zhang et al., 2007 | 1999-2003 | The Yellow Sea | | | 11.43 | 9.91 |
| Shi et al., 2013 | 2007 | The Yellow Sea | Reference day | | 19.2 | 132.17 |
| | | | Dust day | | 104.4 | 227.07 |

| Average of dust and Reference | 179.62 |

[a]The calculation method of the normalized flux of $N_{NH4^{+}+NO3^{-}}$ was discussed in Section 3.7.

[Figure]

**Figure 1.** Location of the aerosol and dust sampling sites.

[Figure]

[Figure]

**Figure 2.** Mass concentrations of TSP, Al, Fe and nss-Ca in aerosol samples collected at the
Baguanshan site on dust and  reference days from 2008 to 2011.

[Figure]

[Figure]

**Figure 3**. Mass concentrations of $NH_4^+$ and $NO_3^-$ in aerosol samples collected at the Baguanshan site

on dust and  reference days during March-May in 2008 to 2011.

[Figure]

[Figure]

**Figure 4.** Source profiles of atmospheric aerosol samples collected on reference (a) and dust
(b) days using the PMF model.

[Figure]

**Figure 5.** The 72-h backward trajectories for dust samples from 2008 to 2011(the yellow domains in the map represent the dust source regions in China).

[Figure]

**Figure 6.** Seasonal mean emissions of NOx (a) and NH₃ (b) over East Asia from March-May 2008.

